# A mitotic chromatin phase transition prevents perforation by microtubules

Maximilian W. G. Schneider[1,2 ✉], Bryan A. Gibson[3,5], Shotaro Otsuka[4,5], Maximilian F. D. Spicer[1,2], Mina Petrovic[1,2], Claudia Blaukopf[1], Christoph C. H. Langer[1], Paul Batty[1,2], Thejaswi Nagaraju[1], Lynda K. Doolittle[3], Michael K. Rosen[3] & Daniel W. Gerlich[1 ✉]

Dividing eukaryotic cells package extremely long chromosomal DNA molecules into discrete bodies to enable microtubule-mediated transport of one genome copy to each of the newly forming daughter cells[1–3]. Assembly of mitotic chromosomes involves DNA looping by condensin[4–8] and chromatin compaction by global histone deacetylation[9–13]. Although condensin confers mechanical resistance to spindle pulling forces[14–16], it is not known how histone deacetylation affects material properties and, as a consequence, segregation mechanics of mitotic chromosomes. Here we show how global histone deacetylation at the onset of mitosis induces a chromatin-intrinsic phase transition that endows chromosomes with the physical characteristics necessary for their precise movement during cell division. Deacetylation-mediated compaction of chromatin forms a structure dense in negative charge and allows mitotic chromosomes to resist perforation by microtubules as they are pushed to the metaphase plate. By contrast, hyperacetylated mitotic chromosomes lack a defined surface boundary, are frequently perforated by microtubules and are prone to missegregation. Our study highlights the different contributions of DNA loop formation and chromatin phase separation to genome segregation in dividing cells.

The material properties of individual cell components have a key role in the dynamic self-organization of cellular structures. In mitotic vertebrate cells, chromosomes must acquire material properties that enable microtubules to move them, first to the spindle centre during prometaphase and then to the spindle poles during anaphase[1,17]. Microtubules attach to and pull on chromosomes at specialized kinetochore regions[18], whereas microtubules contacting chromosome arms generate polar ejection forces that push chromosomes away from the spindle poles[19–22], generating a complex system with high tension[17,23]. Condensins cross-link mitotic chromosomes to confer the mechanical stability required to withstand the tension generated at kinetochores[14–16], but it remains unclear how chromosomes acquire material properties that enable them to resist, and therefore move in response to, polar ejection forces. These material properties must provide chromosome arms with sufficient resistance to prevent penetration by polymerizing microtubule tips, as microtubules growing through the chromatin fibre loops in mitotic chromosomes would result in entanglements that impair segregation.

## Mitotic chromatin excludes microtubules

To investigate how chromosomes resist microtubules acting on their arms, we first studied in human tissue culture cells (HeLa cells) the effect of condensin depletion on the morphology and movement of mitotic chromosomes. To deplete condensin, we modified all endogenous alleles of its essential structural maintenance of chromosomes 4 (SMC4) subunit with a C-terminal auxin-inducible degron (mAID) and HaloTag for visualization and added the auxin analogue 5-PhIAA at 2.5 h before mitotic entry to induce efficient degradation (Extended Data Fig. 1a–c). We visualized the spindle using silicon-rhodamine (SiR)–tubulin and stained DNA with Hoechst 33342 to determine the position of chromosomes. In condensin-depleted mitotic cells, we observed an unstructured mass of compact chromatin forming a plate between the spindle poles (Fig. 1a,b and Extended Data Fig. 1d,e). Immunofluorescence staining of the kinetochore marker centromere protein A (CENP-A) and the spindle pole component pericentrin further showed that many kinetochores were detached from the bulk mass of chromatin and displaced towards the spindle poles, whereas, in control cells, all of the kinetochores were closely linked with chromosomes at the metaphase plate (Extended Data Fig. 1f,g). These observations confirm that condensin is required for chromosomes to resist tension generated at kinetochores[14–16] and further show that bulk chromatin is positioned at the spindle centre by a mechanism that is independent of tightly associated kinetochores.

The chromatin of condensin-depleted cells might be moved by polar ejection forces. To study how chromosomes are moved by the mitotic spindle, we imaged mitotic entry of HeLa cells stably expressing mCherry-tagged core histone 2B (H2B–mCherry) and eGFP-tagged

[1]Institute of Molecular Biotechnology of the Austrian Academy of Sciences, Vienna BioCenter, Vienna, Austria. [2]Vienna BioCenter PhD Program, Doctoral School of the University of Vienna and Medical University of Vienna, Vienna, Austria. [3]Department of Biophysics and Howard Hughes Medical Institute, University of Texas Southwestern Medical Center, Dallas, USA. [4]Max Perutz Labs, a joint venture of the University of Vienna and the Medical University of Vienna, Vienna BioCenter, Vienna, Austria. [5]These authors contributed equally: Bryan A. Gibson, Shotaro Otsuka. ✉e-mail: maximilian.schneider@imba.oeaw.ac.at; daniel.gerlich@imba.oeaw.ac.at

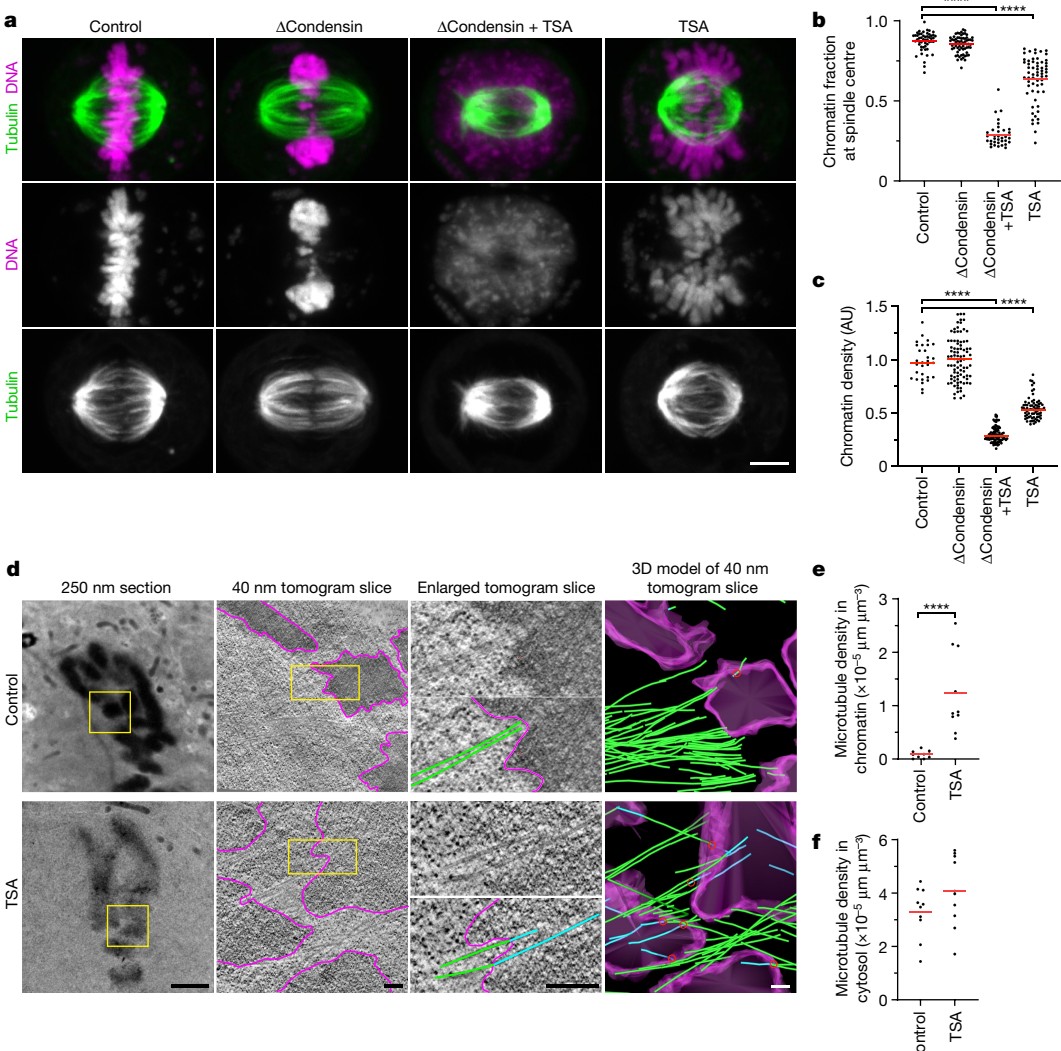

**Fig. 1 | Acetylation-regulated chromatin compaction prevents microtubule perforation in mitosis. a**, The contribution of condensin and histone deacetylases to mitotic chromosome compaction and congression to the spindle centre. HeLa cells with homozygously mAID-tagged SMC4 were treated with 5-PhIAA to deplete condensin (ΔCondensin) or with TSA to suppress mitotic histone deacetylation as indicated. Live-cell images with microtubules stained by SiR–tubulin; DNA was stained with Hoechst 33342. Projection of 5 z-sections. **b**, Quantification of chromosome congression by the fraction of chromatin localizing to the central spindle region. n = 51 (control), n = 65 (ΔCondensin), n = 34 (ΔCondensin + TSA), n = 61 (TSA) cells. The bars indicate the mean. Significance was tested using two-tailed Mann–Whitney U-tests (P < 10⁻¹⁵ (ΔCondensin + TSA); P < 10⁻¹⁵ (TSA); precision limit of floating-point arithmetic). **c**, Quantification of chromatin density in cells treated as described

in **a**. n = 31 (control), n = 89 (ΔCondensin), n = 99 (ΔCondensin + TSA) and n = 74 (TSA) cells. The bars indicate the mean. Significance was tested using two-tailed Mann–Whitney U-tests (P < 10⁻¹⁵ (ΔCondensin + TSA); P < 10⁻¹⁵ (TSA); precision limit of floating-point arithmetic). AU, arbitrary units. **d**, Electron tomography analysis of wild-type prometaphase HeLa cells in the absence or presence of TSA. Magenta, chromatin surfaces; green, microtubules in cytoplasm; cyan, microtubules in chromatin. The red circles show the perforation sites. **e,f**, Quantification of microtubule density in chromatin (**e**) and cytoplasmic (**f**) regions as shown in **d**. n = 10 tomograms from 7 cells for each condition. The bars indicate the mean. Significance was tested using two-tailed Mann–Whitney U-tests (P = 1.083 × 10⁻⁵ (**e**); P = 0.247 (**f**)). Biological replicates: n = 2 (**a–f**). Scale bars, 5 μm (**a**), 2 μm (**d**, 250 nm section); 200 nm (tomogram slices and 3D model).

CENP-A, and visualized microtubules by SiR–tubulin. As it is difficult to distinguish the effect of poleward and anti-poleward forces in a bipolar spindle configuration, we induced a monopolar spindle geometry by inhibiting kinesin-5 using S-trityl-L-cysteine (STLC). When condensin-expressing control cells entered mitosis, the chromosomes first moved towards the spindle pole shortly after nuclear disassembly and then arranged in a rosette with the kinetochores facing towards the pole and the chromosome arms facing away from the pole, such that the region surrounding the spindle pole remained free of chromosomes (Extended Data Fig. 2a,d and Supplementary Video 1). This arrangement is consistent with a balance between pole-directed microtubule pulling at the kinetochores and polar ejection forces pushing on the chromosome arms[19–23]. When condensin-depleted cells entered mitosis,

chromatin formed a compact mass that moved away from the spindle pole, whereas the kinetochores approached the spindle pole, resulting in the detachment of a large fraction of kinetochores from the bulk mass of chromatin (Extended Data Fig. 2b,d,e and Supplementary Video 2). Thus, condensin-depleted chromatin remains responsive to polar ejection forces, whereas it is not stiff enough to resist the tension generated at the kinetochores.

The mechanical resistance of condensin-depleted chromatin towards polar ejection forces might arise from nucleosome-mediated interactions in the chromatin fibre. Nucleosomal interactions are thought to increase when histones are deacetylated during mitotic entry, contributing to global chromatin compaction[9–11,24]. To investigate how acetylation affects the structure and movement of mitotic chromosomes,

we treated condensin-depleted cells with 5 μM of the pan histone deacetylase inhibitor trichostatin A (TSA) 2.5 h before mitotic entry to induce broad hyperacetylation of histone lysine residues (Extended Data Fig. 3a–d), whereby the short duration of the treatment did not induce apoptosis or DNA double-stranded breaks (Extended Data Fig. 3e–h). The hyperacetylated chromatin of condensin-depleted mitotic cells did not compact or enrich at the spindle centre but instead diffusely distributed throughout the cytoplasm, such that spindle pole regions were no longer clear of chromatin (Fig. 1a,b and Extended Data Fig. 1d–g). Treatment with 500 nM TSA also effectively suppressed chromatin compaction and localization to the spindle centre, whereas treatment with 500 nM TSA followed by 8 h removal of TSA resulted in normal chromosome morphology in mitosis, validating the specificity and reversibility of the phenotypes (Extended Data Fig. 4a–d). A complementary approach to induce histone hyperacetylation by overexpressing the histone acetyltransferase p300 also resulted in chromatin decompaction phenotypes similar to those induced by TSA (Extended Data Fig. 5). Thus, histone deacetylation is important for chromatin compaction and positioning at the spindle centre.

To investigate more specifically how hyperacetylation affects the response of chromosomes to polar ejection forces, we imaged mitotic entry of live condensin-depleted cells treated with TSA, using STLC to induce a monopolar spindle geometry. The chromatin of these cells remained diffuse and completely decompacted, while the spindle aster assembled and moved into the decondensed chromatin regions. Kinetochores then moved towards the spindle pole, but the bulk mass of chromatin was not displaced towards the cell periphery such that the regions surrounding the spindle poles did not clear from chromatin, and the kinetochores remained embedded in the chromatin (Extended Data Fig. 2c–e and Supplementary Video 3). Thus, deacetylation has an important role in the response of chromatin to polar ejection forces.

The diffuse distribution of chromatin throughout the cytoplasm resulting from TSA treatment in condensin-depleted cells is in stark contrast to relatively mild mitotic chromosome decompaction phenotypes previously observed in condensin-expressing wild-type cells[9,10]. We hypothesized that the moderate level of TSA-induced decompaction in wild-type cells might be due to condensin-mediated linkages that counteract the dispersion of chromatin fibres. To investigate this, we analysed how TSA affects chromatin density in the presence or absence of condensin. In condensin-depleted mitotic cells, TSA reduced chromatin density to 29% compared with cells that were not treated with TSA, whereas, in the presence of condensin, TSA reduced chromatin density only to 53% and chromosomes remained visible as thread-like structures (Fig. 1a,c and Extended Data Fig. 4a–d). Condensin depletion alone did not reduce mitotic chromatin density (Fig. 1a,c), consistent with previous observations[14,25,26]. Thus, histone deacetylation is necessary and sufficient for complete compaction of mitotic chromatin even in the absence of condensin. By contrast, condensin is neither necessary nor sufficient for complete chromatin compaction during mitosis, yet it can concentrate chromatin to some extent even when histones are hyperacetylated.

We next investigated whether chromatin compaction through histone deacetylation might be necessary to prevent microtubules from penetrating chromosomes. To investigate how acetylation affects the access of microtubules to chromosomes, we performed electron tomography of mitotic HeLa cells. Chromosomes of unperturbed cells appeared as homogeneously compacted bodies with a sharp surface boundary, and 3D segmentation showed that they were almost never penetrated by microtubules (Fig. 1d,e and Extended Data Fig. 6a). By contrast, chromosomes of TSA-treated cells appeared to be less compact, particularly towards the periphery, and microtubules grew extensively through the chromatin (Fig. 1d–f and Extended Data Fig. 6b–d). Thus, active histone deacetylases are required to keep microtubules out of chromosome bodies, providing a basis for resistance towards polar ejection forces.

Microtubule perforation into mitotic chromosomes is expected to cause entanglements between chromatin fibre loops and spindle microtubules that impair chromosome segregation. To investigate how TSA-induced hyperacetylation alone affects chromosome segregation, we recorded time-lapse videos of HeLa cells expressing H2B–mCherry. TSA severely delayed chromosome congression and initiation of anaphase and caused a high incidence of lagging chromosomes (Extended Data Fig. 6e–h). Active histone deacetylases are therefore essential for faithful chromosome segregation.

## A mitotic chromatin phase transition

To elucidate the mechanism that underlies microtubule exclusion from mitotic chromosomes, we investigated how acetylation affects the material properties of chromatin. Recent research has demonstrated that phase-separated biomolecular condensates can form highly dense structures that exert and resist forces[27]. Moreover, we found that purified nucleosome arrays condense into liquid droplets in physiological salt solutions, and these condensates dissolve after acetylation[13], supporting the idea that mitotic chromatin might form an immiscible phase. However, endogenous chromatin contains thousands of different proteins[28] and is subject to various post-translational modifications besides acetylation[12,24], with unknown effects on phase separation. Under which conditions endogenous chromatin might undergo a phase transition, how such chromatin phase transition might affect the material properties of chromosomes and what might be the functional relevance are unclear.

To test the hypothesis that mitotic chromosomes are highly complex biomolecular condensates, we reasoned that fragmenting mitotic chromatin using a nuclease might relieve constraints imposed by the very long length of chromosomes to unveil the underlying phase transition. To investigate this, we developed a live-cell chromatin fragmentation assay based on microinjection of the restriction enzyme AluI. Shortly after microinjection of AluI into mitotic cells, chromosomes lost their elongated shape, forming round condensates that fused to one another, consistent with a liquid-like state (Fig. 2a and Supplementary Video 4). Notably, AluI injection did not decrease chromatin density (Fig. 2b). As chromosome fragmentation is expected to induce cell death in the long run, we imaged chromatin only a few minutes after AluI injection and validated by the early apoptosis marker polarity sensitive indicator of viability and apoptosis (pSIVA) that within this short time frame cells do not enter apoptosis (Extended Data Fig. 7a,b). Overall, these experiments show that the integrity of the chromatin fibre is not required for full compaction of the mitotic chromatin, consistent with a phase separation mechanism of compaction.

To assess the mobility of chromatin, we performed a fluorescence recovery after photobleaching analysis of H2B–mCherry. Native mitotic chromosomes recovered very little H2B–mCherry fluorescence after photobleaching, consistent with constrained mobility within a large polymer network. However, after AluI digestion, H2B–mCherry recovered rapidly and completely from photobleaching (Fig. 2c,d), consistent with a liquid state. Imaging Halo-tagged SMC4 further showed that condensin did not form axial structures inside the chromatin condensates and instead evenly distributed throughout the cell, validating efficient chromosome fragmentation by AluI (Extended Data Fig. 7c,d). Thus, mitotic chromatin is insoluble in the cytoplasm and, when the long-range constraints of the fibre network are eliminated, the short-range dynamics manifest in liquid-like behaviour.

To test whether the formation of an immiscible chromatin phase is suppressed by acetylation, we treated cells with TSA before mitotic entry and then injected AluI. This resulted in homogeneously dispersed chromatin fragments with almost no local condensates (Fig. 2e,f and Supplementary Video 5; the few remaining chromatin foci might represent constitutive heterochromatin that is known to be refractory to TSA-induced hyperacetylation[29]); a similar phenotype was observed

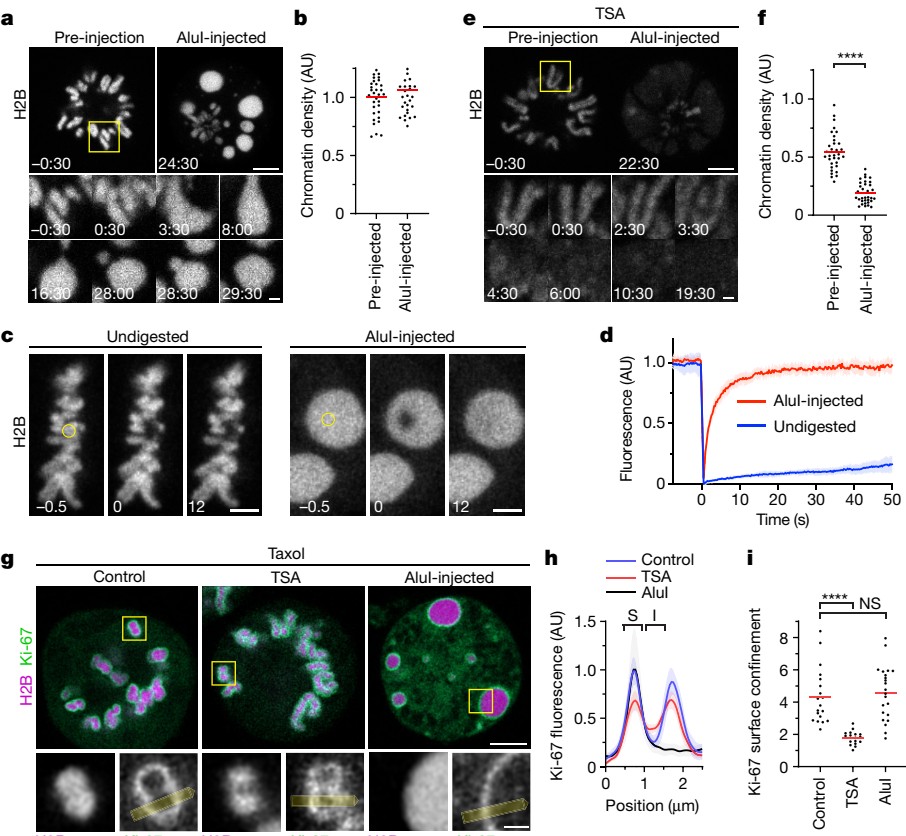

**Fig. 2 | Acetylation regulates chromatin solubility in mitotic cytoplasm.**
**a**, Chromosome fragmentation in live mitotic HeLa cells by AluI injection
($t = 0$ min). Chromatin was visualized with H2B–mCherry. Projection of 3
$z$-sections. Time is shown as min:s. **b**, Quantification of chromatin density for
cells as in **a**. $n = 11$ cells, 3 regions of interest (ROIs) each. The bars indicate the
mean. Significance was tested using a two-tailed Mann–Whitney $U$-test
($P = 0.332$). **c**, Chromatin mobility in undigested metaphase chromosomes and
after AluI injection, measured by fluorescence recovery after photobleaching
in live metaphase cells expressing H2B–mCherry. The circles indicate the
photobleaching region at $t = 0$ s. Time is shown as s. **d**, Quantification of
fluorescence in $n = 8$ (undigested) or $n = 10$ (AluI-digested) cells as described in
**c**. Data are mean ± s.d. **e**, AluI injection as described in **a** for a TSA-treated
mitotic cell. Time is shown as min:s. **f**, Quantification of chromatin density,
normalized to the mean of untreated pre-injection cells shown in **b**. $n = 11$ cells,
3 ROIs each. The bars indicate the mean. Significance was tested using a

two-tailed Mann–Whitney $U$-test ($P < 10^{-15}$; precision limit of floating-point
arithmetic). **g–i**, Ki-67 localization in mitotic cells. **g**, HeLa cells expressing
eGFP–Ki-67 and H2B–mCherry were treated with taxol for mitotic arrest
(control); cells were treated with TSA or microinjected with AluI as indicated.
Ki-67 localization was analysed in chromosomes oriented perpendicularly to
the optical plane (insets). **h**, Line profiles across the chromatin–cytoplasm
boundary as indicated by the yellow lines in **g** were aligned to the first peak in
eGFP–Ki-67 fluorescence and normalized to the mean of Ki-67 fluorescence at
the first peak of control. $n = 19$ (control), $n = 24$ (TSA) and $n = 22$ (AluI) cells. Data
are mean ± s.d. **i**, Quantification of Ki-67 surface confinement by the ratio of Ki-
67 fluorescence on the surface (S) over inside (I). $n = 19$ (control), $n = 24$ (TSA)
and $n = 22$ (AluI) cells. The bars indicate the mean. Significance was tested using
two-tailed Mann–Whitney $U$-tests ($P = 9.305 \times 10^{-10}$ (TSA); $P = 0.476$ (AluI)).
Biological replicates: $n = 3$ (**a**,**b**,**g**–**i**); $n = 2$ (**c**–**f**). Scale bars, 5 μm (**a**, **e** and **g**, main
images), 1 μm (**a**, **e** and **g**, insets) and 3 μm (**c**).

in cells overexpressing p300 to induce histone hyperacetylation
(Extended Data Fig. 7e,f). By contrast, depletion of SMC4 had no
effect on chromatin droplet formation after AluI-mediated fragmen-
tation (Extended Data Fig. 7g,h). Thus, deacetylation is a major factor
in establishing an immiscible chromatin phase in mitotic cells, while
condensin is not required.

We next investigated whether chromatin fragments generated in
interphase nuclei undergo a solubility phase transition after progres-
sion to mitosis. To monitor cell cycle stages, we used a fluorescence
resonance energy transfer (FRET) biosensor for a key mitotic kinase,
aurora B. As chromosome fragmentation blocks mitotic entry owing
to DNA damage signalling, we applied chemical inhibitors to induce
an interphase-to-mitosis transition. We first synchronized cells to G2
using the cyclin dependent kinase 1 (CDK1) inhibitor RO3306 and then
induced a mitosis-like state by removing RO3306 for CDK1 activation
and simultaneously inhibiting counteracting protein phosphatase 2
(PP2A) and protein phosphatase 1 (PP1) using okadaic acid. Mitotic entry
was demonstrated by the aurora B FRET biosensor signal (Extended
Data Fig. 8a,b and Supplementary Video 6). Injection of AluI into G2

cell nuclei resulted in homogeneously distributed chromatin, con-
sistent with a soluble state; furthermore, after induction of mitosis,
chromatin fragments formed spherical condensates that were as dense
as intact chromosomes (Extended Data Fig. 8c,d and Supplementary
Video 7), whereas control cells in which RO3306 was not replaced by
Okadaic acid maintained homogeneously dissolved chromatin frag-
ments (Extended Data Fig. 8e,f). These observations support a model
in which a global reduction in solubility drives chromatin compaction
at the interphase-to-mitosis transition.

To determine whether the loss of chromatin solubility after mitotic
entry depends on deacetylation, we inhibited histone deacetylases
using TSA before injecting AluI into G2-synchronized cells. As in control
cells, chromatin fragments were homogeneously distributed through-
out the nucleus in G2, but mitotic induction did not lead to the forma-
tion of condensed foci (Extended Data Fig. 8g,h and Supplementary
Video 8). Thus, active histone deacetylases are essential to forming an
immiscible chromatin phase after mitotic entry.

Acetylation of chromatin might regulate solubility in the cytoplasm
directly or through other chromatin-associated components. To assess

the effect of histone acetylation on chromatin solubility more specifically, we used synthetic nucleosome arrays[13] as probes. We generated fluorescently labelled arrays of 12 unmodified naive nucleosomes as well as similar arrays that were labelled with a distinct fluorophore and acetylated in vitro with recombinant p300 acetyltransferase. In vitro, the unmodified nucleosome arrays form liquid condensates under physiological salt concentrations, in contrast to the acetylated nucleosome arrays[13] (Extended Data Fig. 8i,j). After co-injection of these nucleosome arrays into live mitotic cells, unmodified arrays almost completely partitioned into the mitotic chromatin phase, whereas acetylated nucleosome arrays predominantly dissolved in the cytoplasm (Extended Data Fig. 8k,l). Thus, acetylation-sensitive interchromatin interactions alone are sufficient to recruit chromatin into mitotic chromosomes, supporting a model in which histone acetylation is a direct regulator of chromatin solubility in the cytoplasm.

To further characterize the boundary between the chromatin and cytoplasm, we studied a component of the mitotic chromosome periphery—the protein Ki-67[30–32]. Ki-67 has an N-terminal region that is excluded from mitotic chromatin and a C-terminal region that is attracted to mitotic chromatin[31]. According to our chromatin phase-separation model, the targeting of Ki-67 to the chromosome surface should arise from its amphiphilic attraction to the phase boundary between the chromatin and cytoplasm and therefore be independent of higher-order chromatin fibre folding but sensitive to chromatin solubilization. To test this hypothesis, we studied the localization of Ki-67 in live cells, finding that Ki-67 still enriched at a sharp boundary around chromatin droplets after AluI injection or around chromatin of cells depleted of condensin, whereas TSA substantially reduced confinement of Ki-67 to the chromatin surface (Fig. 2e–g and Extended Data Fig. 8m,n). We also tested whether Ki-67 is required for the formation of an insoluble chromatin phase. After injection of AluI into *MKI67*-knockout (encoding Ki-67) cells[31], we observed spherical chromatin condensates similar to those in wild-type cells (Extended Data Fig. 8o,p). Thus, Ki-67 targets to the surface of mitotic chromosomes through its amphiphilic attraction to chromatin and cytoplasmic phases, whereas it is not required for the formation of an immiscible mitotic chromatin phase.

Overall, our data show that deacetylation during mitotic entry induces global chromatin phase separation. This chromatin phase separation mediates full compaction of mitotic chromatin, independently of the integrity of the chromatin fibre, condensin-mediated DNA looping[4–8] or potential higher-order chromatin fibre coils[33–35]. Chromosomes have been described as hydrogels, in which a flexible chromatin fibre is cross-linked by condensin and expanded throughout its volume by an aqueous liquid component[36–40]. By elucidating acetylation as a key regulator of chromatin solubility, our study provides a molecular explanation of how such chromatin hydrogels collapse into compact bodies with a sharp boundary in mitosis.

## Macromolecular exclusion from chromatin

To investigate the mechanism underlying microtubule exclusion from mitotic chromosomes, we analysed how soluble tubulin partitions relative to mitotic chromatin. We microinjected fluorescently labelled tubulin into live mitotic cells and applied nocodazole to suppress microtubule polymerization (Fig. 3a). Soluble tubulin was much less concentrated inside mitotic chromosomes compared with in the surrounding cytoplasm (Fig. 3a,b). By contrast, soluble tubulin was not excluded from hyperacetylated chromosomes in TSA-treated cells (Fig. 3a,b). Thus, the immiscible chromatin compartment formed by deacetylated mitotic chromatin excludes soluble tubulin.

To determine whether the exclusion of tubulin is due to a limiting pore size in chromosomes, we expressed DsRed, a fluorescent protein that forms tetramers slightly larger than tubulin dimers[41]. In contrast to tubulin, DsRed distributed evenly across chromatin and cytoplasm

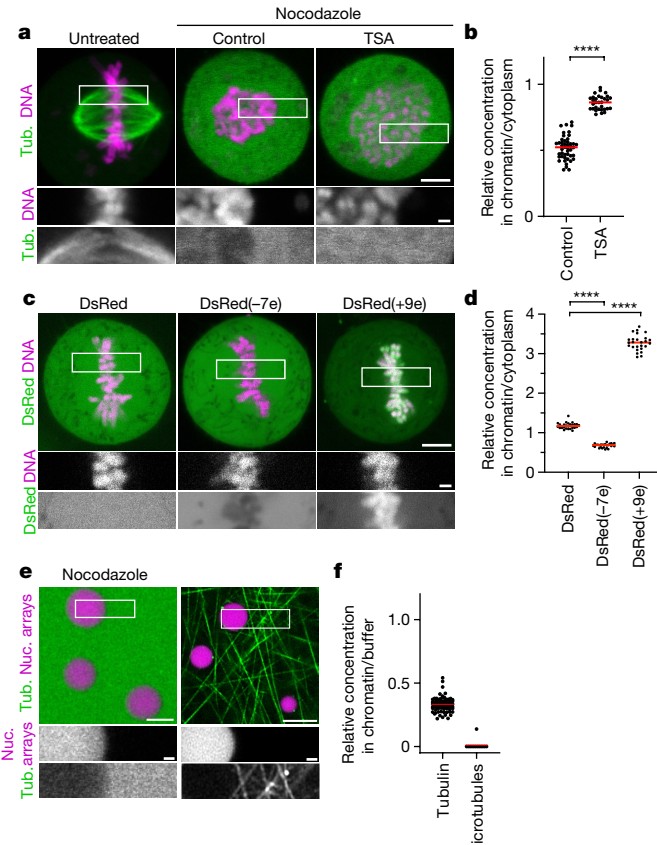

**Fig. 3 | Chromatin condensates limit access of tubulin and other negatively charged macromolecules. a**, The localization of tubulin (tub.) relative to mitotic chromosomes. Rhodamine-labelled tubulin was injected into live mitotic cells that were untreated, treated with nocodazole alone (control) or in combination with TSA. **b**, Quantification of the tubulin concentration for the data shown in **a**. $n = 27$ cells. The bars indicate the mean. Significance was tested using a two-tailed Mann–Whitney $U$-test ($P < 1 \times 10^{-15}$; precision limit of floating-point arithmetic). **c**, Live-cell images of a HeLa cell expressing DsRed or DsRed fused at its N terminus to electrically charged polypeptides. DNA was stained with Hoechst 33342. The numbers in parentheses indicate the predicted elementary charge of the tetramers formed by DsRed fusion constructs. **d**, Quantification of DsRed concentration for the data shown in **c**. $n = 26$ (DsRed), $n = 26$ (DsRed(−7e)), $n = 26$ (DsRed(+9e)) cells. The bars indicate the mean. Significance was tested using two-tailed Mann–Whitney $U$-tests ($P = 0.4 \times 10^{-14}$ (DsRed(−7e)); $P = 0.4 \times 10^{-14}$ (DsRed(+9e)). **e**, The localization of tubulin relative to reconstituted nucleosome (nuc.) array droplets. Nucleosome array droplets were formed by incubation in phase separation buffer and fluorescently labelled tubulin was then added in the presence of nocodazole, or in the absence of nocodazole with subsequent temperature increase to 20 °C to induce microtubule polymerization. **f**, Quantification of the tubulin concentration or microtubule density in nucleosome array condensates relative to buffer for the data shown in **e**. $n = 94$ droplets, $n = 13$ fields of polymerized microtubules. The bars indicate the mean. Biological replicates: $n = 2$ (**a**–**d**); $n = 3$ (**e**,**f**). Technical replicates: $n = 3$ (**a**,**b**); $n = 2$ (**c**,**d**); $n = 3$ (**e**,**f**). For **a**, **c** and **e**, scale bars, 5 μm (main images) and 1 μm (insets).

(Fig. 3c,d). Thus, macromolecules in the size range of tubulin are not generally excluded from mitotic chromatin.

The selective exclusion of tubulin but not DsRed from mitotic chromatin suggests that specific molecular features control access. Tubulin is highly negatively charged at physiological pH (7.2), in contrast to near-neutrally charged DsRed, raising the possibility that macromolecular access is governed by electrostatic interactions. A high concentration of an overall negative electrical charge of chromatin[42] in an immiscible condensate might therefore repel negatively charged

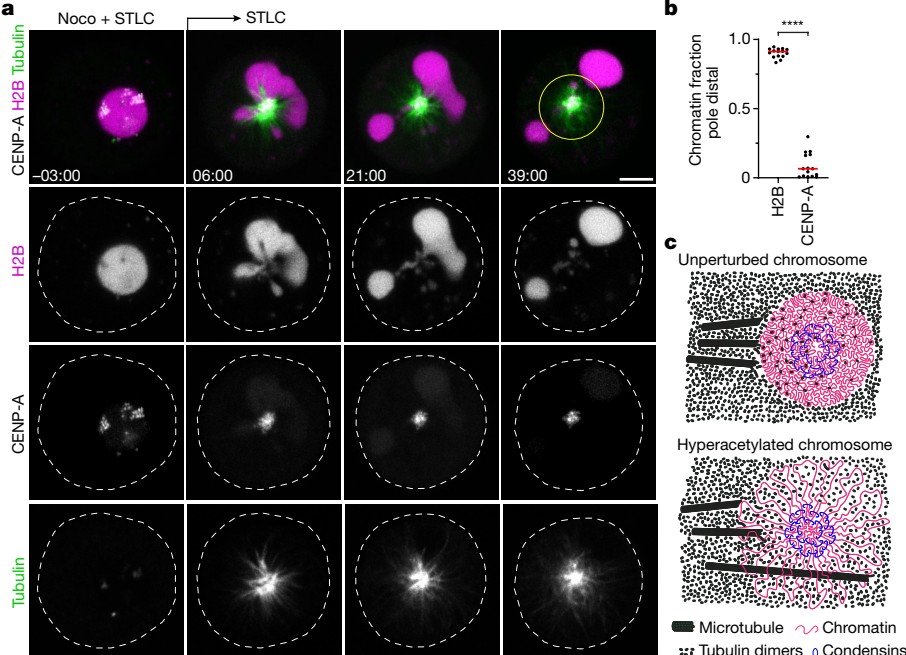

**Fig. 4 | Microtubules push liquified chromatin away from the spindle pole.**
**a**, Time-lapse microscopy analysis of liquified chromatin during monopolar spindle assembly. AluI was injected into live mitotic HeLa cells expressing H2B–mCherry and meGFP–CENP-A, stained with SiR-tubulin, in the presence of nocodazole (noco) and STLC. Nocodazole was then removed at $t = 0$ min during time-lapse imaging to induce monopolar spindle assembly. Projection of 5 z-sections. Time is shown as min:s. **b**, Quantification of bulk chromatin (H2B–mCherry) and centromeric chromatin (meGFP–CENP-A) localizing at the cell periphery relative to the region around the spindle monopole at $t = 36$ min. $n = 15$ cells. The bars indicate the mean. Significance was tested by a two-tailed Mann–Whitney $U$-test ($P = 1.289 \times 10^{-8}$). **c**, Model of chromatin compaction and condensin-mediated DNA looping in mitotic chromosome and spindle assembly. The illustration shows a top-down view of a chromosome cross-section. Biological replicates: $n = 3$ (**a**,**b**). Scale bars, 5 μm.

cytoplasmic proteins. To investigate how charge affects macromolecular access to mitotic chromatin, we fused charged polypeptides to the N terminus of DsRed. DsRed fused to a negatively charged polypeptide (overall predicted charge on the tetramer, −7e) was excluded from mitotic chromatin, whereas DsRed fused to a positively charged polypeptide (overall charge on the tetramer, +9e) concentrated inside chromatin (Fig. 3c,d). Similarly, we found that negatively charged monomeric enhanced green fluorescent protein (meGFP) or negatively charged 4 kDa dextrans were also excluded from mitotic chromosomes, whereas a positively charged surface mutant scGFP (super-charged GFP, +7e)[43] or positively charged 4 kDa dextrans concentrated in the mitotic chromosomes (Extended Data Fig. 9a–d). Two negatively charged microtubule-regulating proteins, p80-katanin and the microtubule plus-tip protein end binding 3 (EB3), were also excluded from the mitotic chromatin, whereby TSA treatment resulted in a less efficient exclusion from the mitotic chromatin (Extended Data Fig. 9e–h). To investigate how molecular size affects the partitioning relative to mitotic chromatin, we injected negatively charged dextrans of 20 or 70 kDa molecular mass. Both dextrans were excluded from chromatin, and the exclusion was more efficient at greater molecular mass; TSA treatment reduced the exclusion of both dextrans from mitotic chromosomes (Extended Data Fig. 9i–l). Thus, electrical charge and molecular size are key determinants of macromolecular access to mitotic chromatin.

The exclusion of negatively charged macromolecules from mitotic chromosomes might be mediated by nucleosome fibres alone, or it might involve other chromosome-associated factors. To investigate how tubulin interacts with intrinsic chromatin condensates formed by reconstituted nucleosome arrays, we reconstituted droplets of recombinant nucleosome arrays in vitro[13] and then added rhodamine-labelled tubulin in the presence of nocodazole. Soluble tubulin was indeed efficiently excluded from nucleosome array condensates (Fig. 3e,f). Consistent with this observation, negatively charged meGFP or dextrans were also excluded from nucleosome array condensates, whereas a positively charged scGFP mutant or positively charged dextrans concentrated in nucleosome array condensates (Extended Data Fig. 9m–p). The exclusion of negatively charged macromolecules, including tubulin, is therefore an intrinsic property of condensed nucleosome fibres.

The efficient exclusion of free tubulin suggests that weak affinity interactions in liquid chromatin condensates might be sufficient to limit microtubule polymerization. To investigate how microtubules interact with reconstituted chromatin droplets in vitro, we added purified tubulin and then induced microtubule polymerization by increasing the temperature. Microtubules formed a dense microtubule network, yet they almost never grew into chromatin condensates (Fig. 3e,f). Thus, chromatin-intrinsic material properties impose a highly impermeable barrier to microtubule polymerization independently of condensin or other chromosome-associated factors.

## Microtubules push liquid chromatin

Polymerizing microtubules exert substantial pushing forces after contact with stiff surfaces[44]. As microtubules did not grow into condensates of purified chromatin fragments in vitro, we wondered whether the surface tension of liquified endogenous chromatin is strong enough to allow microtubule-based pushing. If this were the case, then droplets of digested chromosomes should be pushed away by polar ejection forces of growing astral microtubules. To test this hypothesis, we injected AluI into mitotic cells treated with nocodazole and then washed out the nocodazole to induce microtubule polymerization, while applying STLC to induce a monopolar astral spindle geometry. Time-lapse imaging showed that, initially, all chromatin resided in a single droplet at the cell centre but, soon after nocodazole washout, the chromatin split into several droplets that moved away from growing microtubule asters (Extended Data Fig. 10a,b). Thus, liquid chromatin condensates can be pushed by polymerizing astral microtubules.

Astral microtubules might directly push on chromatin droplets by polymerizing against the chromatin phase boundary or they might couple to chromatin through chromokinesins[19–23,45–48]. To investigate how the spindle moves chromatin droplets, we used RNA interference (RNAi) to co-deplete two chromokinesins that are major contributors to the polar ejection force, Kid and kinesin family member 4A (Kif4a)[22,23,45,47,48]. After AluI-injection and nocodazole washout in the presence of STLC, chromosome droplets moved to the chromosome periphery as efficiently as in the control cells (Extended Data Fig. 10c–e). Thus, spindle asters can move liquid chromatin independently of the chromokinesins Kid and Kif4a, potentially by polymerizing microtubules pushing against the chromatin phase boundary.

Polar ejection forces pushing on chromosome arms are counteracted by pole-directed pulling at centromeres. To investigate how liquified chromatin responds to pulling forces at the kinetochores, we injected AluI into cells expressing eGFP–CENP-A and H2B–mCherry in the presence of nocodazole. We then removed nocodazole to induce monopolar spindle assembly in the presence of STLC. The bulk mass of chromatin, visualized through H2B–mCherry, rapidly moved towards the cell periphery, whereas several much smaller chromatin condensates enriched in eGFP–CENP-A remained close to the spindle monopole (Fig. 4a,b and Supplementary Video 9). Thus, bulk chromatin is pushed away from the spindle poles, while centromeres are transported towards the spindle pole. When the continuous connection between centromeres and the remaining chromosome is lost, bulk chromatin and centromeric chromatin physically separate according to the locally dominating forces.

## Conclusions

Here we show that a substantial global reduction in chromatin solubility caused by deacetylation during mitotic entry converts chromosomes into phase-separated bodies rather than loose bottlebrush structures with chromatin fibre loops extending into the cytoplasm (Fig. 4c). The immiscible mitotic chromatin forms a surface that provides resistance to microtubule perforation while allowing local chromatin fibre sliding internally, as required for continuous dynamic loop formation by condensin[6,14,49]. In parallel, condensin-mediated linkages establish a hydrogel that withstands tension generated at kinetochores[14–16]. Jointly, these molecular activities shape discrete chromosome bodies with a defined surface despite continuous internal remodelling of the chromatin fibre.

Our results show that mitotic cells contain three principal domains with distinct microtubule polymerization propensity: the centrosome matrix, which concentrates soluble tubulin to promote nucleation at the poles[50]; the cytoplasm, which is highly permissive for microtubule growth and amplification; and a chromatin compartment that is refractory towards polymerization. Thus, our study provides a unified view of how chromatin looping by condensin and compaction through a phase transition driven by acetylation-sensitive nucleosome interactions contribute to the material properties and mechanical functions of mitotic chromosomes. It will be interesting to investigate how chromatin adapts its material properties to other physiological processes that involve compaction, such as apoptosis.

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

# Methods

## Cell lines and cell culture

All of the cell lines used in this study have been regularly tested negative for mycoplasm contamination. All cell lines in this study were derived from a HeLa 'Kyoto' cell line that was previously described in ref. [51]. The cells were cultured in Dulbecco's modified Eagle medium (DMEM) (IMP/IMBA/GMI Molecular Biology Service/Media Kitchen) containing 10% (v/v) fetal bovine serum (FBS; Gibco, 10270-106, 2078432), 1% (v/v) penicillin–streptomycin (Sigma-Aldrich), 1% (v/v) GlutaMAX (Gibco; 35050038) and selected antibiotics according to the respective expression constructs: blasticidin S (6 μg ml$^{-1}$, Thermo Fisher Scientific), puromycin (0.5 μg ml$^{-1}$, Calbiochem), hygromycin B (0.25 mg ml$^{-1}$) and G418 (0.5 mg ml$^{-1}$, Invitrogen). HeLa cells were cultured at 37 °C in a 5% $CO_2$-containing atmosphere. Chromatin was visualized by stable expression of histone 2B labelled with mCherry (Figs. 2a–g and 4 and Extended Data Figs. 2, 6e, 7a–h, 8e–h and 10) or with the aurora B-FRET sensor (CFP/YFP) (Extended Data Fig. 8a–h) or alternatively by labelling with Hoechst 33342 (1.62 μM, Invitrogen, H1399) (Figs. 1a–c, 2m and 3a,c and Extended Data Figs. 1a,d,f, 3a,c,e,g, 4a–d, 5a,c,e and 9a,c,e,g,i,k). Tubulin was visualized by stable expression of an N-terminally tagged eGFP–α-tubulin fusion (Extended Data Figs. 10a,c) or by labelling with SiR–tubulin[52] (Spirochrome, SC002, Figs. 1a and 4a and Extended Data Figs. 4a,c, 5c, 6e and 9e,g (100 nM) and Extended Data Fig. 2a–c (50 nM))[52]. Centromeres were visualized by stable expression of an N-terminally tagged eGFP–CENP-A fusion (Fig. 4a) or an N-terminally tagged meGFP–CENP-A fusion (Extended Data Figs. 2a–c and 6e). Ki-67 was visualized in live cells either by stable expression of endogenous eGFP–Ki-67[31] (Fig. 2e–g) or by transient expression of Ki-67-mNeonGreen (Extended Data Fig. 8m,n). To determine the apoptotic index, cells were stained with pSIVA[53] and prodidium iodide (PI) (Bio-Rad, APO004) (Extended Data Figs. 3e and 7a,b).

Live-cell imaging was performed in DMEM containing 10% (v/v) FBS (Gibco, 10270-106, 2078432), 1% (v/v) penicillin–streptomycin (Sigma-Aldrich) and 1% (v/v) GlutaMAX (Gibco; 35050038) but omitting phenol red and riboflavin to reduce autofluorescence (imaging medium)[51]. Cells were grown in 10 cm or 15 cm Cellstar (Greiner) dishes, in 75 cm$^2$ or 175 cm$^2$ Nunc EasYFlask cell culture flasks (Thermo Fisher Scientific) or in 96-well, 48-well, 12-well or 6-well Nunclon Delta Surface multiwell plates (Thermo Fisher Scientific). For live-cell imaging, cells were cultivated on Nunc LabTek II chambered cover glass (Thermo Fisher Scientific), on μ-Slide 8-well covered coverslips (Ibidi), in μ-Dish 35 mm high imaging dishes with a polymer or glass bottom (Ibidi).

## Generation of stable fluorescent reporter and genetically engineered cell lines

Cell lines stably expressing fluorescently labelled marker proteins were generated by random plasmid integration (for the transfection conditions, see below) or by a lentiviral vector system pseudotyped with a mouse ecotropic envelope that is rodent-restricted (RIEP receptor system). Construction of the RIEP receptor parental cell lines and the subsequent generation of stable cell lines that express fluorescent marker proteins was performed as described previously[54].

HeLa genome editing was performed using a CRISPR–Cas9-mediated integration approach, using the chimeric Cas9–human Geminin fusion (Cas9–hGem) modification for enhanced editing efficiency[55]. Single-guide RNA (sgRNA) was cloned into pX330-U6-Chimeric_BB-CBh-hSpCas9-hGem-P2A-mCherry*aBpil (gift from S. Ameres). For tagging of endogenous *SMC4* genes with an auxin-inducible degron tag[56], a repair template for targeting the C terminus of the protein was designed, with homology flanks of 800 and 901 bp length for the 5′ and 3′ flanks, respectively. The repair template containing the coding sequence for a C-terminal mAID-HaloTag, including mutations of the protospacer adjacent motif, was synthesized as a gBlock (IDT) and cloned into plasmid pCR2.1 for amplification. sgRNA/Cas9 and homology repair template plasmids were co-transfected using the Neon transfection system (Thermo Fisher Scientific). Transfection was performed according to the manufacturer's guidelines (for HeLa cells: pulse voltage, 1.005 V; pulse width, 35 ms; pulse number, 2), using the 100 μl tip cuvette, with 10 μg homology repair template and 10 μg gRNA/Cas9–hGEM for 1 × 10$^6$ cells. Then, 9 days after electroporation, cell pools were stained with Oregon Green HaloTag (Promega, G2801) ligand and single clones were isolated using fluorescence-activated cell sorting (FACS) and sorted into 96-well plates. Genotyping was performed as in a previous study[54].

Integration of the *Os*TIR1(F74G) ligase into the adeno-associated virus integration site (AAVS1, designated safe harbour) locus for establishment of AID2 auxin-inducible degradation of SMC4 was achieved using a sgRNA and homology repair template strategy described in ref. [57]. Expression of the E3-ubiquitin ligase *Os*TIR1(F74G) enables degradation of degron-tagged proteins by labelling them for proteasomal degradation in the presence of a small-molecule auxin analogue. sgRNA was cloned into pX330-U6-Chimeric_BB-CBh-hSpCas9-hGem-P2A-mCherry*aBpil. For integration of the *Os*TIR1(F74G), a homology template containing an OsTIR1(F74G)-SnapTAG-MycA1-NLS and homology flanks of 804 and 837 bp length for the 5′ and 3′ flanks, respectively, was generated. Co-transfection of sgRNA/Cas9 and homology repair template plasmids was performed as described above. Then, 8 days after electroporation, transfected cells were selected with 6 μg ml$^{-1}$ blasticidin. Single clones were isolated from the stable pool by single-cell dilution into 96-well plates. To identify clones that depleted SMC4-mAID-Halo, clones were treated for 90 min with 1 μM 5-Ph-IAA24, stained with Oregon Green HaloTag (Promega, G2801) ligand and then analysed by flow cytometry using the iQue Screener Plus system.

## Transfection of plasmids and small interfering RNA

For expression of fluorescently labelled markers, the respective genes were cloned into bicistronic vectors containing an IRES or a 2A 'self-cleaving' peptide sequence with antibiotic resistance genes that enable the protein of interest and the resistance gene to be expressed from the same transcript. For transient transfection or transfection for subsequent selection and colony picking, plasmids were transfected using X-tremeGENE 9 DNA transfection reagent (Roche, 6365787001) according to the manufacturer's protocol (1 μg plasmid, 4.5 μl X-tremeGENE 9 in 100 μl serum-free OptiMEM) or PEI transfection reagent (1 mg ml$^{-1}$ stock, Polyscience 24765-2, 4 μg of transfection reagent per 1 μg of plasmid). For stable expression, plasmids were transfected using PEI, 10 μg plasmid for one ~70% confluent 15 cm dish and incubated for 48 h before antibiotic selection.

Small interfering RNAs (siRNAs) were delivered with lipofectamine RNAiMax (Invitrogen) according to the manufacturer's instructions. hKid (Kif22) was targeted using 16 nM custom silencer select siRNA (sense strand CAAGCUCACUCGCCUAUUGTT, Thermo Fisher Scientific, including a 3′ overhang TT dinucleotide for increased efficiency). Kif4A was targeted using 16 nM custom silencer select siRNA (sense strand GCAAGAUCCUGAAAGAGAUTT, Thermo Fisher Scientific, including a 3′ overhang TT dinucleotide for increased efficiency). Custom silencer select siXWNeg (sense strand UACGACCGGUCUAUCGUAGTT, Thermo Fisher Scientific, including a 3′ overhanging TT dinucleotide for increased efficiency) was used as a non-targeting siRNA control. Cells were imaged 30 h after siRNA transfection. Co-transfection of *KIF22* and *KIF4A* siRNAs was performed using 16 nM final concentrations each. Both hKid and *KIF4A* siRNAs were described in ref. [22].

## Inhibitors and small molecules

To degrade Smc4-mAID-Halotag, cells were incubated in 1 μM 5-phenylindole-3-acetic acid (5-PhIAA) (Bio Academia, 30-003) for 2.5–3 h. To induce histone hyperacetylation, cells were incubated with 5 μM TSA[58] (Sigma-Aldrich, T8552). To arrest cells in prometaphase with monopolar spindle configuration, cells were incubated in 5 μM

STLC[59] (Enzo Life Sciences, ALX-105-011-M500) for 2–3 h. To arrest cells in prometaphase, cells were incubated for 2–3 h in nocodazole (200 ng ml[−1] for live-cell imaging only) or 100 ng ml[−1] for microinjection and/or subsequent washout; Sigma-Aldrich, M1404). Nocodazole washout was performed by washing cells five times with prewarmed imaging medium on the microscope. To arrest cells in metaphase for microinjection, cells were incubated with 12 µM proTAME (R&D systems, I-440-01M) for 2 h. To synchronize cells to the G2–M boundary, cells were first synchronized with a double thymidine block followed by a single RO3306[60] block. One day after seeding, cells were transferred into medium containing 2 mM thymidine (Sigma-Aldrich, T1895). Then, 16 h later, cells were washed three times with prewarmed medium and released for 8 h. The thymidine block was repeated once. Then, 6 h after the second thymidine release cells were transferred into medium containing 8 µM RO3306 (Sigma-Aldrich, SML0569) for 4 h. After G2 arrest, RO3306 was removed, and the cells were washed three times with imaging medium containing 1.5 µM okadaic acid (LC laboratories, O-5857). To induce apoptosis as a positive control for apoptotic index measurements, cells were treated with 5 µM anisomycin (Sigma-Aldrich, A9789) for 3 h. To induce DNA double-stranded breaks as a positive control for DNA damage measurements, cells were treated with 50 ng ml[−1] neocazinostatin (Sigma-Aldrich, N9162) for 3 h.

## Immunofluorescence

For all immunofluorescence experiments, cells were grown on sterilized 18 mm round Menzel cover glasses in 24-well plates, except immunofluorescence against acetylated histones, which was performed on Nunc LabTek II chambered cover glass. For co-staining spindle poles and centromeres, cells were fixed and extracted at the same time with 1× PHEM (60 mM K-PIPES (Sigma-Aldrich), pH 6.9, 25 mM K-HEPES (Sigma-Aldrich), pH 7.4, 10 mM EGTA (Merck, 324626), 4 mM MgSO$_4$·7H$_2$O (Sigma-Aldrich)) containing 0.5% Tween-20 (Sigma-Aldrich) and 4% methanol-free formaldehyde (Thermo Fisher Scientific) for 10 min. The fixation reaction was quenched with 10 mM Tris-HCl (Sigma-Aldrich), pH 7.4, in phosphate-buffered saline (PBS), washed again with PBS and blocked in 10% normal goat serum (Abcam), 0.1% Tween-20 (Sigma-Aldrich) for 1 h.

For staining of acetylated histone tails or cyclin B1, cells were fixed in PBS containing 4% methanol-free formaldehyde (Thermo Fisher Scientific, 28906) for 10 min. The fixation was quenched with 10 mM Tris-HCl (Sigma-Aldrich) pH 7.4 in PBS for 5 min, and cells were permeabilized with PBS containing 0.5% Triton X-100 for 15 min, washed again with PBS and blocked using 2% bovine serum albumin (BSA) (Sigma-Aldrich, A7030) in PBS containing 0.2% Tween-20 (Sigma-Aldrich) for 1 h. Antibody dilution buffer was composed of PBS with 2% BSA (Sigma-Aldrich, A7030), containing 0.1% Tween-20 (Sigma-Aldrich).

For staining of γH2A.X, cells were fixed in PBS containing 4% methanol-free formaldehyde (Thermo Fisher Scientific, 28906) for 10 min. The fixation was quenched with 100 mM glycin (Sigma-Aldrich) in PBS for 10 min, and the cells were permeabilized with PBS containing 0.5% Triton X-100 for 15 min, washed again with PBS and blocked using 10% normal goat serum (Abcam), 0.1% Tween-20 (Sigma-Aldrich) in PBS for 1 h. After incubations with primary and secondary antibodies, the samples were washed three times with PBS containing 0.1% Tween-20 (Sigma-Aldrich) for 10 min each time. Cyclin B1 was detected with a monoclonal rabbit antibody (Cell Signaling, 12231S, 7, 1:800) and visualized using a donkey anti-rabbit Alexa Fluor 488 secondary antibody (Molecular Probes, A21206, 1:1,000) (Extended Data Fig. 1d). CENP-A was detected with a monoclonal mouse antibody (Enzo Life Sciences, ADI-KAM-CC006-E, 10161910, 1:1,000) and visualized using a goat anti-mouse Alexa Fluor 488 secondary antibody (Molecular Probes, A11001, 1:1,000) (Extended Data Fig. 1f). Pericentrin was detected with a recombinant rabbit antibody (Abcam, ab220784, GR3284309-1, 1:2,000) and visualized using a goat anti-rabbit Alexa Fluor 633 secondary antibody (Molecular Probes, A21071, 1:1,000) (Extended Data

Fig. 1f). Acetylated histone2B was detected using a polyclonal rabbit antibody (Millipore, 07-373, 3092508, 1:500) and visualized using a donkey anti-rabbit Alexa Fluor 488 secondary antibody (Molecular Probes, A21206, 1:1,000) (Extended Data Figs. 3a–d and 5a). Acetylated histone 3 was detected using a polyclonal rabbit antibody (Merck, 06-599, 3260200, 1:500) and visualized using a donkey anti-rabbit Alexa Fluor 488 secondary antibody (Molecular Probes, A21206, 1:1,000) (Extended Data Fig. 3a–d). Acetylated histone 4K16 was detected using a recombinant rabbit antibody (Abcam, ab109463, GR284778-8, 1:400) and visualized using a donkey anti-rabbit Alexa Fluor 488 secondary antibody (Molecular Probes, A21206, 1:1,000) (Extended Data Figs. 3a–d). γH2A.X was detected with a monoclonal mouse antibody (BioLegend, 613402, B283251, 1:1,000) and visualized using a goat anti-mouse Alexa Fluor 488 secondary antibody (Molecular Probes, A11001, 1:1,000) (Extended Data Fig. 3g). Immunofluorescence samples prepared in Nunc LabTek II chambered cover glass wells were stored in PBS containing 1.62 µM Hoechst 33342 (Invitrogen). Immunofluorescence samples prepared on 18 mm round Menzel cover glasses were embedded in Vectashield with or without DAPI (Vectorlabs, H-1000 or H-1200).

## Preparation of 384-well microscopy plates and coverslips for in vitro assays

µCLEAR microscopy plates (384-well; Greiner Bio-One, 781906) were washed with 5% Hellmanex III (Lactan, 105513203) in ≥18 MΩ MonoQ H$_2$O at 65 °C in a tabletop Incu-Line oven (VWR) for 4 h and then rinsed 10 times with ≥18 MΩ MonoQ H$_2$O. Silica was etched with 1 M KOH (Sigma-Aldrich) for 1 h at room temperature rinsed again 10 times with ≥18 MΩ MonoQ H$_2$O. The etched multiwell plate was treated with 5k-mPEG-silane (Creative PEGworks, PLS-2011) suspended in 95% ethanol (VWR) for 18 h at room temperature. The plate was washed once with 95% ethanol, then 10 times with ≥18 MΩ MonoQ H$_2$O and dried in a clean chemical hood overnight. Until individual wells were used, the plate was sealed with adhesive PCR plate foil (Thermo Fisher Scientific) and kept in a dry and dark space. Before an experiment, the foil above individual wells was cut and 50 µl of 100 mg ml[−1] BSA (Sigma-Aldrich, A7030) was added.

For microtubule/nucleosome array droplet experiments (Fig. 3e), thin layer cover glass sandwiches were constructed from passivated 24 × 60 mm Menzel coverglass (VWR). To clean coverslips, they were vertically stacked into a Coplin jar (Canfortlab, LG084). Coplin jars were then sonicated in acetone (Sigma-Aldrich) for 15 min, in 100% ethanol (VWR) for 15 min and then washed 10 times with ≥18 MΩ MonoQ H$_2$O. All of the sonication steps were performed using an ultrasonic cleaning bath (Branson, 2800 Series Ultrasonic Cleaner, M2800). In a separate Erlenmeyer flask, 31.5 ml 30% aqueous hydrogen peroxide solution (Sigma-Aldrich, 31642) was added to 58.5 ml concentrated sulfuric acid (Sigma-Aldrich, 258105) (piranha acid). The flask was gently swirled until bubbling and heating occurred, then the whole contents of the flask was added to the coverslips in the Coplin jars, ensuring that all coverslip surfaces were covered. The jar was transferred into a 95 °C water bath and heated for 1 h. Afterwards, the piranha solution was discarded, and the cover glasses were washed once with ≥18 MΩ MonoQ H$_2$O and etched with 0.1 M KOH for 5 min. The cover glasses were transferred to a fresh, dry Coplin jar and dried to completion in a 65 °C benchtop oven, and afterwards left to cool to room temperature. In a separate Erlenmeyer flask, 4 ml dichlorodimethylsilane (DCDMS) (Sigma-Aldrich, 440272, anhydrous) was injected into 80 ml heptane (Sigma-Aldrich, 246654, anhydrous). The contents of the flask were immediately transferred to the jar containing the cover glasses, and the jar was incubated for 1 h at room temperature. Next, silane was decanted, and the cover glasses were sonicated in chloroform (Sigma-Aldrich) for 5 min, washed in chloroform once, sonicated in chloroform again for 5 min and sonicated twice in ≥18 MΩ MonoQ H$_2$O. Finally, the cover glasses were washed once more in chloroform, air dried and stored in a sealed container in a dry and dark, dust-free space (for up to 6 months).

To passivate cover glasses, on the day of an experiment, a cover glass was transferred to a drop of 5% Pluronic F-127 (Thermo Fisher Scientific, P6866) dissolved in BRB80 buffer (80 mM K-PIPES, pH 6.9 (Sigma-Aldrich, P6757), 1 mM $MgCl_2$ (Sigma-Aldrich), 1 mM EGTA (Merck, 324626)) for >2 h at room temperature. Directly before assembly of the imaging chamber, the cover glass was rinsed once with ≥18 MΩ MonoQ $H_2O$ and once with BRB80 (80 mM K-PIPES, pH 6.9 (Sigma-Aldrich, P6757), 1 mM $MgCl_2$ (Sigma-Aldrich) and 1 mM EGTA (Merck, 324626)).

### Expression and purification of GFP proteins

pET-based constructs for the expression and purification of meGFP(−7e) and scGFP(+7e) were a gift from D. Liu[43]. An overnight culture of Rosetta 2 (pLysS) *Escherichia coli* (Novagen) transformed with pET_−7GFP or pET_+7GFP plasmids encoding GFP with a theoretical peptide charge of −7e or +7e, respectively, were grown on an agar plate by replating a single transformant on LB supplemented with 100 ng μl$^{-1}$ of ampicillin at 37 °C. The bacterial lawn was suspended in LB supplemented with the 100 ng μl$^{-1}$ of ampicillin and grown at 37 °C to an optical density at 600 nm of 0.4, cooled over 1 h to 18 °C, and recombinant protein expression was induced by addition of IPTG to 0.5 mM for 18 h at 18 °C. The cells were collected by centrifugation, resuspended in NiNTA lysis buffer (50 mM HEPES·NaOH, pH 7, 150 mM NaCl, 10% (w/v) glycerol, 15 mM imidazole, 5 mM β-mercaptoethanol, 1 mM benzamidine, 100 μM leupeptin, 100 μM antipain, 1 μM pepstatin), the cellular suspension was flash frozen in liquid $N_2$ and stored at −80 °C.

Bacterial cultures with expressed GFP proteins in NiNTA lysis buffer were thawed in a water bath and lysed by multiple passages through an Avestin Emulsiflex-C5 high-pressure homogenizer at around 10,000 psi. An equal volume of NiNTA dilution buffer (50 mM HEPES·NaOH, pH 7, 1.85 M NaCl, 10% (w/v) glycerol, 15 mM imidazole, 5 mM β-mercaptoethanol, 1 mM benzamidine, 100 μM leupeptin, 100 μM antipain, 1 μM pepstatin) was added to the lysate to increase NaCl concentration to 1 M. Soluble bacterial lysate was isolated by centrifugation of cellular debris in a Beckman Avanti J-26 XPI centrifuge in a JA25.5 rotor at 20,000 rpm. Soluble lysate was incubated with NiNTA resin (Qiagen) equilibrated in NiNTA wash buffer A (50 mM HEPES·NaOH, pH 7, 1 M NaCl, 10% (w/v) glycerol, 15 mM imidazole, 5 mM β-mercaptoethanol, 1 mM benzamidine, 100 μM leupeptin, 100 μM antipain, 1 μM pepstatin) for 2 h in batch with end-over-end mixing. NiNTA resin was poured into a Bio-Rad EconoColumn and the resin was washed with 20 column volumes of NiNTA wash buffer A and 20 column volumes of NiNTA wash buffer B (50 mM HEPES·NaOH, pH 7, 150 mM NaCl, 10% (w/v) glycerol, 15 mM imidazole, 5 mM β-mercaptoethanol, 1 mM benzamidine, 100 μM leupeptin, 100 μM antipain, 1 μM pepstatin) before elution in NiNTA elution buffer (50 mM HEPES·NaOH, pH 7, 150 mM NaCl, 10% (w/v) glycerol, 350 mM imidazole, 5 mM β-mercaptoethanol, 1 mM benzamidine, 100 μM leupeptin, 100 μM antipain, 1 μM pepstatin).

GFP proteins were diluted with 9 volumes of buffer SA (20 mM HEPES·NaOH, pH 7, 10% (w/v) glycerol, 1 mM DTT) and applied to Source 15S (meGFP) or Source15Q (scGFP) resin (GE Healthcare) equilibrated in 98.5% buffer SA and 1.5% buffer SB (20 mM HEPES·NaOH, pH 7, 1 M NaCl, 10% (w/v) glycerol, 1 mM DTT) and eluted with a linear gradient to 100% buffer SB. Fractions containing GFP proteins were concentrated in a centrifugal concentrator with a 3,000 Da MWCO and purified further by size-exclusion chromatography using a Superdex 200 10/300 GL gel filtration column equilibrated with gel filtration buffer (20 mM Tris·HCl, pH 8, 150 mM NaCl, 10% (w/v) glycerol, 1 mM DTT). Peak fractions of purified meGFP(−7e) and scGFP(+7e) proteins were concentrated in a centrifugal concentrator, as described above, and quantified by measuring protein absorbance at 280 nm using their calculated molar extinction coefficient (https://web.expasy.org/protparam/) of 23,380 M$^{-1}$ cm$^{-1}$ for meGFP and 18,910 M$^{-1}$ cm$^{-1}$ for scGFP. Purified proteins were flash-frozen in liquid $N_2$ and stored at −80 °C in single-use aliquots.

### Nucleosome array in vitro experiments

Generation, fluorophore labelling, assembly and quality control of nucleosome arrays has been described previously[13]. Here 12X601 nucleosome arrays carrying no label or an Alexa Fluor 488 or Alexa Fluor 594 label were used. The covalently fluorescently labelled nucleosome arrays used for microinjection were labelled at an 8% fluorophore density (8 in 100 histone2B proteins labelled with a fluorophore) and dialysed against TE (10 mM Tris-HCl (Sigma-Aldrich), pH 7.4 (Sigma-Aldrich), 1 mM EDTA (Sigma-Aldrich), pH 8.0, 1 mM DTT (Roche, 10708984001)) to remove glycerol.

To induce phase separation of nucleosome arrays (Extended Data Figs. 4e and 5e,g), arrays were first equilibrated in chromatin dilution buffer (25 mM Tris·OAc, pH 7.5 (Sigma-Aldrich), 5 mM DTT (Roche, 10708984001), 0.1 mM EDTA (Sigma-Aldrich), 0.1 mg ml$^{-1}$ BSA (Sigma-Aldrich, A7030), 5% (w/v) glycerol (Applichem, A0970)) in the presence of 2.03 μM Hoechst 33342 (Invitrogen, H1399) when using unlabelled arrays for 10 min at room temperature. Phase separation was induced by addition of 1 volume of phase separation buffer (25 mM Tris·OAc, pH 7.5 (Sigma-Aldrich), 5 mM DTT (Roche, 10708984001), 0.1 mM EDTA (Sigma-Aldrich), 0.1 mg ml$^{-1}$ BSA (Sigma-Aldrich, A7030), 5% (w/v) glycerol (Applichem, A0970), 300 mM potassium acetate (Sigma-Aldrich), 1 mM Mg[OAc]$_2$, 2 μg ml$^{-1}$ glucose oxidase (Sigma-Aldrich, G2133), 350 ng ml$^{-1}$ catalase (Sigma-Aldrich, C1345) and 4 mM glucose (AMRESCO, 0188)). For in vitro assays containing tubulin, EGTA was substituted for EDTA in the dilution and phase separation buffers. After addition of phase-separation buffer, the final concentration of nucleosome arrays per reaction was 500 nM, and the reactions were incubated at room temperature for 10 min before transferring the suspension to the imaging chambers.

To visualize soluble tubulin partitioning relative to chromatin droplets in vitro, 20 μl of chromatin droplet suspension was transferred to a passivated well of a 384-well microscopy plate. Chromatin droplets were allowed to sediment for 15 min. Next, 5 μl of soluble TRITC-labelled tubulin (Cytoskeleton, TL590M) in phase-separation buffer containing 500 ng ml$^{-1}$ nocodazole was added for a final tubulin concentration of 5 μM. Soluble tubulin was equilibrated for 10 min before images were recorded.

To visualize GFP surface charge variant partitioning relative to chromatin droplets in vitro, 20 μl of chromatin droplet suspension was transferred to a passivated well of a 384-well microscopy plate. Chromatin droplets were allowed to sediment for 15 min, after which 5 μl of GFP solution in phase-separation buffer was added to a final concentration of 5 μM. GFP proteins were equilibrated for 10 min before images were recorded.

To visualize chemically modified dextran partitioning into chromatin droplets in vitro, 20 μl of chromatin droplet suspension was transferred to a passivated well of a 384-well microscopy plate. Chromatin droplets were allowed to sediment for 15 min, after which 5 μl dextran solution in phase-separation buffer was added to a final concentration of 500 μg ml$^{-1}$. The dextran used was a fluorescein (FITC)-labelled 4.4 kDa dextran fraction (negative overall charge conferred to dextran by fluorophore charge) (Sigma-Aldrich, FD4) or a FITC-labelled 4.4 kDa dextran fraction modified with diethylaminoethyl (DEAE) groups conferring an overall positive charge (TdB, DD4).

All of the images of macromolecule partitioning relative to chromatin droplets in vitro (Fig. 3e and Extended data Fig. 9m,o) were recorded on the incubated stage of a customized Zeiss LSM980 microscope using a ×40 1.4 NA oil-immersion DIC plan-apochromat objectives (Zeiss).

### Nucleosome array acetylation in vitro

To generate acetylated nucleosome arrays for microinjection, recombinant p300 histone acetyl transferase domain was generated (VBCF Protein technologies Facility) using plasmid pETduet+p300$_{HAT}$ according to a purification strategy described previously[13]. To induce histone

acetylation of nucleosome arrays, arrays with 8% Alexa Fluor 488 or Alexa Fluor 594 label density at a concentration of 3.85 µM were incubated with 6.12 µM p300-HAT (enzyme stock 61.2 µM in gel-filtration buffer (20 mM Tris·HCl, pH 8.0, 150 mM NaCl, 10% (w/v) glycerol, 1 mM DTT)) in the presence of 750 µM Acetyl-CoA (Sigma-Aldrich, A2056) in gel-filtration buffer at room temperature for 2 h with occasional agitation. The acetylation was next stopped by the addition of A-485 (Tocris, 6387) to a final concentration of 9 µM and the reaction mixture was stored in the dark. To 10 µl of quenched acetylation reaction, 10 µl dilution buffer containing 5 µM A-485 (Tocris, 6387) was added and the reaction was allowed to equilibrate at room temperature for 10 min. Next, 1 volume of phase-separation buffer containing 5 µM A-485 (Tocris, 6387) was added and the mixture was incubated for 10 min at room temperature before transferring the suspension to a well of a passivated 384-well microscopy plate.

### Microtubule in vitro polymerization

To generate stabilized microtubule seeds for nucleation of microtubule networks in vitro, 22 µl of 5 mg ml$^{-1}$ tubulin protein (Cytoskeleton, T240) was mixed with 2 µl of 5 mg ml$^{-1}$ TRITC-labelled tubulin protein (Cytoskeleton, TL590M) and 1 µl of 5 mg ml$^{-1}$ Biotin-XX-labelled tubulin (Pursolutions, 033305). All tubulin storage solutions were prepared in BRB80 (80 mM K-PIPES, pH 6.9 (Sigma-Aldrich, P6757), 1 mM MgCl$_2$ (Sigma-Aldrich), 1 mM EGTA (Merck, 324626)). The tubulin mixture was centrifuged at 4 °C for 15 min in a tabletop centrifuge (Eppendorf, 5424R) at 21,000$g$. To 22.5 µl of the resulting supernatant, 2.5 µl 10 mM guanylyl-(alpha,beta)-methylene-diphosphonate (Jena Biosciences, NU-405) was added to a final concentration of 1 mM and the resulting solution was incubated at 37 °C in a water bath in the dark for 30 min. The resulting seeds were sheared using a 22-gauge needle Hamilton syringe (Sigma-Aldrich, 20788). The resulting suspension was stored in the dark. Seeds were prepared freshly each day and used for subsequent experiments.

Soluble tubulin polymerization mix containing 50 µM tubulin protein (Cytoskeleton, T240), 1 µM TRITC-labelled tubulin protein (Cytoskeleton, TL590M) and 1 mM GTP (Sigma-Aldrich, G8877) in BRB80 (80 mM K-PIPES, pH 6.9 (Sigma-Aldrich, P6757), 1 mM MgCl$_2$ (Sigma-Aldrich), 1 mM EGTA (Merck, 324626)) was centrifuged at 4 °C for 15 min in a tabletop centrifuge (Eppendorf, 5424R) at 21,000$g$. The resulting supernatant was used to assemble a microtubule polymerization mix containing 20 µM total soluble tubulin (19.61 µM unlabelled tubulin dimers, 0.39 µM labelled tubulin), 50 mM DTT (Roche, 10708984001), 120 µM glucose (AMRESCO, 0188), 1 mM GTP (Sigma-Aldrich, G8877), 20 µg ml$^{-1}$ glucose oxidase (Sigma-Aldrich, G2133), 175 ng ml$^{-1}$ (Sigma-Aldrich, C1345) in BRB80 (80 mM K-PIPES, pH 6.9 (Sigma-Aldrich, P6757), 1 mM MgCl$_2$ (Sigma-Aldrich), 1 mM EGTA (Merck, 324626)). Composition of soluble tubulin and tubulin polymerization mix was adapted from a procedure for visualization of microtubule plus-end tracking (+TIP) proteins[61].

To visualize chromatin droplets along with polymerized microtubules, imaging chambers were constructed as described previously[62]. Before Pluronic F-127 passivation, each silanized 24 × 60 mm cover glass was cut into a 24 × 25 mm (top piece) and 24 × 35 mm (bottom piece) using a diamond-tipped steel scribe (Miller, DS-60-C). The bottom coverslip was mounted into the support slide and fixed in place with preheated VALAP (1 part Vaseline (Sigma-Aldrich, 16415), 1 part lanolin (Sigma-Aldrich, L7387), 1 part paraffin wax (Sigma-Aldrich, 327204); all parts by weight). In the central region of the bottom cover glass, a two-well silicon culture-insert (Ibidi, 80209) was attached and 40 µl of nucleosome array droplet suspension was added to one of the wells. The suspension was sedimented for 15 min in a humidified chamber at room temperature. Next, 32 µl of buffer was removed from the well, the culture-insert was removed from the coverslip and 20 µl soluble tubulin mix was added to the remaining chromatin droplet suspension. The top coverslip was added on top with the passivated side facing the

reaction mixture, and the droplet was allowed to spread fully (~10 s). The sample was next sealed with VALAP to prevent evaporation and thermal streams within the liquid film. This procedure yielded a liquid film of ~30 µm thickness. The assembled imaging cell was transferred to the incubated stage of a customized Zeiss LSM980 microscope combined with the Airyscan2 detector, using a ×63 1.4 NA oil-immersion DIC plan-apochromat objective (Zeiss). The sample was incubated at 37 °C for >30 min before images were recorded.

### CLEM and electron tomography

HeLa cells stably expressing H2B–mCherry were cultured on carbon-coated Sapphire discs (0.05 mm thick, 3 mm diameter; Wohlwend). To enrich prometaphase cells, cells were synchronized to G2–M using a double thymidine (Sigma-Aldrich) block for 16 h with 2 mM thymidine each and a subsequent RO3306 (Sigma-Aldrich) block for 6 h with 8 µM RO3306. During the RO-3306 block, the cells were treated with 5 µM TSA for 3 h before RO3306 washout. RO3306 was washed out by rinsing the cells three times with prewarmed imaging medium. The cells were observed on a customized Zeiss LSM780 microscope, using a ×20 0.5 NA EC PlnN DICII air objective (Zeiss). Then, 25 min after the RO3306 washout, most of the cells reached prometaphase (based on DIC and chromatin morphology), and the cells were subsequently processed for electron microscopy analysis. Immediately before freezing, the cells were immersed in cryoprotectant solution (imaging medium containing 20% Ficoll PM400; Sigma-Aldrich), and instantly frozen using a high-pressure freezing machine (HPF Compact 01, Wohlwend).

Frozen cells were freeze-substituted into Lowicryl HM20 resin (Polysciences, 15924-1) using freeze-substitution device (Leica EM AFS2, Leica Microsystems) as follows: cells were incubated with 0.1% uranyl acetate (UA) (Serva Electrophoresis, 77870) in acetone at −90 °C for 24 h. The temperature was increased to −45 °C at a rate of 5 °C h$^{-1}$ and the cells were incubated for 5 h at −45 °C. Cells were washed three times in acetone at −45 °C and then incubated in increasing concentrations of resin Lowicryl, HM20 in acetone (10, 25, 50 and 75% for 2 h each) while the temperature was further increased to −25 °C at a rate of 2.5 °C h$^{-1}$. Cells were then incubated in 100% resin Lowicryl, HM20 at −25 °C for a total of 16 h, changing the pure resin solution after 12 h, 14 h and 16 h. The resin was polymerized under ultraviolet light at −25 °C for 48 h. The temperature was increased to 20 °C (5 °C h$^{-1}$) and ultraviolet polymerization was continued for another 48 h.

After resin polymerization, sections of 250 nm thickness were cut with an ultramicrotome (Ultracut UCT; Leica) and collected on copper–palladium slot grids (Science Services) coated with 1% formvar (Plano). The sections were first observed on the Zeiss LSM710 microscope, using a ×40 1.4 plan-apochromat oil-immersion objective (Zeiss). H2B fluorescence was used to choose cells and subcellular ROIs for tomography. The sections were then post-stained with 2% UA in 70% methanol at room temperature for 7 min and lead citrate at room temperature for 5 min. The sections were observed by Tecnai F20 transmission EM (200 kV; FEI). For tomography, a series of tilt images was acquired over a −60° to +60° tilt range with an angular increment of 1° using Serial EM software[63] at a final $xy$ pixel size of 1.14 nm. Tomograms were reconstructed using the R-weighted back projection method implemented in the IMOD software package[64].

### Western blotting

Samples were separated by Novex NuPAGE SDS–PAGE system (Thermo Fisher Scientific) using 4–12% Bis-Tris in MES running buffer according to the manufacturer's guidelines, and transferred to a 0.45 µm nitrocellulose membrane (Bio-Rad) by tank-blot wet transfer (Bio-Rad) at room temperature for 1 h. Blocking, primary antibody incubations (4 °C, 16 h) and secondary antibody incubations (room temperature, 1.5 h) were performed in 5% (w/v) milk (Maresi, Fixmilch instant milk powder) in PBS, containing 0.05% Tween-20 (Sigma-Aldrich). SMC4 was probed

using a rabbit polyclonal antibody (Abcam, ab229213, GR3228108-5, 1:1,000). GAPDH was probed with a rabbit polyclonal antibody (Abcam, ab9485, GR3212164-2, 1:2,500). hKid was probed with a monoclonal rabbit antibody (Abcam, ab75783, GR129278-4, 1:1,000). KIF4A was probed with a recombinant rabbit antibody (Abcam, ab124903, GR96215-7, 1:1000). Horseradish peroxidase (HRP)-conjugated anti-mouse or anti-rabbit secondary antibodies (Bio-Rad, 1:10,000) were visualized using ECL Plus Western Blotting Substrate (Thermo Fisher Scientific) on a Bio-Rad ChemiDoc Imager operated by analysed using Bio-Red Image Lab v.6.0.1, which was also used for analysis.

## Microscopy

Images of G2-to-mitosis inductions, DsRed transfections, HaloTag stainings of wild-type and Smc4-mAID-HaloTag cells, histone acetylation, p300 overexpression and fluorescence recovery after photobleaching (FRAP) experiments were recorded on a customized Zeiss LSM780 microscope, using ×40 or ×63 1.4 NA oil-immersion DIC plan-apochromat objectives (Zeiss), operated by ZEN Black 2011 software. Images of chromatin density, live-microtubule stains, AluI-digestion time-lapse videos, nucleosome array microinjections, tubulin microinjections, CENP-A/pericentrin immunofluorescence experiments and cyclin B1 staining were recorded on a customized Zeiss LSM980 microscope combined with the Airyscan2 detector, using ×40 or ×63 1.4 NA oil-immersion DIC plan-apochromat objectives (Zeiss), operated by ZEN3.3 Blue 2020 software. For all of the confocal microscopes, an incubator chamber (EMBL) provided a humidified atmosphere and a constant 37 °C temperature with 5% $CO_2$.

Images of fields of cells to determine the apoptotic index were recorded on an inverted Axio Observer Z1 with a sCMOS camera equipped with a ×20/0.5 plan-neofluar air objective, controlled by ZEN Blue. For detection of Hoechst 33342 fluorescence, a Ex 377/350 nm, Em 447/60 nm filter was used; for detection of pSIVA fluorescence, a Ex 470/40 nm, Em 525/50 nm filter was used; and for detection of PI fluorescence, a Ex 530/75 nm, Em 560/40 nm was used.

For FRAP experiments, selected image regions were bleached using a laser intensity 200-fold higher than the laser intensity used for image acquisition, and each bleached pixel was illuminated 20 times with the pixel dwell time used for acquisition (1.79 µs). Images were acquired every 25 ms for the course of the experiment.

Images of nucleosome array droplets and in vitro polymerized microtubules were recorded on a customized Zeiss LSM980 microscope combined with the Airyscan2 detector, using the ×40 or ×63 1.4 NA oil-immersion DIC plan-apochromat objectives (Zeiss), operated by ZEN3.3 Blue 2020 software. For mounting 384-well microscopy plates, the Pecon Universal Mounting Frame KM adapter was used. For mounting microtubule imaging slides, a custom aluminium mounting block for 24 mm × 24–60 mm coverslips was used (IMP/IMBA workshop and BioOptics).

## Microinjection experiments

Live-cell microinjection experiments were performed using a FemtoJet 4i microinjector in conjunction with an InjectMan 4 micromanipulation device (Eppendorf). All microinjections were performed using pre-pulled Femtotips injection capillaries (Eppendorf). The microinjection device was directly mounted onto a customized confocal Zeiss LSM780 (Fig. 2c and Extended Data Fig. 8a–h) or a customized Zeiss LSM980 microscope (Fig. 2a,e and Extended Data Figs. 3a, 4a, 7a,c,e,g, 8k,l,o, 9a,c,l,k and 10a,c) with live-cell incubation (37 °C, 5% $CO_2$). For all microinjections, cells were cultured in µ-Dish 35 mm high-wall imaging dishes with a polymer or glass bottom (Ibidi) to reach near-confluency on the day of the injection.

For injection of AluI (Fast Digest, Thermo Fisher Scientific, FD0014), 1.5 µl AluI stock was added to 3 µl of 5 mg ml⁻¹ FITC-labelled 500 kDa dextran fraction (Sigma-Aldrich, FD500S) dissolved in injection buffer (50 mM K-HEPES, pH 7.4 (Sigma-Aldrich, H3375), 5% glycerol (Applichem, A0970), 1 mM Mg(OAc)₂ (Sigma-Aldrich, M5661)). Using an Eppendorf Microcapillary Microloader (Eppendorf, 5242956003), 3 µl diluted enzyme was loaded into a Femtotip microinjection capillary. Microinjection of mitotic cells (Figs. 2a,c,e,g and 4a and Extended Data Figs. 7a,c,e,g, 8g and 10a,c) was performed using injection settings of 130–150 hPa injection pressure, 0.15 to 0.25 s injection time and 30 hPa compensation pressure. Microinjection of G2 interphase cells (Extended Data Fig. 8a–h) was performed using injection settings of 120 hPa injection pressure, 0.4 to 0.5 s injection time and 20 hPa compensation pressure.

For injection of nucleosome arrays (Extended Data Fig. 8k), 1.5 µl injection buffer (50 mM K-HEPES, pH 7.4, 25% glycerol buffer) was added to 1.5 µl unmodified and 1.5 µl acetylated nucleosome array solution (1.3 µM final concentration for each nucleosome in the injection solution). Microinjection of mitotic cells was performed using injection settings of 180–190 hPa injection pressure, 0.35 s injection time and 85 hPa compensation pressure.

For injection of tubulin protein (Fig. 3a), TRITC-labelled tubulin (Cytoskeleton, TL590M) was dissolved in 5% GPEM (80 mM K-PIPES, pH 6.9 (Sigma-Aldrich, P6757), 1 mM MgCl₂ (Sigma-Aldrich, 63065), 1 mM EGTA (Merck, 324626)) supplemented with 1 mM GTP (Sigma-Aldrich, G8877) to a concentration of 0.5 mg ml⁻¹. Protein was clarified by centrifugation at 4 °C for 15 min in a tabletop centrifuge at 21,000g. The supernatant was microinjected into mitotic cells using injection settings of 175 hPa injection pressure, 0.25 s and 85 hPa compensation pressure. For microinjection of cells arrested in prometaphase with nocodazole, the G-PEM was additionally supplemented with 100 ng ml⁻¹ nocodazole (Sigma-Aldrich, M1404).

For microinjection of GFP surface charge variants (Extended Data Fig. 9a), recombinant meGFP(−7e) or scGFP(+7e) were dissolved in injection buffer (50 mM K-HEPES, pH 7.4 (Sigma-Aldrich, H3375), 25% glycerol (Applichem, A0970)) to a concentration of 15 µM. Microinjection of mitotic cells was performed using injection settings of 125 hPa injection pressure, 0.2 s injection time and 40 hPa compensation pressure.

For microinjection of charge-modified dextran fractions (Extended Data Fig. 9c), a FITC-labelled 4.4 kDa dextran fraction (negative overall charge conferred to dextran by fluorophore charge) (Sigma-Aldrich, FD4) or a FITC-labelled 4.4 kDa dextran fraction modified with DEAE groups conferring an overall positive charge (TdB, DD4) were dissolved in injection buffer (50 mM K-HEPES, pH 7.4 (Sigma-Aldrich, H3375), 25% glycerol (Applichem, A0970)) to a concentration of 5 mg ml⁻¹. Microinjection of mitotic cells was performed using injection settings of 125 hPa injection pressure, 0.1–15 s injection time and 20 hPa compensation pressure.

## Image analysis

**DNA congression analysis.** To quantify DNA congression to the spindle equator in live cells, the DNA distribution along a line profile parallel to the pole-to-pole axis was measured (for Fig. 1a,b, 7.06 µm width, 22.5 µm length; for Extended Data Fig. 5c,d, 12.04 µm width, 22.5 µm length). Along each line profile, the accumulated DNA density was measured in the central 5 µm interval around the centre position between the poles (determined by highest SiR–tubulin peak intensities) and divided by the total DNA density along the entire profile, after subtraction of the extracellular background fluorescence. Each line profile represents an average intensity projection of z-slices around the pole-to-pole axis (for Fig. 1a,b, 2.4 µm range; for Extended Data Fig. 5c,d, 0.75 µm range).

**Chromatin/DNA compaction analysis.** To quantify DNA density, in a central z-section of a mitotic cell (determined by visual inspection on the basis of the highest SiR–tubulin staining intensity at the poles (Fig. 1a,c and Extended Data Figs. 4a–d and 5c,e) the DNA channel was denoised using a Gaussian blur filter ($\sigma = 2$) and thresholded using the

Otsu dark method in Fiji. The resulting binary mask was converted into a selection and the DNA mean fluorescence within this ROI was measured. All data points were normalized to the mean of unperturbed control cells.

In STLC-treated cells used for AluI microinjection experiment (Fig. 2a,b,e,f and Extended Data Figs. 7e–h and 8g,h), the DNA density was measured in line profiles. In a single $z$-slice, a line profile (3 pixels wide, 2 μm long) through a chromosome parallel to the imaging plane (before AluI injection), a chromatin droplet (20 min after AluI injection) or the diffuse chromatin mass in TSA treated cells (20 min after AluI injection) was measured. The mean histone 2B fluorescence intensity in a 200 nm interval around the peak value was measured. Values were normalized to the mean of the non-injected control (Fig. 2b,f and Extended Data Figs. 7f and 8h) or the non-injected condensin-degraded control (Extended Data Fig. 7h).

In cells subjected to G2–mitosis induction (Extended Data Fig. 8a–h), the YFP channel was denoised using a Gaussian blur filter ($\sigma = 2$) and thresholded using the Otsu dark method in Fiji. The resulting mask was converted into a ROI and the histone 2B–YFP mean fluorescence recorded within this ROI. The values were normalized to the mean of the G2 measurements.

**DNA displacement analysis after mitotic entry with monopolar spindle configuration.** To quantify DNA displacement away from the spindle pole after mitotic entry in the presence of STLC to induce a monopolar spindle geometry, the DNA/chromatin distribution was measured along a radial line profile having the centre of mass of SiR–tubulin fluorescence as a seeding point. From videos of mitotic entry, the time point corresponding to 20 min after prophase entry was used for measurement (Extended Data Fig. 2a–c). The measurement was performed in a maximum intensity $z$-projection of 2 confocal slices around the centre of mass of SiR–tubulin fluorescence, indicating the centre of the monopolar spindle ($z$-offset of 2.5 μm). A radial line profile was drawn, using the ImageJ Plugin Radial Profile Angle (v.1-14-2014 by P. Carl and K. Miura, based on Radial Profile from P. Baggethun), with an integration angle Θ = 180°. Along the integrated histone 2B fluorescence around the monopole, the fluorescence signal within the inner 30% distance of the pole to the outer edge of chromatin distance (~99% of cumulative histone 2B fluorescence) was divided by the outer 70%. The resulting value represents the relative amount of chromatin detectable in pole-proximal regions.

**Kinetochore displacement analysis.** In $z$-projections of Airyscan or confocal slices (7 Airyscan $z$-sections with a $z$-offset of 0.15 μm for Extended Data Fig. 1f, 2 confocal $z$-sections with a 2.5 μm $z$-offset for Extended Data Fig. 2a–c), the histone 2B channel was denoised using a Gaussian blur filter ($\sigma = 2$) and thresholded using the Otsu dark method in Fiji and converted into a mask. Kinetochores were detected after denoising the CENP-A channel using a Gaussian blur filter ($\sigma = 1$) using the Find Maxima function in Fiji (strict, excluding edge maxima, prominence = 100 (Extended Data Fig. 1g), prominence = 2,000 (Extended Data Fig. 2e)). Of the total number of detected kinetochores, the fraction of kinetochores outside chromatin (>0.5 μm distance to the nearest chromatin mask surface) was quantified manually.

**Electron tomography analysis.** Analysis of electron tomograms were performed in IMOD/3dmod, v.4.11. Annotation of all structures was performed in the Slicer Window view, using running $z$ average intensity projections to increase contrast (microtubule annotation: projection of 10–20 $z$-slices, chromatin annotation: 35–50 $z$-slices). To account for the loss in image sharpness towards the top and bottom of the recorded tomograms, for all of the tomograms, only the centre section of the recorded volume was analysed (tomograms with 120 to 150 slices, no annotation in the top and bottom 20 slices). For microtubule annotation, a zoom factor of 0.8 to 1.0 was used. For chromatin annotation, a zoom factor of 0.35–0.45 was used. Assignment of microtubules was based on ultrastructural morphology, and assignment of chromatin boundaries was performed on the basis of local grain size, considering the transition of a large-grained, coarsely interspersed particle-containing area (cytoplasm with ribosomes; typical coefficient of variation (CV) > 0.4) to a fine-grained, finely interspersed particle containing area (chromatin with nucleosomes; typical CV < 0.3 for control, CV < 0.4 for TSA) as the chromosome surface. The electron density as a determining factor of a chromatin surface was only used as a secondary direction as TSA treatment perturbs the chromatin density/morphology and therefore decreases the annotation accuracy. On average, an annotation landmark was set every 5–10 nm for both microtubules and the chromatin surface. After manual annotation of microtubules and a chromatin surface, a model of a meshed chromatin surface was generated by averaging five consecutive annotated slices to increase surface smoothness. Microtubule segment length was measured within the cytoplasm and chromatin-internal regions and normalized to the total volume of the respective domain.

**Mitotic duration measurements in the absence or presence of TSA.** Fields of asynchronous cells stably expressing histone 2B–mRFP were imaged for up to 4 h. The timing of nuclear envelope breakdown and anaphase onset was determined by visual inspection of chromatin morphology.

**FRAP analysis of chromatin mobility.** To measure FRAP, raw data measurement, background subtraction, data correction and normalization were performed according to ref. [65]. The measurement of FRAP ROI and background fluorescence signal was performed directly in ZEN software. Total chromatin fluorescence per time-lapse movie was measured using the Time Series Analyzer v.3 (J. Balaji).

**FRET analysis of aurora B–FRET biosensor.** To determine the mitotic state of control and AluI-injected cells at the G2 to M transition (Extended data Fig. 8a–h), CFP, FRET and YFP signals of the histone 2B fused Aurora B-FRET biosensor[66] were recorded at the same time. To quantify FRET efficiency, a custom Fiji script was used. In average-intensity projections of 9 $z$-slices (1 μm each), a nuclear mask was generated using the YFP fluorescence, after denoising using a Gaussian blur filter ($\sigma = 5$), by thresholding using the Otsu dark method. For each time point, the background for the FRET and CFP channel was measured in an extracellular ROI (square of around 1 μm × 1 μm). After background subtraction, FRET and CFP signals within the nuclear ROI were denoised using a Gaussian blur filter ($\sigma = 5$) and a FRET/CFP ratio was calculated. The resulting ratiometric image was depicted using the Fire lookup table in Fiji, at an interval of 0 to 1.4 for FRET/CFP.

**Measurement of Ki-67 surface confinement.** To determine the surface confinement of Ki-67 to the surface of mitotic chromatin under various conditions, the distribution of Ki-67 was measured along line profiles across the chromatin–cytoplasm boundary.

In mitotic cells arrested in taxol, chromosomes were identified, which were perpendicular to the imaging section in 3 consecutive slices ($z$-offset between Airyscan slices of 0.15 μm). The centre slice was used for analysis. A line profile of 10 px width was manually drawn through one chromatid in a single Airyscan section, orthogonal to the surface of the chromatid and near-orthogonal to the longer axis of the chromosome. The Ki-67 fluorescence was measured along the line and aligned to the first peak in fluorescence intensity (Fig. 2e,f). For measuring Ki-67 distribution across the chromatin–cytoplasm boundary in cells after AluI-digestion of mitotic chromatin, the line profile was drawn perpendicular to the chromatin droplet surface, otherwise following the same measurement parameters as described above. To quantify the surface confinement of Ki-67, the fluorescence value along the line profile for the highest value of the control (surface) was divided by

the value in the centre of the chromatid for control (inside). The same positions along the line profiles were used to calculate the ratios for TSA-treated and AluI-digested cells. (Fig. 2f,g).

To measure the Ki-67 distribution on mitotic chromatin after degradation of SMC4 or degradation of SMC4 and TSA treatment, a line profile of 10 px width was manually drawn across the chromatin surface in a central (based on visual inspection) Airyscan $z$-section ($z$-offset, 0.15 μm). The Ki-67-fluorescence was measured along these profiles and aligned around the 50% maximum value of the histone 2B fluorescence, indicative of the edge of the chromatin mass (Extended Data Fig. 8e,f).

**Macromolecule partitioning relative to chromatin.** In images of microinjected nucleosome arrays into metaphase cells (Extended Data Fig. 8k), the DNA channel was denoised using a Gaussian blur filter ($\sigma = 2$) and thresholded using the Otsu dark method in Fiji. The chromatin mask was converted into a ROI by using the Analyze Particles function in Fiji. For the two injected nucleosome arrays of different colours, the mean fluorescence intensity was measured in the DNA ROI and divided by the fluorescence intensity in the cytoplasm measured in a circular ROI of ~5 μm diameter, with at least 1 μm distance to the chromosomes.

The TRITC–tubulin partition coefficient (Fig. 3a,b) was measured >10 min after microinjection in nocodazole-arrested cells. At the surface of a chromosome at the edge of the chromosome cluster, mean fluorescence intensity of TRITC was measured along a line profile (5 px width, 1.3 μm length) orthogonal to the chromosome surface. Values 1 μm apart within and outside of chromatin were divided, resulting in the measured equilibrium partition coefficient.

The distribution of differently expressed DsRed fusion proteins (Fig. 3c,d) was measured using a custom Fiji script. A chromatin mask was generated using thresholding (default method, Fiji), after Gaussian blur filtering ($\sigma = 2$) the chromatin channel. The mask was then shrunk by 0.2 μm and the mean fluorescence was measured within the mask. Cytoplasmic fluorescence was measured within a ring (width, 0.5 μm), generated by extending the mask by 1.5 μm. Background mean fluorescence was measured within a small circular region outside the cell and subtracted from chromatin and cytoplasmic values. Partitioning coefficients were obtained by dividing chromatin by cytoplasmic fluorescence values. Measurements were performed on the central $z$-slice.

The distribution of microinjected differently charged GFPs (Extended Data Fig. 9a,b) and differently modified FITC-labelled dextran fractions (Extended Data Fig. 9c,d) was measured analogously along line profiles orthogonal to the metaphase plate at the orthogonal surface of the chromatin mass (5 px width, 2.5 μm length). Values 1 μm apart within and outside of chromatin were divided, resulting in the measured equilibrium partition coefficient.

The distribution of soluble TRITC–tubulin dimers (Fig. 3e,f), differently charged GFPs (Extended Data Fig. 9m,n) and differently modified FITC-labelled dextran fractions (Extended Data Fig. 9o,p) within nucleosome array droplets in vitro was measured by determining the mean fluorescence intensity in the droplet in a central section of the droplet, in a small circular region covering ~25% of the droplet area in the respective section. In a large, rectangular buffer ROI without droplets, the mean fluorescence in the buffer was measured and the partition coefficient was calculated for each field of droplets.

**Chromatin droplet distribution analysis in cells.** Chromatin distribution in pole-peripheral regions was measured at 36 min after nocodazole washout. The spindle-pole position was determined as the $z$-slice with highest tubulin signal (SiR–tubulin (Fig. 4a) or eGFP–α-tubulin (Extended Data Fig. 10a,c)) and the total histone 2B (Fig. 4a and Extended Data Fig. 10a,c) and eGFP–CENPA fluorescence intensity in circular ROIs ($r = 5$ μm) in maximum intensity projections of 5 $z$-slices (1 μm offset between slices) around the central section was subtracted from the total cellular fluorescence in maximum intensity

projections (the cell outline was determined by DIC channel). Each dataset was normalized to the mean of the total fluorescence intensities of all cells.

**Microtubule density distribution in vitro.** After deconvolution in ZEN, Airyscan images of fields of microtubules were denoised using a Gaussian blur filter ($\sigma = 2$). The background was then removed using the subtract background function in Fiji, using a rolling ball radius of 740 px. The image was converted into 8-bit, and thresholded using the Auto Local Threshold (v.1.10.1) plugin in Fiji, with the Phansalkar method and a radius of 100. The resulting binary image was skeletonized using the Skeletonize function in Fiji. The resulting skeleton length was measured. The chromatin channel was denoised using a Gaussian blur filter ($\sigma = 2$) and thresholded using the Otsu method in Fiji. The resulting binary image was transformed into a ROI using the Analyze particles function (size range, 5–infinity). Microtubule skeleton lengths were measured within chromatin ROIs and the surrounding buffer, and the ratio of total segment length in chromatin/buffer was calculated per image.

**Validation of auxin-inducible degradation of SMC4-mAID-HaloTAG in live cells.** In images of fields of live cells stained with Oregon-Green488 HaloTAG ligand, the total fluorescence intensity in average intensity projection was recorded in Fiji. To normalize the values to the cell count, the ratios of HaloTAG fluorescence over SiR–DNA intensity were calculated, and the values were normalized to the mean of the Hela WT cells.

**Acetylated histone fluorescence intensity measurements.** After indirect immunofluorescence analysis of the respective histone-acetyl marks, a mask of the DNA channel was generated by thresholding using the Otsu method in Fiji. Next, the obtained binary image was converted to ROIs using the Analyze particles function (size range, 5–infinity). Mitotic and interphase cells were differentiated manually on the basis of chromatin morphology. Within the obtained ROI per cell, the mean fluorescence intensity of the acetyl mark was recorded, the extracellular background measured in a rectangular ROI was subtracted and a ratio of the histone mark over DNA mean intensity was generated to account for changes in DNA compaction between the different conditions. For comparison of untreated interphase and mitotic cells, obtained ratios were normalized to the mean of untreated interphase cells (Extended Data Fig. 3a,b). For comparison of untreated and TSA-treated mitotic cells, ratios were normalized to the mean of untreated mitotic cells (Extended Data Fig. 3c,d). For comparison of p300-expressing mitotic cells, ratios were normalized to the mean of the mock-plasmid transfected mitotic cells (Extended Data Fig. 5a,b).

**Measurement of cyclin B1 stain.** In interphase mitotic cells, determined by the absence or presence of cell rounding, in the central section of the recorded $z$-stack (7 $z$-slices, 1 μm each, central section manually determined), the cell outline in DIC images was used to determine a ROI per cell. The total fluorescence intensity of the cyclin B1 fluorescence within this ROI was recorded in Fiji. The values were normalized to the mean of the cyclin B1 stain in interphase cells.

**Quantification of the apoptotic index.** In fields of cells, the DNA channel was denoised using a Gaussian blur filter ($\sigma = 2$) and thresholded using the Otsu dark method in Fiji, and one ROI per cell was derived using the Analyze Particle function in Fiji. Within each ROI, the mean fluorescence signal of pSIVA and PI was quantified and any cell with a mean fluorescence signal of greater than 1.2× the median of all untreated control cells was considered to be positive for the respective marker and therefore dead/apoptotic. The fraction of all cells scoring positive for pSIVA and PI stain reflects the apoptotic index (Extended Data Fig. 3e,f).

**Quantification of DNA damage foci by γH2A.X staining.** A confocal stack of 21 slices with a z-offset of 0.5 μm was converted into a maximum intensity projection. The z-projection was denoised using a Gaussian blur filter ($\sigma = 2$), and γH2A.X foci were detected using the Find Maxima tool in Fiji (strict detection, 3,000 prominence) within a segmented chromatin mask obtained by thresholding the DNA channel using the Otsu dark method in Fiji (Extended Data Fig. 3g,h).

**Determination of coefficient of variation of chromatin droplets in vitro.** In entire fields of chromatin droplets or nucleosome array solution (Extended Data Fig. 8c) (both recorded ~3 μm above the cover glass using laser powers adjusted to the unmodified condition), the mean fluorescence intensity and s.d. was measured, and the CV ($CV = \alpha \mu^{-1}$) was calculated.

**Airyscan processing and deconvolution.** Raw images recorded using the Airyscan2 detector using AS-SR or multiplex airyscan modes were processed using ZEN3.3 Blue 2020 software.

**Fiji version details.** The Fiji-integrated distribution/version used for analyses in this study was ImageJ v.1.53c, using Java v.1.8.0_66 (64 bit), with in part custom ImageJ plugins as indicated.

## Statistical analyses and data reporting

No statistical methods were used to predetermine sample size. To test significance, the robust, nonparametric Mann–Whitney $U$-test was used. The statistical test used are indicated and the exact $P$ values are been provided wherever possible. In some cases, the $P$ values calculated were below the precision limit of the data type and algorithm used. In those cases, this has been noted in the legend and an upper bound was given. NS, $P > 0.05$; *$P \le 0.05$, **$P \le 0.01$, ***$P \le 0.001$, ****$P \le 0.0001$.

## Statistics and reproducibility

Figure 1a,b: representative examples and quantification of chromatin fraction at spindle centre: control ($n = 51$ cells), ΔCondensin ($n = 65$ cells), ΔCondensin + TSA ($n = 34$ cells), TSA ($n = 61$). Figure 1c: quantification of DNA density: control ($n = 31$ cells), ΔCondensin ($n = 89$ cells), ΔCondensin + TSA ($n = 99$ cells), TSA ($n = 74$). Figure 1d–f: representative examples and quantification of microtubule densities in chromatin (Fig. 1e) and cytoplasm (Fig. 1f): control ($n = 10$ tomograms from 7 different cells), TSA ($n = 10$ tomograms from 8 different cells). Figure 2a,b: representative example and quantification of 11 cells, 3 ROIs per cell. Figure 2c,d: representative examples and quantification of 8 cells (undigested) and 10 cells (AluI-digested). Figure 2e,f: representative example and quantification of 11 cells, 3 ROIs per cell. Figure 2g–j: line scans and ratios measured for control ($n = 19$ cells), TSA ($n = 24$ cells) and AluI ($n = 22$ cells). Figure 3a: representative example of an untreated metaphase cell ($n = 27$). Figure 3a,b: representative examples and quantification of nocodazole treated cells: control ($n = 46$), TSA ($n = 31$). Figure 3c,d: representative examples and quantification of DsRed ($n = 26$), DsRed(−7e) ($n = 26$), DsRed(+9e) ($n = 26$). Figure 3e,f: representative examples and quantification of: nocodazole ($n = 94$ droplets from 45 fields), polymerized microtubules ($n = 13$ fields of droplets). Figure 4a,b: representative example and quantification of $n = 15$ cells.

Extended Data Figure 1a,b: representative examples and quantification of HeLa WT ($n = 25$), HeLa Smc4-mAID-Halo ($n = 25$), HeLa Smc4-mAID-Halo + 5-PhIAA ($n = 25$). Extended Data Figure 1d,e: representative examples and quantification of interphase ($n = 60$), mitotic ($n = 89$), ΔCondensin ($n = 59$), ΔCondensin + TSA ($n = 83$) and TSA ($n = 67$). Extended Data Figure 1f,g: representative examples and quantification of control ($n = 40$), ΔCondensin ($n = 59$), ΔCondensin + TSA ($n = 35$) and TSA ($n = 40$). Extended Data Figure 2a–e: representative examples and quantification of control ($n = 42$), ΔCondensin ($n = 38$) and ΔCondensin + TSA ($n = 60$) cells (Extended Data Fig. 2d) and control ($n = 42$), ΔCondensin ($n = 42$) and ΔCondensin + TSA ($n = 62$) cells (Extended Data Fig. 2e). Extended Data Figure 3a,b: representative examples and quantification of interphase (H2B-Ac ($n = 20$), H3-Ac ($n = 20$), H4K16-Ac ($n = 20$)) and mitotic (H2B-Ac ($n = 20$), H3-Ac ($n = 20$), H4K16-Ac ($n = 20$)) cells. Extended Data Figure 3c,d: representative examples and quantification of control (Extended Data Fig. 2b) and TSA-treated (H2B-Ac ($n = 20$), H3-Ac ($n = 20$), H4K16-Ac ($n = 20$)). Extended Data Figure 3e,f: representative fields of cells and quantification of untreated (13 fields with $n = 4,574$ cells), 500 nM TSA (13 fields with $n = 4,926$ cells), 5 μM TSA (13 fields with $n = 4,653$) and anisomycin (13 fields with $n = 4,188$ cells). Extended Data Figure 3g,h: representative examples and quantification of control ($n = 48$), TSA ($n = 48$) and NCS ($n = 32$) cells. Extended Data Figure 4a,b: representative examples and quantification of control ($n = 39$), 500 nM TSA/washout ($n = 35$), 5 nM TSA ($n = 40$) and 500 nM TSA ($n = 40$) cells. Extended Data Figure 4c,d: representative examples and quantification of ΔCondensin control ($n = 45$), 500 nM TSA/washout ($n = 47$), 5 nM TSA ($n = 40$) and 500 nM TSA ($n = 40$) cells. Extended Data Figure 5a,b: representative examples and quantification of mock ($n = 20$), p300$_{HAT}$ ($n = 26$), p300(D1399Y) ($n = 20$) transfected cells. Extended Data Figure 5c–e: representative examples and quantification of chromatin fraction at spindle centre (Extended Data Fig. 5d) and chromatin density (Extended Data Fig. 5e) of mock ($n = 20$), ΔCondensin + p300$_{HAT}$ ($n = 20$), ΔCondensin + p300(D1399Y) ($n = 24$) transfected cells. Extended Data Figure 6a–c: more examples of electron tomograms in the absence or presence of TSA; control prometaphase ($n = 3$), control metaphase ($n = 7$), TSA prometaphase ($n = 5$) and TSA metaphase ($n = 5$) examples drawn from 10 tomograms per condition from 7 different cells each. Extended Data Figure 6e: quantification of mitotic duration in control ($n = 44$) and TSA-treated ($n = 36$) cells. Extended Data Figure 6f: 3 representative examples of control anaphase cells (top row) and 3 representative examples of TSA treated anaphase cells (bottom row). Extended Data Figure 6g,h: quantification of control ($n = 64$) and TSA-treated ($n = 110$) cells from 3 biological replicates. Extended Data Figure 7a,b: representative examples and quantification of control ($n = 77$), AluI-injected ($n = 41$) and anisomycin-treated ($n = 83$) cells. Extended Data Figure 7c,d: representative examples and quantification of control ($n = 75$) and AluI-injected ($n = 60$) cells. Extended Data Figure 7e,f: representative example of a p300 expressing cell during AluI digestion and quantification of $n = 12$ cells, 3 ROIs for each cell. Extended Data Figure 7g,h: representative example of a condensin-depleted cell during AluI digestion and quantification of $n = 7$ cells, 3 ROIs each. Extended Data Figure 8a,b: representative example and quantification of 13 cells. Extended Data Figure 8c,d: representative example and quantification of 8 cells. Extended Data Figure 8e,f: representative example and quantification of $n = 11$ cells. Extended Data Figure 8g,h: representative example and quantification of 10 cells. Extended Data Figure 8i,j: representation examples and quantification of fields of unmodified (AF488, $n = 26$; AF594, $n = 25$) and acetylated (AF488, $n = 25$; AF594, $n = 30$) nucleosome arrays. Extended Data Figure 8k,l: representative example and quantification of 28 cells. Extended Data Figure 8m,n: representative example and quantification of control ($n = 5$), ΔCondensin ($n = 7$) and ΔCondensin + TSA ($n = 8$) cells, 2–3 ROIs per cell. Extended Data Figure 8o,p: representative example of a *MKI67*-KO cell during AluI digestion and quantification of $n = 10$ cells, 3 ROIs each. Extended Data Figure 9a,b: representative examples and quantification of micro-injected meGFP(−7) ($n = 17$) and scGFP(+7) ($n = 20$). Extended Data Figure 9c,d: representative examples and quantification of microinjected FITC–dextran (−) ($n = 21$) and FITC–DEAE-dextran (+) ($n = 10$). Extended Data Figure 9e,f: representative examples of metaphase cells ($n = 4$) and quantification of nocodazole-treated cells for control ($n = 13$) and TSA ($n = 8$). Extended Data Figure 9g,h: representative examples of metaphase cells ($n = 4$) and quantification of nocodazole-treated cells for control ($n = 33$) and TSA ($n = 31$). Extended Data Figure 9i,j: representative examples of 20 kDa dextran partitioning in control

($n = 33$) and TSA treated ($n = 33$) cells. Extended Data Figure 9k,l: representative examples of 70 kDa dextran partitioning in control ($n = 22$) and TSA-treated ($n = 33$) cells. Extended Data Figure 9m,n: representative examples and quantification of nucleosome array droplets with meGFP(−7) ($n = 69$) and scGFP(+7) ($n = 73$). Extended Data Figure 9o,p: representative examples and quantification of nucleosome array droplets with FITC–dextran (−) ($n = 69$) and FITC–DEAE-dextran (+) ($n = 57$). Extended Data Figure 10a,b: representative example and quantification of $n = 13$ cells. Extended Data Figure 10c,d: representative example and quantification of $n = 16$ cells. All of the experiments in this study were performed in at least two biological replicates.

## Reporting summary
Further information on research design is available in the Nature Research Reporting Summary linked to this paper.

## Data availability
Raw microscopy data are available and will be provided by the corresponding authors on request, given the large file sizes that are involved. No restrictions apply to the availability of the microscopy data. Source data are provided with this paper.

## Code availability
All code used in this study is available at GitHub (https://github.com/gerlichlab/schneider_et_al_Nature_2022).

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

**Acknowledgements** We thank the staff at the IMBA/IMP/GMI BioOptics and Molecular Biology Service and the VBCF Electron Microscopy and Protein Technologies facilities for technical support; I. Patten, A. Khodjakov, H. Maiato and C. Janke for comments on the manuscript; A. M. Rodrigues Viana for advice and reagents for genome engineering. Research in the laboratory of D.W.G. is supported by the Austrian Academy of Sciences, the Austrian Science Fund (FWF; Doktoratskolleg 'Chromosome Dynamics' DK W1238), the Vienna Science and Technology Fund (WWTF; projects LS17-003 and LS19-001), and the European Research Council (ERC) under the European Union's Horizon 2020 research and innovation programme (grant agreement no. 101019039). Research in the laboratory of S.O. is supported by the Vienna Science and Technology Fund (WWTF; project LS19-001). Research in the laboratory of M.K.R. is supported by the Howard Hughes Medical Institute, a Paul G. Allen Frontiers Distinguished Investigator Award (to M.K.R.) and grants from the NIH (F32GM129925 to B.A.G.) and the Welch Foundation (I-1544 to M.K.R.). M.W.G.S. and M.P. have received a PhD fellowship from the Boehringer Ingelheim Fonds. T.N. has received a fellowship from the VIP2 postdoc program.

**Author contributions** D.W.G. and M.W.G.S. conceived the project, with input from B.A.G. and M.K.R. M.W.G.S. designed, performed and analysed all of the experiments, except for those shown in Figs. 1d–f, Extended Data Fig. 6a–d (M.W.G.S. together with S.O.), Extended Data Figs. 3e,f, 7e–h, 8o,p and 9a–d,i–p (M.W.G.S. together with C.B.), Extended Data Figs. 1a,b, 3a–d and 6e (M.F.D.S.), Extended Data Fig. 5a–e (M.W.G.S. with M.F.D.S.) and Fig. 3c,d (M.P.). B.A.G. and L.K.D. generated nucleosome arrays and developed in vitro chromatin condensate assays. C.C.H.L. developed image analysis procedures. P.B. and C.B. performed genome editing to generate AID2 conditional auxin-inducible degradation cell lines expressing the OsTIR1(F74G) ligase. T.N. performed lentiviral transductions to generate transgenic cells expressing H2B–mCherry and CENP-A–eGFP in the Smc4-mAID-HaloTAG background. D.W.G and M.K.R. acquired funding and supervised the project. D.W.G. and M.W.G.S. wrote the manuscript, with input from all of the authors.

**Competing interests** M.K.R. is a co-founder of Faze Medicines. The other authors declare no competing interests.

**Additional information**
**Correspondence and requests for materials** should be addressed to Maximilian W. G. Schneider or Daniel W. Gerlich.

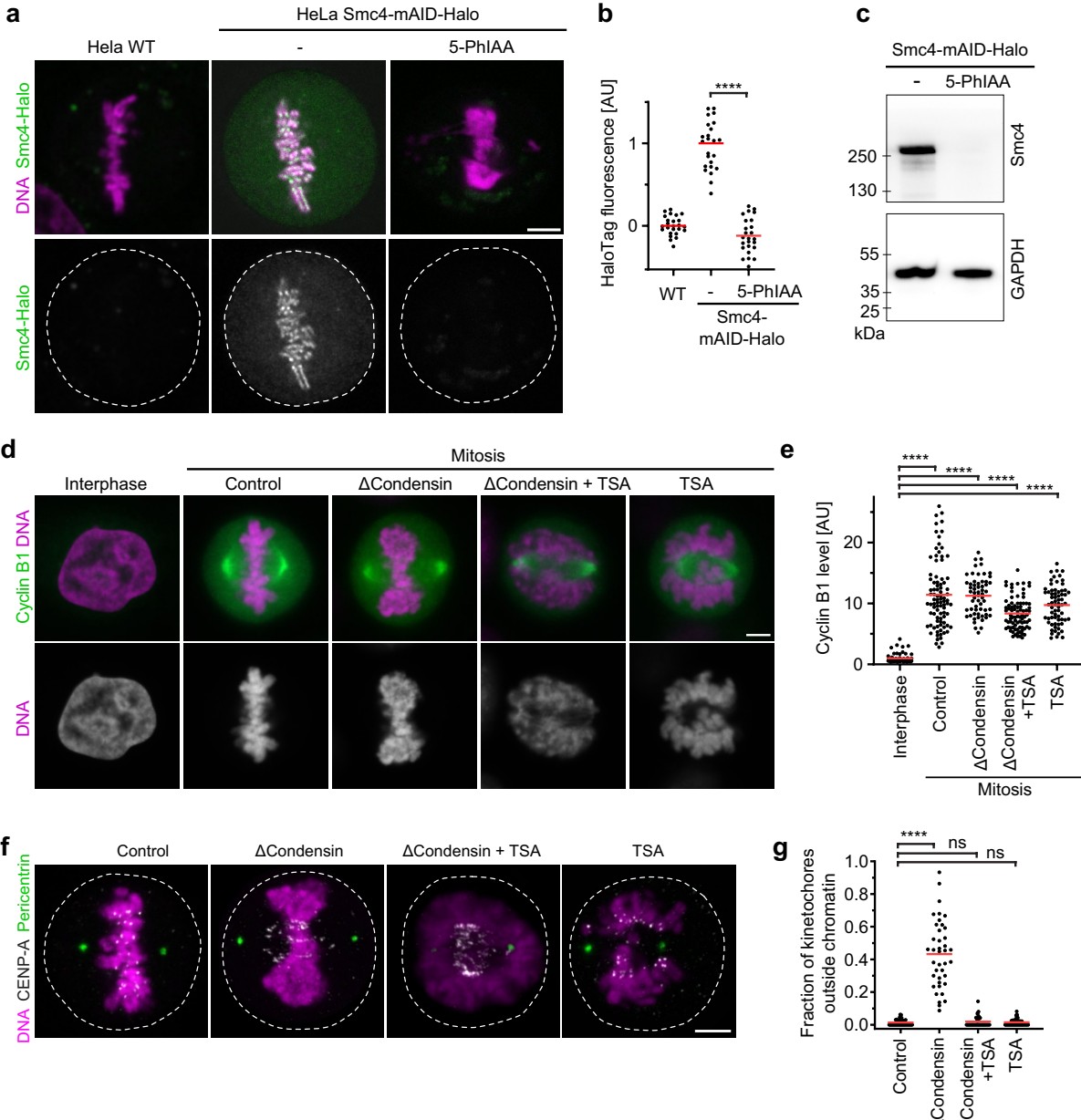

**Extended Data Fig. 1 | Characterization of HeLa Smc4-mAID-HaloTAG cells and analysis of mitotic phenotypes. a-c**, Smc4 expression analysis. **a**, HeLa cells with homozygous Smc4-mAID-Halo alleles and stably expressing OsTIR(F74G) were analysed with or without treatment of 5-Phenylindole-3-eacetic acid (5-PhIAA) for 3 h and stained with OregonGreen-488 HaloTAG ligand; HeLa wild-type (WT) cells serve as control to measure fluorescence background. DNA was stained with Hoechst 33342. Representative examples of HeLa WT (*n* = 25), HeLa Smc4-mAID-Halo (*n* = 25), HeLa Smc4-mAID-Halo + 5-PhIAA (*n* = 25). **b**, Quantification of HaloTAG fluorescence in live mitotic cells as shown in **a**. *n* = 25 for WT, *n* = 25 for Smc4-mAID-Halo without 5-PhIAA, *n* = 25 for Smc4-mAID-Halo with 5-PhIAA. Bars indicate mean; significance was tested by a two-tailed Mann-Whitney test (Smc4-mAID-Halo with 5-PhIAA, *P* = 1.4$^{-14}$). **c**, Immunoblot analysis of Smc4-mAID-Halo cells with or without 3 h 5-PhIAA treatment. Representative examples of *n* = 2 experiments. For gel source data, see Supplementary Figure 1a. **d**, **e**, Analysis of cell cycle state by immunofluorescence staining against Cyclin B1. HeLa cells with homozygously mAID-tagged SMC4 were treated with 5-PhIAA to deplete condensin (ΔCondensin) or with TSA to suppress mitotic histone deacetylation as

indicated. DNA was stained with Hoechst 33342. Classification of interphase and mitotic cell is based on overall cell shape and spindle morphology. **e**, Quantification of Cyclin B1 fluorescence in cells as in **d**. Data normalized to the mean of untreated interphase cells. *n* = 60 (interphase), *n* = 89 (mitotic), *n* = 59 (ΔCondensin), *n* = 83 (ΔCondensin+TSA), *n* = 67 (TSA) cells. Bars indicate mean; significance was tested by a two-tailed Mann-Whitney test (CTRL, *P*<10$^{-15}$; ΔCondensin, *P*<10$^{-15}$; ΔCondensin+TSA, *P*<10$^{-15}$; TSA, *P*<10$^{-15}$, precision limit of floating-point arithmetic). **f**, **g**, Immunofluorescence analysis of kinetochores and spindle poles in mitotic Smc4-mAID-Halo cells after 3 h degradation of Smc4 (ΔCondensin) and/or 3 h treatment with TSA as indicated. Projection of 7 Z-sections with Z-offset of 0.15 μm. **g**, Quantification of fraction of kinetochores outside chromatin regions (>0.5 μm distance from chromatin surface) of cells as in **f**. *n* = 40 cells for control *n* = 59 for ΔCondensin, *n* = 35 for ΔCondensin+TSA, *n* = 40 for TSA. Bars indicate mean, significance was tested by a two-tailed Mann-Whitney test (ΔCondensin, *P*<10$^{-15}$, precision limit of floating-point arithmetic; ΔCondensin+TSA, *P* = 0.852; TSA, *P* = 0.911). Biological replicates: *n* = 2 (**a**–**e**); *n* = 3 (**f**, **g**). Scale bars, 5 μm.

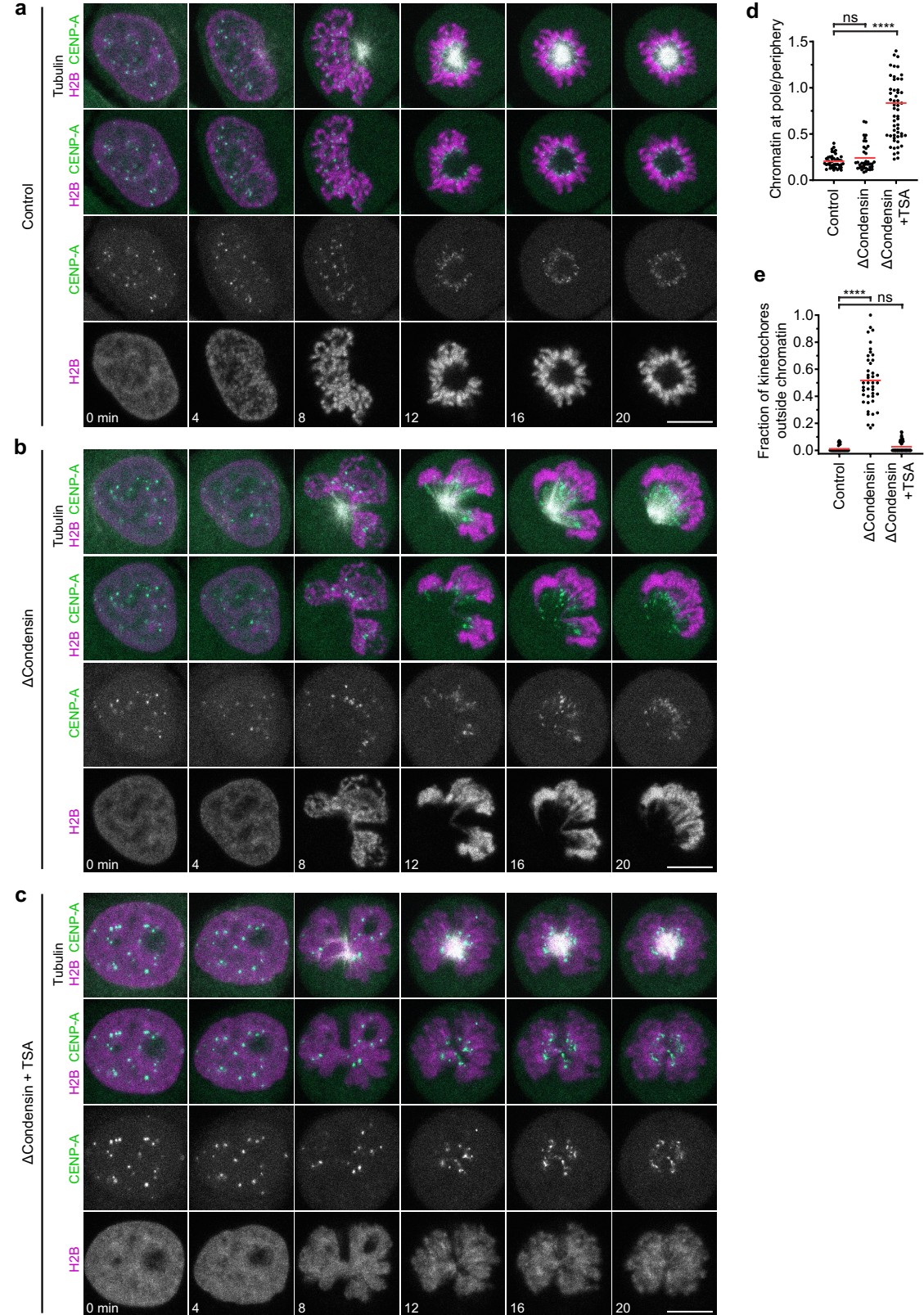

**Extended Data Fig. 2** | See next page for caption.

**Extended Data Fig. 2 | Chromosome and spindle organization in condensin-depleted and TSA-treated cells entering mitosis. a–c**, 3D confocal time-lapse microscopy of live HeLa cells entering mitosis in the presence of STLC to induce monopolar spindle geometry. Cells have homozygous Smc4-mAID-Halo alleles and stably express OsTIR(F74G), H2B-mCherry, and CENP-A-meGFP and are stained with SiR-tubulin. Images show projection of 2 Z-sections with Z-offset of 2.5 μm, centred around the spindle pole. **a**, Mitotic entry of a control cell. **b**, Cell entering mitosis after treatment with 5-PhIAA to deplete Smc4 (ΔCondensin). **c**, Cell entering mitosis after treatment with 5-PhIAA to deplete Smc4 and TSA to suppress mitotic histone deacetylation (ΔCondensin+TSA). **d**, Quantification of chromatin distribution relative to spindle monopole of cells in a-c 20 min after prophase onset. Total H2B-mCherry fluorescence in pole-proximal region divided by total H2B-mCherry-fluorescence of pole-distal regions. $n = 42$ cells for control, $n = 38$ for ΔCondensin, $n = 60$ for ΔCondensin+TSA. Bars indicate mean, significance was tested by a two-tailed Mann-Whitney test (ΔCondensin, $P = 0.988$; ΔCondensin+TSA, $P < 10^{-15}$, precision limit of floating-point arithmetic). **e**, Quantification of fraction of kinetochores outside chromatin (>0.5 μm distance from chromatin surface) of cells as in **a–c** 20 min after prophase onset. $n = 42$ cells for control, $n = 42$ for ΔCondensin, $n = 62$ for ΔCondensin+TSA. Bars indicate mean, significance was tested by a two-tailed Mann-Whitney test (ΔCondensin, $P < 10^{-15}$, precision limit of floating-point arithmetic; ΔCondensin+TSA, $P = 0.061$). Biological replicates: $n = 2$ (**a**,**c**–**e**); $n = 3$ (**b**). Scale bars, 10 μm.

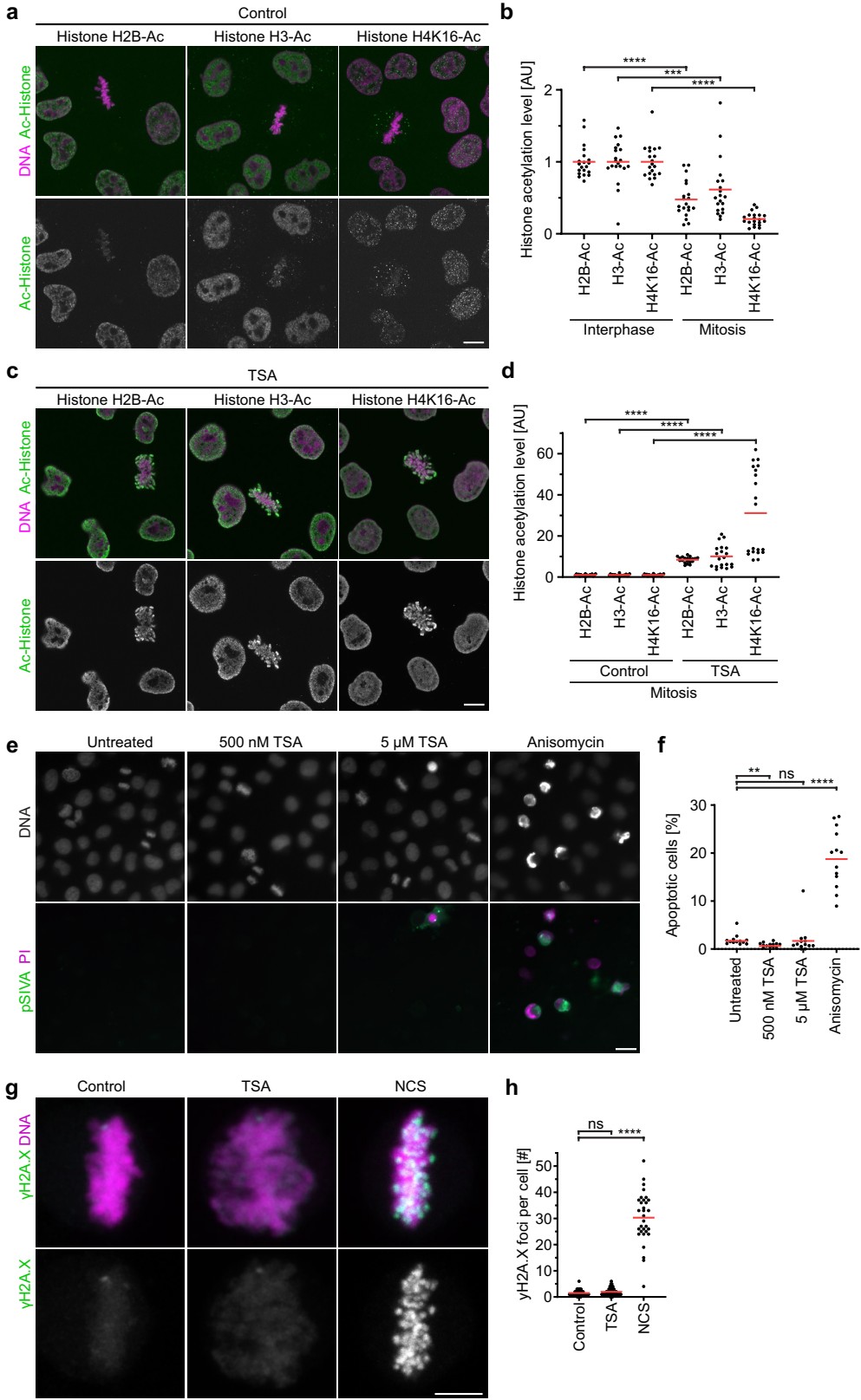

**Extended Data Fig. 3** | See next page for caption.

**Extended Data Fig. 3 | Histone acetylation during cell cycle progression and effect of TSA. a,b,** Immunofluorescence analysis of histone acetylation in interphase and mitosis. **a,** HeLa cells were fixed and stained with antibodies against different acetylated histones as indicated. **b,** Quantification of histone acetylation by immunofluorescence as in **a**, by the ratio of antibody fluorescence to DNA reference staining by Hoechst 33342. For each acetylated histone, all data points were normalized to the mean of interphase cells. $n = 20$ for H2B-Ac, Interphase, $n = 20$ for H3-Ac, Interphase, $n = 20$ for H4K16-Ac, Interphase, $n = 20$ for H2B-Ac, Mitosis, $n = 20$ for H3-Ac, Mitosis, $n = 20$ for H4K6-Ac, Mitosis. Bars indicate mean; significance was tested by a two-tailed Mann–Whitney test (H2B-Ac, $P = 2.117 \times 10^{-7}$; H3-Ac, Mitosis, $P = 4.72 \times 10^{-4}$; H4K16-Ac, $P = 1.451 \times 10^{-11}$). **c,d,** Histone acetylation in mitotic cells after TSA treatment. **c,** HeLa cells were treated with TSA for 3 h, fixed, and histone acetylation analysed by immunofluorescence as in **a**. **d,** Quantification of histone acetylation as in b for mitotic cells 3 h after TSA treatment. For each antibody, all data points were normalized to the mean of control metaphase cells. $n = 20$ for H2B-Ac, control, $n = 20$ for H3-Ac, control, $n = 20$ for H4K16-Ac, control, $n = 20$ for H2B-Ac, TSA, $n = 20$ for H3-Ac, TSA, $n = 20$ for H4K16-Ac, TSA. Bars indicate mean; significance was tested by a two-tailed Mann–Whitney test (H2B-Ac, $P = 1.451 \times 10\text{-}11$; H3-Ac, $P = 1.451 \times 10^{-11}$; H4K16-Ac, TSA, $P = 1.451 \times 10^{-11}$). **e,f,** Analysis of apoptosis after TSA treatment. **e,** Fields of cells stained with Hoechst 33342, pSIVA, and PI to detect apoptotic cells were untreated (control) or treated for 3 h with 500 nM or 5 µM TSA, or 5 µM anisomycin as positive control. **f,** Quantification of apoptotic index after drug treatment of asynchronous cells, each dot representing a field of cells shown in **e**, with $n = 4574$ for untreated, $n = 4926$ for 500 nM TSA, $n = 4653$ for 5 µM TSA, and $n = 4188$ anisomycin cells in total. Bars indicate mean, significance was tested by a two-tailed Mann–Whitney test (500 nM TSA, $P = 2.971 \times 10\text{-}4$; 5 µM TSA, $P = 0.078$; anisomycin, $P = 1.923 \times 10^{-7}$). **g,h,** Immunofluorescence analysis of yH2A.X DNA damage foci in mitosis after treatment with 5 µM TSA or the DNA-damaging agent neocarzinostatin (NCS) as positive control. **g,** Mitotic cells were stained for yH2A.X foci and DNA by Hoechst 33342 after indicated treatments. Z-projection of 11 Z-sections with Z-offset of 0.5 µm. **h,** Quantification of DNA-damage foci in cells as shown in **g**. $n = 48$ cells for control, $n = 48$ for TSA, $n = 32$ for NCS. Bars indicate mean, significance was tested by a two-tailed Mann–Whitney test (TSA, $P = 0.103$; NCS, $P < 10^{-15}$, precision limit of floating-point arithmetic). Biological replicates: $n = 2$ (**a**–**d**,**g**,**h**); $n = 3$ (**e**,**f**). Scale bars, **a**,**c**,**e**, 10 µm, **g**, 5 µm.

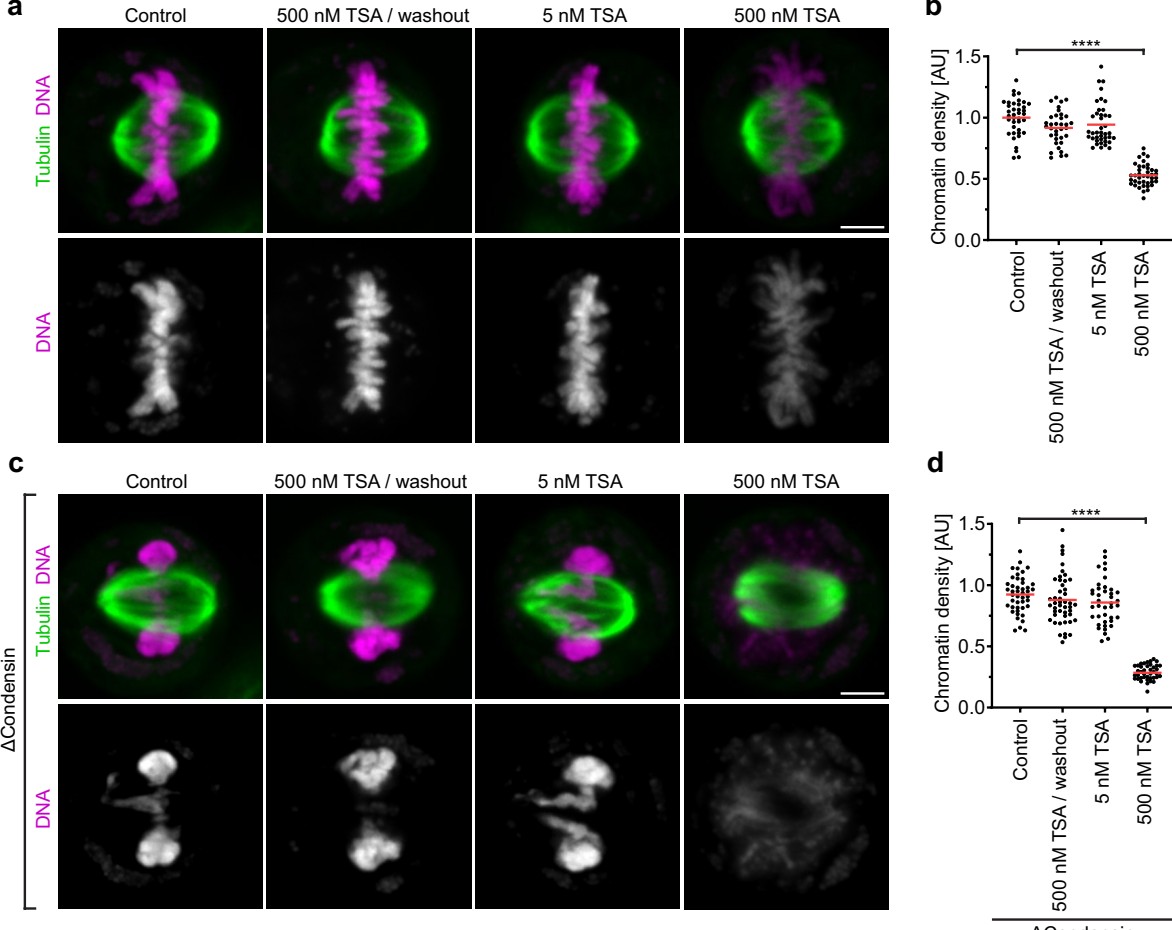

**Extended Data Fig. 4 | Effect of various TSA treatments on mitotic chromosome organization. a**,**b**, Mitotic chromosome morphology and DNA density after various TSA treatment conditions. **a**, Live HeLa cells stained with Hoechst 33342 and SiR-tubulin were analysed without perturbations (control), after 3 h treatment with 500 nM TSA followed by 8 h removal of TSA (500 nM TSA/washout), or after 3 h treatment with 5 nM or 500 nM TSA, as indicated. Mitotic cells were identified based on their rounded morphology and the presence of a bipolar spindle. Z-projection of 4 Z-sections with Z-offset of 0.25 µm. **b**, Quantification of DNA density in mitotic chromatin for cells as shown in a. Data normalized to mean of control mitotic cells. $n$ = 39 cells for control, $n$ = 35 for TSA washout, $n$ = 40 for 5 nM TSA and $n$ = 40 for 500 nM TSA. Bars indicate mean, significance was tested by a two-tailed Mann-Whitney test ($P < 10^{-15}$, precision limit of floating-point arithmetic). **c**,**d**, Mitotic chromosome morphology and DNA density in condensin-depleted cells, in various TSA

treatment conditions. HeLa cells with homozygous Smc4-mAID-Halo alleles and stably expressing OsTIR(F74G) were treated with 5-PhIAA for 3 h to degrade Smc4 (ΔCondensin). **c**, Live cells stained with Hoechst 33342 and SiR-tubulin were analysed without additional perturbations (control), after 3 h treatment with 500 nM TSA followed by 8 h removal of TSA (500 nM TSA/washout), or after 3 h treatment with 5 nM or 500 nM TSA, as indicated. Mitotic cells were identified based on their rounded morphology and the presence of a bipolar spindle. Z-projection of 4 confocal slices with Z-offset of 0.25 µm. **d**, Quantification of DNA density in mitotic chromatin for cells as shown in c. Data normalized to mean of control mitotic cells. $n$ = 45 cells for control, $n$ = 47 for TSA washout, $n$ = 40 for 5 nM TSA and $n$ = 40 for 500 nM TSA. Bars indicate mean, significance was tested by a two-tailed Mann-Whitney test ($P < 10^{-15}$, precision limit of floating-point arithmetic). Biological replicates: $n$ = 2 (**a**–**d**). Scale bars, 5 µm.

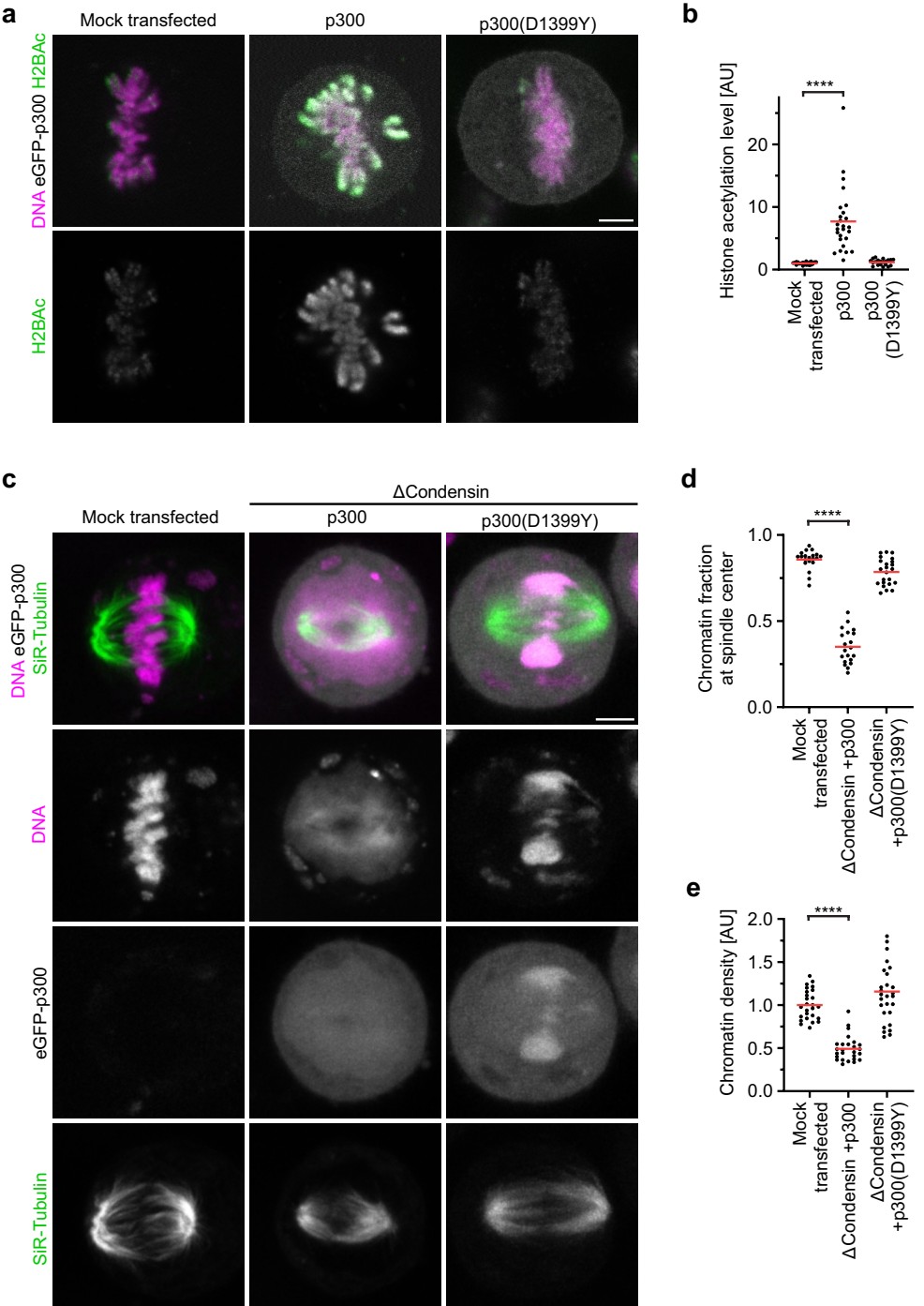

**Extended Data Fig. 5 | Histone acetylation and chromosome organization in cells overexpressing eGFP-p300. a, b,** Immunofluorescence analysis of histone acetylation in mitotic cells overexpressing p300 histone acetyltransferase or catalytically dead p300(D1399Y). **a,** Cells were transfected with a plasmid coding for eGFP-p300 or eGFP-p300(D1399Y) as indicated and fixed after 48 h. Histone 2B acetylation was analysed by immunofluorescence. DNA was stained with Hoechst 33342. **b,** Quantification of histone acetylation in metaphase cells as in **a.** Data points were normalized to the mean of mock-transfected mitotic cells. $n = 20$ for mock-transfected, $n=26$ for p300, $n = 20$ for p300(D1399Y). Bars indicate mean; significance was tested by a two-tailed Mann-Whitney test ($P = 3.57 \times 10^{-13}$). **c–e,** Analysis of chromatin density and chromosome congression to the spindle centre in cells after SMC4-AID-Halo degradation (ΔCondensin) and overexpression of p300 or

catalytically dead p300(D1399Y). **c,** Cells were transfected with a plasmid coding for eGFP-p300 or eGFP-p300(D1399Y) as indicated and analysed by live-cell imaging after 48 h. DNA was stained with Hoechst 33342 and microtubules stained by SiR-Tubulin to identify mitotic cells with bipolar spindles. Projection of 5 Z-sections. **d,** Quantification of chromosome congression by the fraction of chromatin localizing to the central spindle region. $n = 20$ for mock-transfected, $n = 24$ for ΔCondensin+p300, $n=20$ for ΔCondensin+p300(D1399Y). Bars indicate mean; significance was tested by a two-tailed Mann-Whitney test ($P = 1.451 \times 10^{-11}$). **e,** Quantification of chromatin density in cells treated as in **c.** $n = 20$ for mock-transfected, $n = 24$ for ΔCondensin+p300, $n = 20$ for ΔCondensin+p300(D1399Y). Bars indicate mean; significance was tested by a two-tailed Mann-Whitney test ($P = 7.86 \times 10^{-13}$). Biological replicates: $n = 2$ (**a,b**); $n = 3$ (**c–e**). Scale bars, 5 µm.

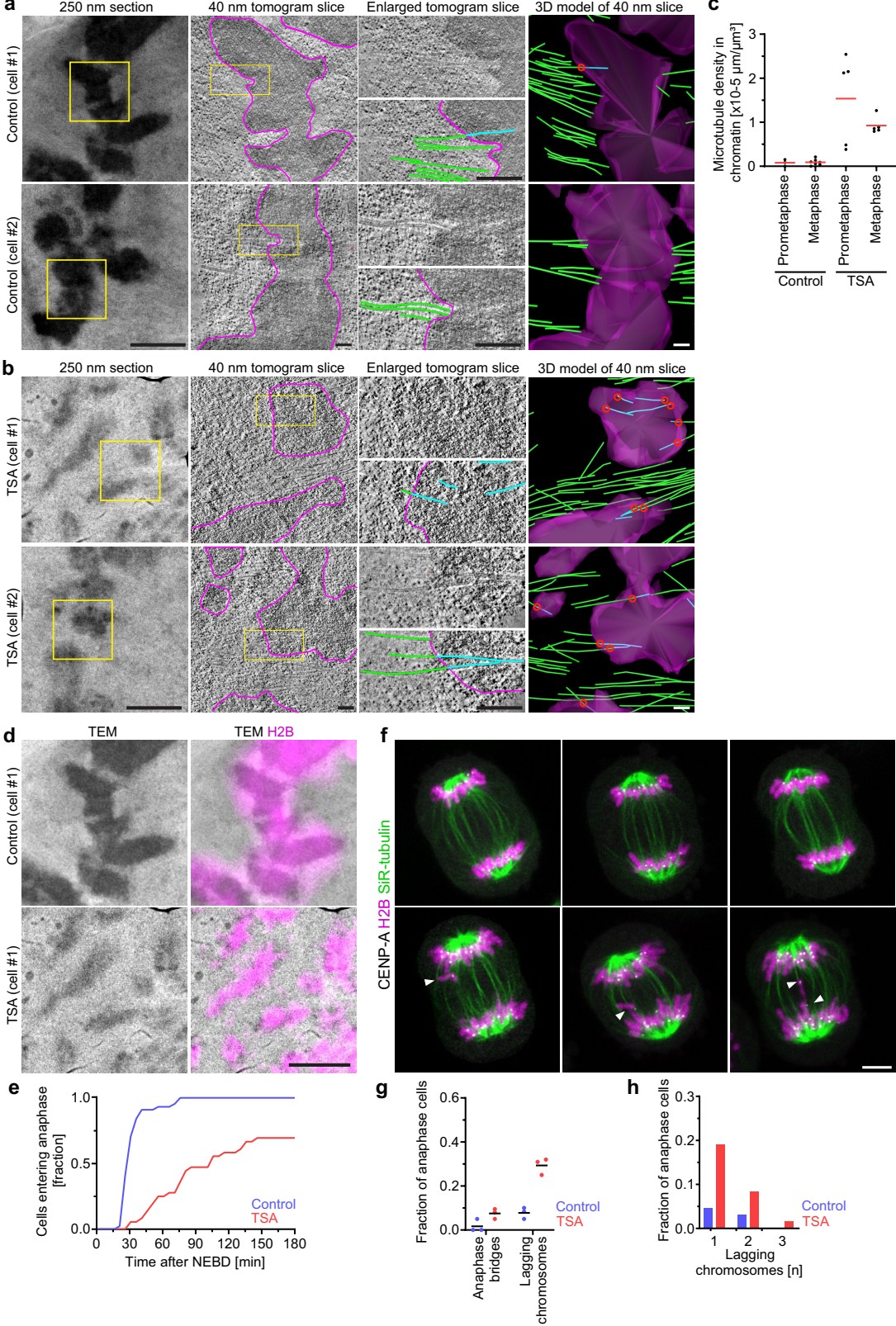

**Extended Data Fig. 6** | See next page for caption.

**Extended Data Fig. 6 | Correlative fluorescence and electron microscopy of mitotic cells and analysis of chromosome segregation by live imaging after TSA treatment. a**, **b**, Electron tomograms of prometaphase WT Hela cells, untreated (a) or treated with TSA (b). Magenta: chromatin surfaces; green: microtubules in cytoplasm; cyan: microtubules in chromatin; red circles: microtubule perforation sites at chromosome surface. Representative example regions for control prometaphase ($n$ = 3), control metaphase ($n$ = 7), TSA prometaphase ($n$ = 5) and TSA metaphase ($n$ = 5); example regions are from 10 tomograms per condition from 7 different cells each. **c**, Quantification of microtubule density in chromatin regions of prometaphase or metaphase cells in the absence or presence of TSA. Data shown in Fig.1e, f separated by mitotic stage, $n$ = 10 tomograms from 7 different cells for each condition. Bars indicate mean. **d**, Correlative transmission electron microscopy and fluorescence microscopy of chromatin/H2B-mCherry in prometaphase WT Hela cells (related to **a**, control cell #1 and **b**, TSA cell #1). **e**, Mitotic progression analysis by time-lapse microscopy of HeLa cells expressing H2B-mRFP, in untreated control and TSA-treated cells. $n$ = 44 for control from 5 biological replicates, $n$ = 36 for TSA from 4 biological replicates. Time is relative to nuclear envelope disassembly (NEBD). **f**, Chromosome missegregation analysis by Airyscan imaging of live anaphase HeLa cells expressing H2B-mCherry and meGFP-CENP-A and stained with SiR-tubulin. Representative images of $n$=64 control cells and $n$ = 110 TSA-treated cells. Single Z-sections. **g**, Quantification of chromosome missegregation of cells as illustrated in **f**. Dots indicate biological replicates, bars indicate mean. $n$=64 cells for control, $n$ = 110 for TSA. **h**, Quantification of number of lagging chromosomes in cells as illustrated in **f**. Fraction of cells with 1, 2, or 3 lagging chromosomes. $n$ = 64 cells for control, $n$ = 110 for TSA. Biological replicates: $n$ = 2 (**a**–**h**). Scale bars, **a**,**b**, 250 nm section, 2 µm; tomogram slices and 3D model, 200 nm; **d**, 2 µm; **f**, 5 µm.

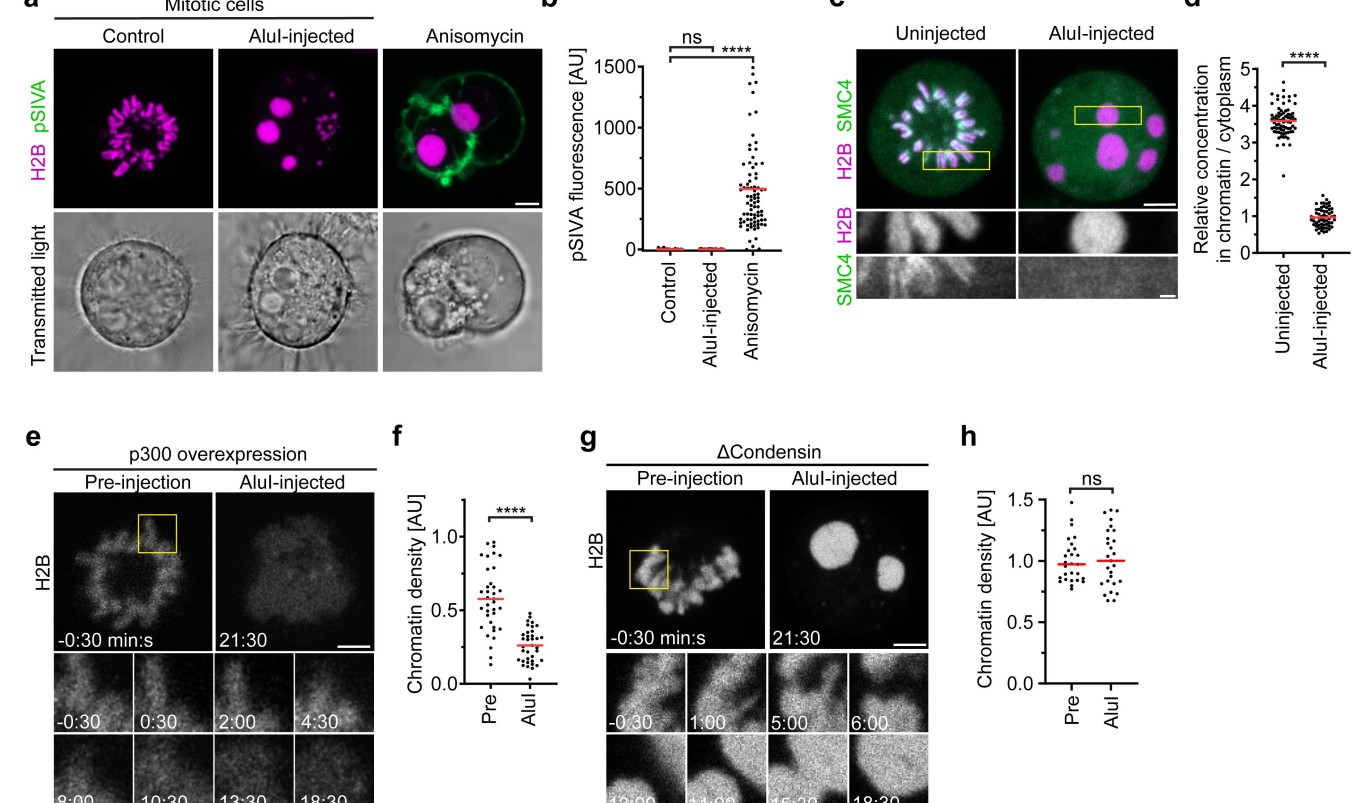

**Extended Data Fig. 7 | Analysis of AluI-fragmented chromosomes.**
**a**,**b**, Measurement of early apoptosis marker pSIVA after AluI-injection.
**a**, Hela cells expressing H2B-mCherry were treated with STLC and microinjected with AluI to induce chromatin fragmentation as in Fig. 2a, or treated with anisomycin to induce apoptosis as positive control. AluI-injected cells were imaged 1 h after injection. **b**, Quantification of pSIVA fluorescence on cell surface of untreated, AluI-injected, and anisomycin treated cells. $n = 77$ cells for control, $n = 41$ for AluI-injected, $n = 83$. Bars indicate mean, significance was tested by a two-tailed Mann-Whitney test (AluI-injected, $P = 0.446$; anisomycin, $P < 10^{-15}$, precision limit of floating-point arithmetic). **c**,**d**, Analysis of Smc4 localization after AluI-injection. **c**, Hela Smc4-HaloTAG cells expressing H2B-mCherry were stained with OregonGreen-488 HaloTAG ligand and mitotic cells were injected with AluI to fragment chromosomes. **d**, Quantification of Smc4-Halo fluorescence on chromatin relative to the cytoplasm of cells as in **c**. $n = 75$ uninjected cells, $n = 60$ AluI-injected cells. Bars indicate mean,

significance was tested by a two-tailed Mann-Whitney test ($P < 10^{-15}$, precision limit of floating-point arithmetic). **e**,**f**, Chromatin fragmentation in cells overexpressing p300-HAT. **e**, AluI injection (t = 0 min) during time-lapse microscopy of cells expressing H2B-mCherry and expressing p300-HAT. **f**, Quantification of chromatin density in cells as in **e**, normalized to the mean of mock-transfected, non-injected cells. $n = 12$ cells, 3 ROIs per cell. Bar indicates mean, significance was tested by a two-tailed Mann-Whitney test ($P = 2.075 \times 10^{-10}$). **g**,**h**, AluI-fragmentation after condensin depletion. **g**, Smc4-AID HeLa cells expressing H2B-mCherry were treated 3 h with 5-PhIAA to deplete condensin and mitotic cells were then injected with AluI (t = 0 min) during time-lapse microscopy. **h**, Quantification of chromatin density before and after injection of AluI, normalized to the mean of untreated pre-injection cells. $n = 7$ cells, 3 ROIs each. Bars indicate mean, significance was tested by a two-tailed Mann-Whitney test ($P = 0,887$). Biological replicates: $n = 3$ (**a**,**b**,**e**,f); $n = 2$ (**c**,**d**); $n = 5$ (**g**,**h**). Scale bars, **a**,**c**,**e**,**g** 5 µm; insert **c**,**e**,**g** 1 µm.

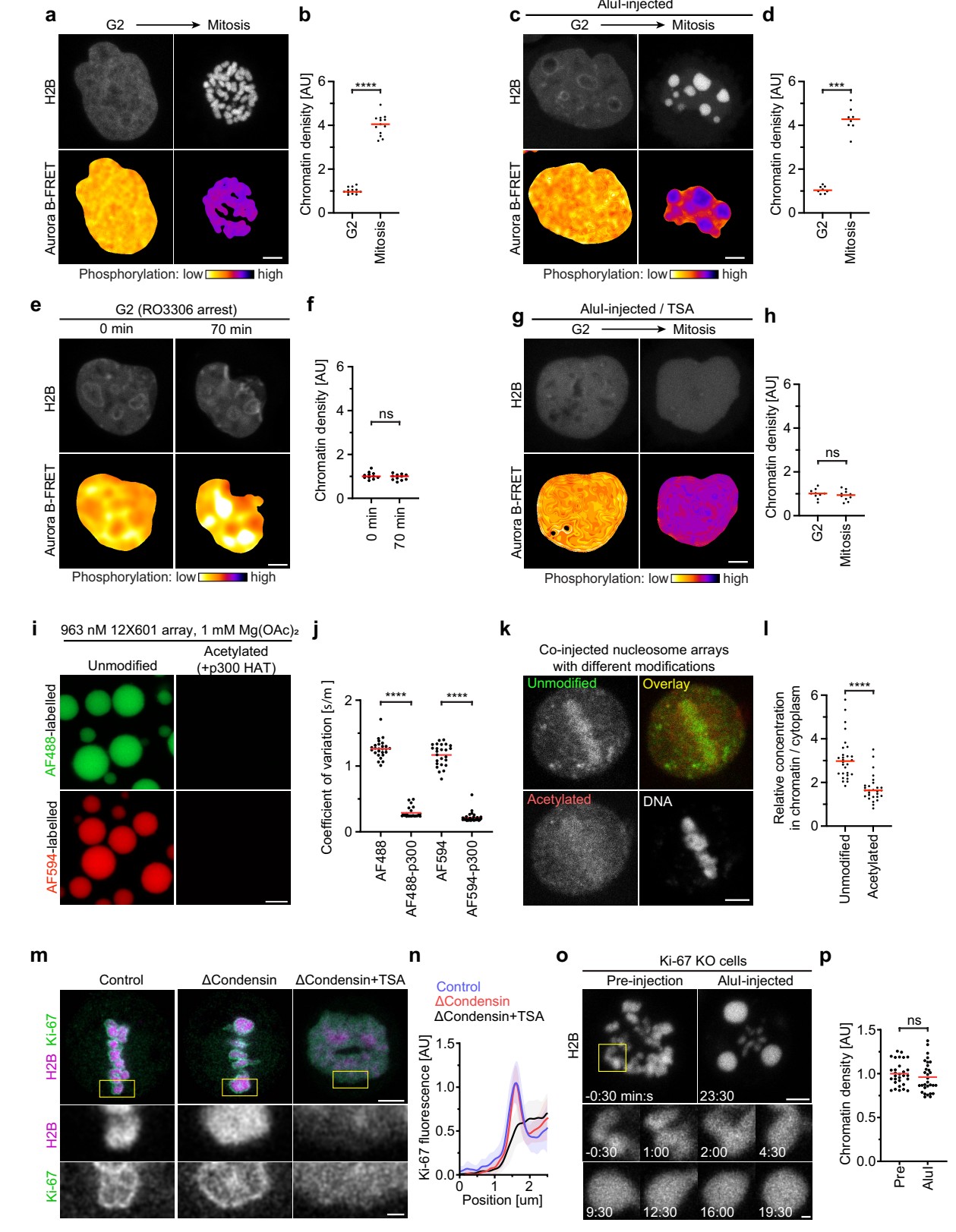

**Extended Data Fig. 8** | See next page for caption.

**Extended Data Fig. 8 | Analysis of chromatin phase transitions and role of Ki-67. a**, Chemical induction of G2-to-mitosis transition. HeLa cell expressing Aurora B-FRET biosensor was synchronized to G2 by RO3306 and then induced to enter mitosis by removing RO3306 and adding okadaic acid (OA). Mitotic entry was detected by chromosome compaction and FRET signal. Projection of 9 Z-sections. **b**, Quantification of chromatin density in G2 and mitosis as in **a** for $n = 13$ cells. Bars indicate mean; significance was tested by a two-tailed Mann-Whitney test ($P = 1.923 \times 10^{-7}$). **c**, Chromatin was fragmented in G2 cells by injection of AluI and mitosis subsequently induced as in **a**. Projection of 9 Z-sections. **d**, Quantification of chromatin density in G2 and mitosis as in **c** for $n = 8$ cells. Bars indicate mean; significance was tested by a two-tailed Mann-Whitney test ($P = 1.554 \times 10^{-4}$). **e,f**, Chromatin fragment localization in G2-arrested cells. **e**, HeLa cells expressing Aurora B-FRET biosensor were synchronized to G2 by RO3306 and microinjected in the nucleus with AluI. G2 state was retained in presence of RO3306 as indicated by FRET signal. t = 0 minutes refers to the first time point of the recorded time-lapse. **f**, Quantification of chromatin density in cells as in **e**, normalized to the mean of t = 0 min. $n = 11$ cells. Bars indicate mean; significance tested by a two-tailed Mann-Whitney test ($P = 0.438$). **g**, Chromatin was fragmented in TSA-treated G2 cells by injection of AluI and mitosis was subsequently induced as in **a**. Projection of 9 Z-sections. **h**. Quantification of chromatin density in G2 and mitosis as in **g** for $n = 10$ cells. Bars indicate mean; significance was tested by a two-tailed Mann-Whitney test ($P = 0.481$). **i,j**, In vitro liquid-liquid phase separation behaviour of unmodified or acetylated nucleosome arrays. **i**, 12X601 Nucleosome arrays labelled with fluorophores as indicated were treated with recombinant p300 histone acetyltransferase or no enzyme and then subjected to identical phase separation buffers for 30 min. **j**, Quantification of nucleosome array self-association into condensates by coefficient of variation (CV = $\sigma/\mu$) in images as in **i**. $n = 26$ for AlexaFlour488

array (AF488), $n = 25$ for acetylated AlexaFluor488 array (AF488-p300), $n = 25$ for AlexaFluor594 array (AF594), $n = 30$ for acetylated AlexaFluor594 array (AF594-p300). Bars indicate mean; significance tested by a two-tailed Mann-Whitney test (AF488-Ac, $P = 0.8 \times 10^{-14}$; AF594-Ac, $P < 10^{-15}$, precision limit of floating-point arithmetic). **k**, Microinjection of synthetic nucleosome arrays that were either untreated or pre-incubated with p300 acetyltransferase into live mitotic cells, for $n = 28$ cells. Unmodified and acetylated nucleosome arrays were labelled by distinct fluorescent dyes. DNA was counterstained with DAPI. **l**, Quantification of unmodified and acetylated nucleosome array partitioning into mitotic chromatin. Bars indicate mean; significance was tested by a two-tailed Mann-Whitney test ($P = 1.645 \times 10^{-9}$). **m,n**, Ki-67 localization in mitotic cells after Smc4-degradation in the absence and presence of TSA. **m**, Cells expressing H2B-mCherry were transfected with a construct for expression of mNeonGreen-tagged Ki-67 and imaged without further perturbations (control) or treated with 5-PhIAA for 3 h to degrade Smc4 (ΔCondensin) or 5-PhIAA and TSA to additionally suppress mitotic histone deacetylation (ΔCondensin+TSA). Single Airyscan Z-section. **n**, Distribution of Ki-67 across the surface of mitotic chromatin in cells as in **e**. Line profiles were drawn perpendicularly across the chromatin/cytoplasm boundary in a single Airyscan Z-section. $n = 5$ cells for control, $n = 7$ for ΔCondensin, $n = 8$ for ΔCondensin+TSA. 2-3 line profiles per cell. Curves indicate mean +/− SD. **o,p** AluI chromatin fragmentation in Ki-67 knockout cells. **o**, Mitotic Ki-67 knockout HeLa cell expressing H2B-mCherry was injected with AluI (t = 0 min) during time-lapse microscopy. **p**, Quantification of chromatin density before and after injection of AluI, normalized to the mean of untreated pre-injection cells. $n = 10$ cells, 3 ROIs each. Bars indicate mean, significance was tested by a two-tailed Mann-Whitney test ($P = 0.201$). Biological replicates: $n = 2$ (**a**–**l**); $n = 3$ (**m,n**); $n = 2$ (**o,p**). Scale bars, 5 μm, inserts 1 μm.

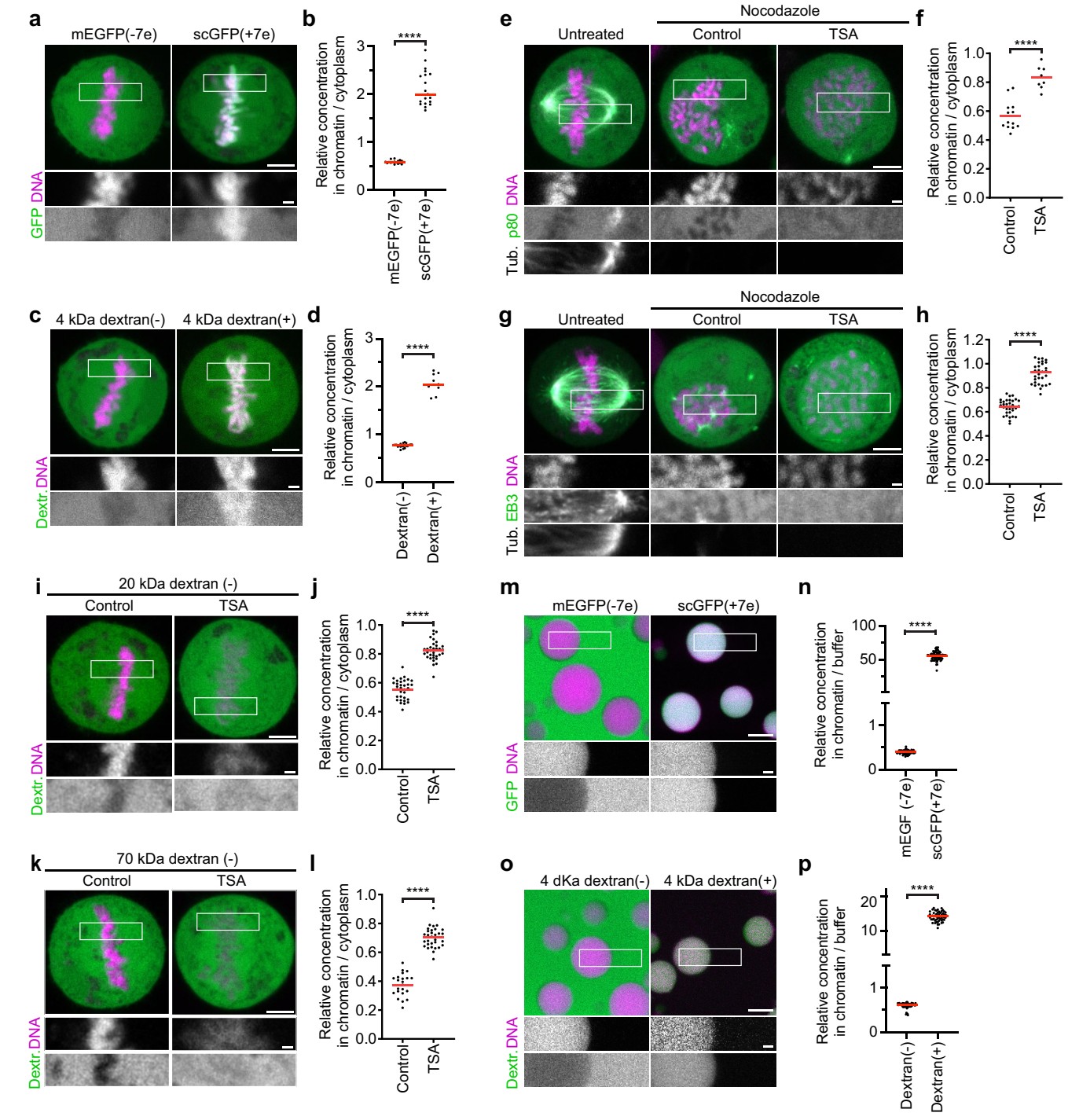

**Extended Data Fig. 9** | See next page for caption.

**Extended Data Fig. 9 | Partitioning of proteins and dextrans relative to mitotic chromatin and synthetic nucleosome array condensates.**
**a**, **b**, Partitioning of GFP surface charge variants relative to chromatin in metaphase cells. **a**, Live metaphase cells after injection of recombinant mEGFP(−7e) or scGFP(+7e). DNA was stained with Hoechst 33342. **b**, Quantification of GFP fluorescence in chromatin relative to cytoplasm. $n = 17$ for mEGFP(−7e), $n = 20$ for scGFP(+7e). Bars indicate mean; significance tested by a two-tailed Mann-Whitney test ($P = 1.257 \times 10^{-10}$). **c**, **d**, Partitioning of charge-modified fluorescent dextrans relative to chromatin in metaphase cells. **c**, Live metaphase cells after injection of negatively or positively charged 4.4 kDa FITC-dextrans. DNA was stained with Hoechst 33342. **d**, Quantification of dextran fluorescence in chromatin relative to cytoplasm. $n = 21$ for 4.4 kDa FITC-dextran(−), $n = 10$ for 4.4 kDa FITC-dextran(+). Bars indicate mean; significance tested by a two-tailed Mann-Whitney test ($P = 4.509 \times 10^{-8}$). **e**−**h** Partitioning of microtubule-associated proteins (MAPs) with mitotic chromatin in the absence and presence of TSA. DNA was stained with Hoechst 33342 and microtubules with SiR-Tubulin. **e**, HeLa cells expressing p80-Katanin-mNeongreen were treated with nocodazole and TSA for 3 h as indicated and imaged live. Single confocal Z-sections. **f**, Quantification of p80-Katanin-mNeongreen fluorescence in mitotic chromatin relative to cytoplasm. $n = 13$ cells for control, $n = 8$ for TSA. Bars indicate mean, significance was tested by a two-tailed Mann-Whitney test ($P = P = 3.931 \times 10^{-5}$). **g**, HeLa cells expressing EB3-mNeongreen were treated with nocodazole and TSA for 3 h as indicated and imaged live. Single confocal Z-sections. **h**, Quantification of EB3-mNeongreen fluorescence in mitotic chromatin relative to cytoplasm. $n = 33$ cells for control, n = 31 for TSA. Bars indicate mean, significance was tested by a two-tailed Mann-Whitney test ($P < 10^{-15}$, precision limit of floating-point arithmetic). **i**−**l**, Partitioning of negatively charged higher molecular weight dextrans relative to mitotic chromatin in the absence and presence of TSA. DNA was stained with Hoechst 33342. **i**, Live metaphase cells after injection of 20 kDa FITC-dextran. **j**, Quantification of dextran fluorescence in mitotic chromatin relative to cytoplasm. $n = 33$ for control, $n = 33$ for TSA. Bars indicate mean; significance tested by a two-tailed Mann-Whitney test ($P < 10^{-15}$, precision limit of floating-point arithmetic). **k**, Live metaphase cells after injection of 70 kDa FITC-dextran. **l**, Quantification of dextran fluorescence in mitotic chromatin relative to cytoplasm. $n = 22$ for control, $n = 33$ for TSA. Bars indicate mean; significance tested by a two-tailed Mann-Whitney test ($P = 4 \times 10^{-14}$). **m**, **n**, Partitioning of GFP surface charge variants relative to nucleosome array condensates *in vitro*. **m**, Chromatin condensates were formed *in vitro* by exposing 12X601 nucleosome arrays to phase separation buffer. GFP charge variants were added for 10 minutes and then imaged. DNA was stained with Hoechst 33342. **n**, Quantification of GFP fluorescence in chromatin relative to buffer. $n = 69$ for mEGFP(−7e), $n = 73$ for scGFP(+7e). Bars indicate mean; significance tested by a two-tailed Mann-Whitney test ($P < 10^{-15}$, precision limit of floating-point arithmetic). **o**, **p**, Partitioning of charge modified dextrans relative to liquid nucleosome array condensates *in vitro*. **o**, Liquid chromatin droplets were formed were formed as in **m**. Charge modified 4.4 kDa dextrans were added for 10 minutes and then imaged. DNA was stained with Hoechst 33342. **p**, Quantification of dextran fluorescence in chromatin relative to buffer. $n = 69$ for 4.4 kDa dextran(−), $n = 57$ for 4.4 kDa dextran(+). Bars indicate mean; significance tested by a two-tailed Mann-Whitney test ($P < 10^{-15}$, precision limit of floating-point arithmetic). Biological replicates: $n = 2$ (**a**−**h**,**m**−**p**); $n = 3$ (**i**−**l**). Scale bars, 5 μm, inserts 1 μm.

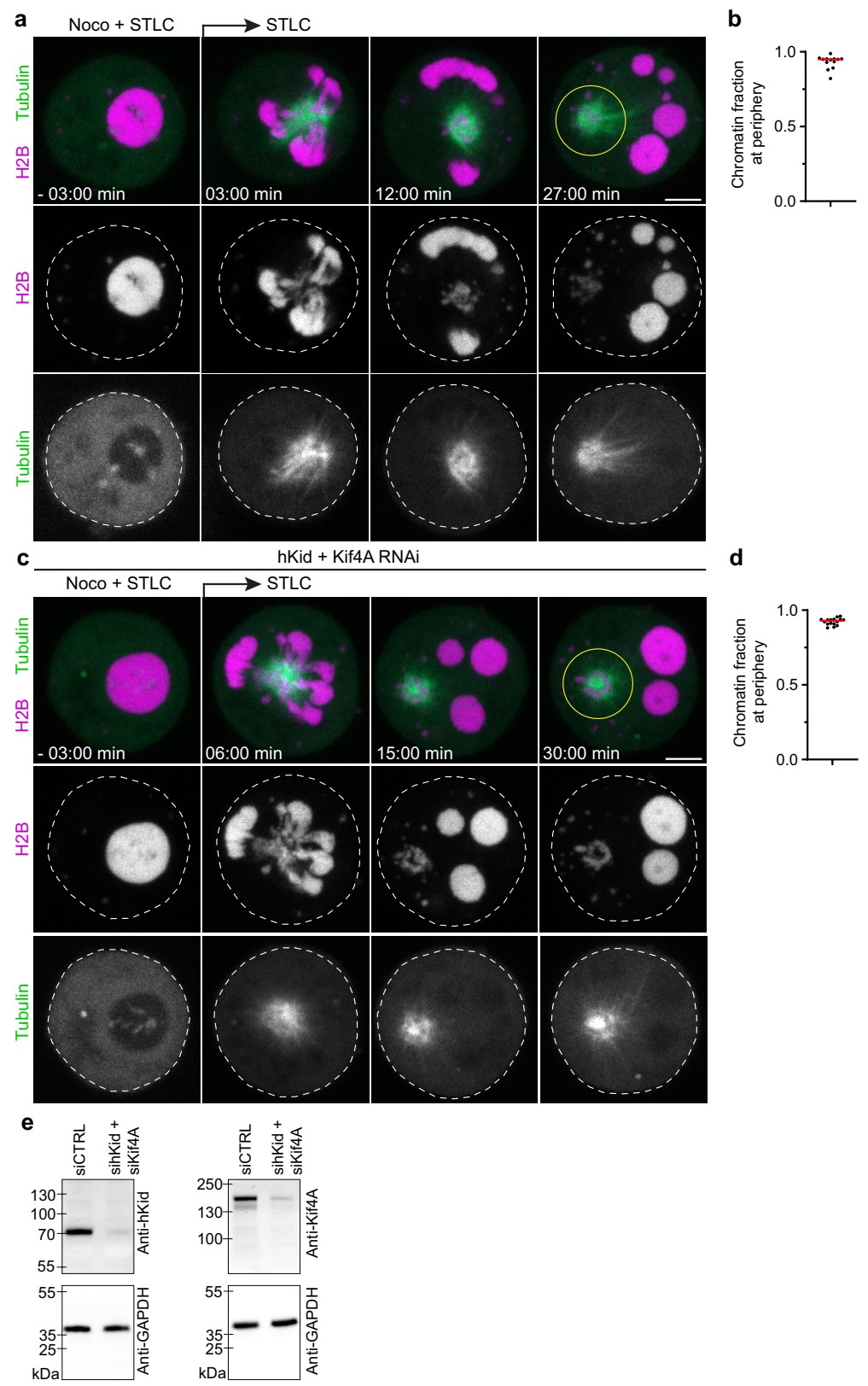

**Extended Data Fig. 10** | See next page for caption.

**Extended Data Fig. 10 | Microtubules push liquified chromatin away from the spindle pole independently of hKid and Kif4A. a**, Time-lapse microscopy of liquified chromatin during chemically-induced assembly of monopolar spindles. Live mitotic HeLa cells expressing H2B-mCherry and eGFP-a-tubulin were treated with nocodazole and STLC and then injected with AluI. Nocodazole was removed at t = 0 min during time-lapse imaging. Projection of 4 z-sections. Representative example of $n$ = 13 cells. **b**, Quantification of chromatin localization at the cell periphery relative to the region around the spindle monopole at = 36 min for cells as shown in **a**. **c**, Time-lapse microscopy of liquified chromatin during spindle assembly as in **a** for cells depleted of Kid and Kif4a by RNAi. **d**, Quantification as in **b** for hKid/Kif4a-RNAi cells. $n$ = 16 cells. Bar indicates mean; significance tested by a two-tailed Mann-Whitney test ($P$ = 0.215) **e**, Validation of RNAi efficiency by Western Blotting. Samples were collected 30 h after transfection of siRNAs targeting hKid and Kif4a and probed by antibodies as indicated; $n$ = 2 experiments. For gel source data, see Supplementary Figure 1b. Biological replicates: $n$ = 4 (**a,b**); $n$ = 3 (**c,d**); $n$ = 2 (**e**). Scale bars, 5 μm.

Maximilian W.G. Schneider

# Reporting Summary

## Statistics

For all statistical analyses, confirm that the following items are present in the figure legend, table legend, main text, or Methods section.

| n/a | Confirmed | |
|---|---|---|
| ☐ | ☒ | The exact sample size (*n*) for each experimental group/condition, given as a discrete number and unit of measurement |
| ☐ | ☒ | A statement on whether measurements were taken from distinct samples or whether the same sample was measured repeatedly |
| ☐ | ☒ | The statistical test(s) used AND whether they are one- or two-sided *Only common tests should be described solely by name; describe more complex techniques in the Methods section.* |
| ☒ | ☐ | A description of all covariates tested |
| ☐ | ☒ | A description of any assumptions or corrections, such as tests of normality and adjustment for multiple comparisons |
| ☐ | ☒ | A full description of the statistical parameters including central tendency (e.g. means) or other basic estimates (e.g. regression coefficient) AND variation (e.g. standard deviation) or associated estimates of uncertainty (e.g. confidence intervals) |
| ☐ | ☒ | For null hypothesis testing, the test statistic (e.g. *F*, *t*, *r*) with confidence intervals, effect sizes, degrees of freedom and *P* value noted *Give P values as exact values whenever suitable.* |
| ☒ | ☐ | For Bayesian analysis, information on the choice of priors and Markov chain Monte Carlo settings |
| ☒ | ☐ | For hierarchical and complex designs, identification of the appropriate level for tests and full reporting of outcomes |
| ☒ | ☐ | Estimates of effect sizes (e.g. Cohen's *d*, Pearson's *r*), indicating how they were calculated |

*Our web collection on statistics for biologists contains articles on many of the points above.*

## Software and code

Policy information about availability of computer code

| Data collection | - Images: LSM780 operated by ZEN 2011 software or LSM980 operated by ZEN Blue 2020 Software.<br>- Western blotting: BioRad Imager operated by Image Lab 6.0.1. |
|---|---|
| Data analysis | - Image analysis: ImageJ/Fiji (version 1.53c, Java 1.8.0_66 (64-bit).<br>- Plots and statistical analysis: GraphPad Prism (version 8.1.1 (330)) or Python (3.7).<br>- Electron tomogram analysis: SerialEM 4.0 and IMOD/3dmod 4.11 |

For manuscripts utilizing custom algorithms or software that are central to the research but not yet described in published literature, software must be made available to editors and reviewers. We strongly encourage code deposition in a community repository (e.g. GitHub). See the Nature Portfolio guidelines for submitting code & software for further information.

## Data

Policy information about availability of data

All manuscripts must include a data availability statement. This statement should provide the following information, where applicable:

- Accession codes, unique identifiers, or web links for publicly available datasets
- A description of any restrictions on data availability
- For clinical datasets or third party data, please ensure that the statement adheres to our policy

All datasets generated in this study are available from the corresponding authors upon request. Source data are provided with this paper. Raw microscopy data are available and will be provided from by the corresponding authors upon request, given the large file sizes that are involved. No restrictions apply to the availability of microscopy data.

# Field-specific reporting

Please select the one below that is the best fit for your research. If you are not sure, read the appropriate sections before making your selection.

☒ Life sciences        ☐ Behavioural & social sciences        ☐ Ecological, evolutionary & environmental sciences

For a reference copy of the document with all sections, see nature.com/documents/nr-reporting-summary-flat.pdf

# Life sciences study design

All studies must disclose on these points even when the disclosure is negative.

| | |
|---|---|
| Sample size | No sample-size calculations were performed. Sample sizes were chosen as large as possible while taking into account the experimental effort required to generate the respective data. Adequate statistics has been applied throughout the manuscript in order to make sure that the observed effects are significant given the reported sample size. |
| Data exclusions | These two criteria were pre-established before any analyses after data acquisition:<br>- Cells that die or move out of the field of view during movie acquisition during long time-lapse acquisitions<br>- For microinjection experiments: cells in which the plasma membrane ruptured or co-injected fluorescent tracer leaked out of the cytoplasm within <5 minutes after micro injection. |
| Replication | Reported experiments were repeated in at least 2 biological replicates with consistent results. Unless otherwise noted, in all analyses the biological replicates have been combined. |
| Randomization | Not relevant as grouping was not applied. |
| Blinding | To minimize potential human bias, most experiments were  analyzed by prerecorded Fiji command macros (Fig. 1 a-c, 2a-i, 3a-f, ED Fig. 1a-e, ED Fig. 2a-d, 3a-h, 4a-d, 5a-e, 7a-h, 8a-l, o, p, 9a-d, m-p). When manual annotation was required, blinding precautions were made. Annotation of chromatin and microtubules in electron tomograms was performed manually (Fig. 1d-f, ED Fig. 6a-c). Ki-67 distribution was measured along line profiles drawn perpendicular to the chromatin marker surface (Fig. 2g-i), ED Fig. 8 m, n). Center of mass of spindle pole was chosen manually based on point of highest fluorescence density the respective tubulin marker (Fig. 4a, b, ED Fig. 2a-d, Fig. 10a-d). Fraction of detached kinetochores was determined manually by measuring the distance of CENP-A marker to closest chromatin marker surface (ED Fig. 1f, g, Fig. 2e). Time form NEBD to anaphase onset was manually determined in time lapse movies and fraction of lagging chromosomes was determined manually in high-resolution 3D images of anaphase cells (ED FIg. 6f-h). Partitioning coefficient of MAPs and dextrans was measured along manually defined line profiles perpendicular to the metaphase plate (ED Fig. e-l). |

# Reporting for specific materials, systems and methods

We require information from authors about some types of materials, experimental systems and methods used in many studies. Here, indicate whether each material, system or method listed is relevant to your study. If you are not sure if a list item applies to your research, read the appropriate section before selecting a response.

### Materials & experimental systems

| n/a | Involved in the study |
|---|---|
| ☐ | ☒ Antibodies |
| ☐ | ☒ Eukaryotic cell lines |
| ☒ | ☐ Palaeontology and archaeology |
| ☒ | ☐ Animals and other organisms |
| ☒ | ☐ Human research participants |
| ☒ | ☐ Clinical data |
| ☒ | ☐ Dual use research of concern |

### Methods

| n/a | Involved in the study |
|---|---|
| ☒ | ☐ ChIP-seq |
| ☒ | ☐ Flow cytometry |
| ☒ | ☐ MRI-based neuroimaging |

# Antibodies

| | |
|---|---|
| Antibodies used | Primary antibodies:<br>- anti-CENP-A (ENZO Life Sciences, clone 3-19, ADI-KAM-CC006-E, 10161910), dilution  1:1000<br>- Pericentrin (Abcam, clone EPR21987, ab220784, GR3284309-1), dilution  1:2000<br>- Acetylated histone 2B (Millipore, 07-373, 3092508), dilution 1:500<br>- Acetylated histone 3 (Merck, 06-599, 3260200), dilution 1:500<br>- Acetylated histone 4K16 (Abcam, clone EPR1004, ab109463, GR284778-8), dilution  1:500<br>- γH2A.X (BioLegend, clone 2F3, 613402, B283251), dilution 1:1000<br>- Cyclin B1 (Cell Signaling, clone D5C10, 12231S, 7), dilution  1:800<br>- Smc4 (Abcam, ab229213, GR3228108-5), dilution  1:1000<br>- GAPDH (Abcam, ab9485, GR3212164-2), dilution  1:2500<br>- hKid/Kif22 (Abcam, clone EP2747Y, ab75783, GR129278-4), dilution  1:1000 |

- Kif4a (Abcam, clone EPR5459, ab124903, GR96215-7), dilution  1:1000

Secondary antibodies:
- goat anti-mouse Alexa Fluor 488 (Molecular Probes, A11001, 1787787), dilution  1:1000
- goat anti-rabbit Alexa Fluor 633 (Molecular Probes, A21071, 99E2-1), dilution  1:1000
- donkey anti-rabbit Alexa Fluor 488 (Molecular Probes, A21206, 1796375), dilution  1:1000
- horseradish peroxidase goat anti-mouse (Biorad, cat. number 1706516), dilution 1:5000
- horseradish peroxidase goat anti-rabbit (Biorad, cat. number 1706515), dilution 1:5000

**Validation**

anti-CENP-A antibody was validated by immunoblotting using an appropriate molecular weigth marker in Jurkat and Raji cell lines. References: Gene replacement strategies validate the use of functional tags on centromeric chromatin and invalidate an essential role for CENP-AK124ub: C. Salinas-Luypaert, et al.; Cell Rep. 37, 10924 (2021), Human Artificial Chromosomes that Bypass Centromeric DNA: G.A. Logsdon, et al.; Cell 178, 624 (2019), Phosphorylation of CENP-A on serine 7 does not control centromere function: V. Barra, et al.; Nat. Commun. 10, 175 (2019), CENP-A Modifications on Ser68 and Lys124 Are Dispensable for Establishment, Maintenance, and Long-Term Function of Human Centromeres: D. Fachinetti, et al.; Dev. Cell 40, 104 (2017), Centromeres are maintained by fastening CENP-A to DNA and directing an arginine anchor-dependent nucleosome transition: L.Y. Guo, et al.; Nat. Commun. 8, 15775 (2017).

anti-Pericentrin antibody was validated by IP and immunoblotting usign appropriate molecular weigth markers, intracellular flow cytometry analysis and immunofluorescence staining in HepG2, NIH/3T3 and HeLa cell lines. References: Vergarajauregui S  et al. AKAP6 orchestrates the nuclear envelope microtubule-organizing center by linking golgi and nucleus via AKAP9. Elife 9:N/A (2020).

anti-acetylated histone 2B antibody was validated using immunoblot with appropriate molecular weigth markers using extract from HeLa cells with an extract of sodium butyrate treated cells as positive control.

anti-acetylated histone 3 antibody was validated by immunoblot using appropriate molecular weigth markers in extracts of Hela cells with sodium butyrate treated cells as positive control and recombinant histone 3 as a negative control.

anti-acetylated histone 4K16 antibody was validated using immunoblot with appropriate molecular weigth markers using extract from HeLa and C6 cell lines and mouse spleen lysate, with TSA treated cells as positive control, by immunofluorescence in HeLa cells with TSA treated cells as a positive control and by intracellular flow cytometry staining in HeLa cells. References (limited to the last 2 years): Song Z  et al. Effects of histone H4 hyperacetylation on inhibiting MMP2 and MMP9 in human amniotic epithelial cells and in premature rupture of fetal membranes. Exp Ther Med 21:515 (2021), Shalini V  et al. Genome-wide occupancy reveals the localization of H1T2 (H1fnt) to repeat regions and a subset of transcriptionally active chromatin domains in rat spermatids. Epigenetics Chromatin 14:3 (2021), Contreras SM  et al. Resveratrol induces H3 and H4K16 deacetylation and H2A.X phosphorylation in Toxoplasma gondii. BMC Res Notes 14:19 (2021), Huang R  et al. HDAC11 inhibition disrupts porcine oocyte meiosis via regulating a-tubulin acetylation and histone modifications. Aging (Albany NY) 13:8849-8864 (2021), Navarro-Carrasco E & Lazo PA VRK1 Depletion Facilitates the Synthetic Lethality of Temozolomide and Olaparib in Glioblastoma Cells. Front Cell Dev Biol 9:683038 (2021), Sui L  et al. HDAC11 promotes meiotic apparatus assembly during mouse oocyte maturation via decreasing H4K16 and a-tubulin acetylation. Cell Cycle 19:354-362 (2020), Du L  et al. Loss of SIRT4 promotes the self-renewal of Breast Cancer Stem Cells. Theranostics 10:9458-9476 (2020), Sun X  et al. Histone deacetylase inhibitor valproic acid attenuates high glucose-induced endoplasmic reticulum stress and apoptosis in NRK-52E cells. Mol Med Rep 22:4041-4047 (2020), Koziol K  et al. Changes in ?H2AX and H4K16ac levels are involved in the biochemical response to a competitive soccer match in adolescent players. Sci Rep 10:14481 (2020), Kubatka P  et al. Rhus coriaria L. (Sumac) Demonstrates Oncostatic Activity in the Therapeutic and Preventive Model of Breast Carcinoma. Int J Mol Sci 22:N/A (2020).

anti-yH2A.X antibody was validated by immunoblot using appropriate molecular weigth markers, with extract from HeLa cells, using extract from UV-treated cells as positive control and by immunofluorescence of HeLa cells using UV treated cells as a positive control. Application references: Akbay A, et al. 2008. Am J Pathol. 173:536. (IHC), Mochizuki K, et al.2008.J cell Sci.121:2148. (IF), Xiao R, et al. 2007. Mol Cell Biol.27:5393. (IF), Rossi DJ, et al. 2007. Nature. 447:725. (IF), Loidl J, et al. 2009. Mol Cell Biol. 20:2048. (IF), Beels L, et al. 2009. Circulation. 120:1903. (IF), Yamada C, et al. 2010 J. Biol. Chem. 285:16693. (WB), Bu Y, et al. 2010, Biochem Biophys Res Commun. 397:157. (WB), Massignan T, et al. 2010. J. Biol Chem. 285:7752. (WB).

anti-CyclinB1 antibody was validated by immunoblot using appropriate molecular weigth markers in extracts from HT-29, HeLa, Jurkat and HCT 116 cells and in extracts from synchronized HT-29 cells following thymidine block and release for various times, indicating cell cycle dependent chagnes in Cyclin B1 levels as well as immunofluorescence in HT-29 cells. REferences: Zhang J, Li A, Sun H, Xiong X, Qin S, Wang P, Dai L, Zhang Z, Li X, Liu Z. Amentoflavone triggers cell cycle G2/M arrest by interfering with microtubule dynamics and inducing DNA damage in SKOV3 cells. Oncol Lett. 2020 Nov;20(5):168. doi: 10.3892/ol.2020.12031. Epub 2020 Aug 27. PMID: 32934735; PMCID: PMC7471765, Lee CAA, Banerjee P, Wilson BJ, Wu S, Guo Q, Berg G, Karpova S, Mishra A, Lian JW, Tran J, Emmerich M, Murphy GF, Frank MH, Frank NY. Targeting the ABC transporter ABCB5 sensitizes glioblastoma to temozolomide-induced apoptosis through a cell-cycle checkpoint regulation mechanism. J Biol Chem. 2020 May 29;295(22):7774-7788. doi: 10.1074/jbc.RA120.013778. Epub 2020 Apr 20. PMID: 32317280; PMCID: PMC7261782., Zheng Z, Wu M, Zhang J, Fu W, Xu N, Lao Y, Lin L, Xu H. The Natural Compound Neobractatin Induces Cell Cycle Arrest by Regulating E2F1 and Gadd45α. Front Oncol. 2019 Jul 17;9:654. doi: 10.3389/fonc.2019.00654. PMID: 31380287; PMCID: PMC6653061., Zheng Z, Wu M, Zhang J, Fu W, Xu N, Lao Y, Lin L, Xu H. The Natural Compound Neobractatin Induces Cell Cycle Arrest by Regulating E2F1 and Gadd45α. Front Oncol. 2019 Jul 17;9:654. doi: 10.3389/fonc.2019.00654. PMID: 31380287; PMCID: PMC6653061, Tan X, Yuan G, Wang Y, Zou Y, Luo S, Han H, Qin Z, Liu Z, Zhou F, Liu Y, Yao K. RAB20 Promotes Proliferation via G2/M Phase through the Chk1/cdc25c/cdc2-cyclinB1 Pathway in Penile Squamous Cell Carcinoma. Cancers (Basel). 2022 Feb 22;14(5):1106. doi: 10.3390/cancers14051106. PMID: 35267417; PMCID: PMC8909501, Zhang X, Zhao J, Gao X, Pei D, Gao C. Anthelmintic drug albendazole arrests human gastric cancer cells at the mitotic phase and induces apoptosis. Exp Ther Med. 2017 Feb;13(2):595-603. doi: 10.3892/etm.2016.3992. Epub 2016 Dec 22. PMID: 28352336; PMCID: PMC5348670.

anti-Smc4 antibody was validated by immunoblot using appropriate molecular weigth markers using extracts from HEK-293T, A431, HeLa and HepG2 cells. We verified the molecular weight of the target protein by immunoblotting with extracts from HeLa cells by using appropriate molecular weight range markers. 5-PhIAA treatment caused loss of the band of Smc4-mAID-HaloTag.

anti-GAPDH antibody was validated by immunoblot using appropriate molecular weigth markers in lysate from Hela, Jurkat, A431, HEK-293 and HepG2 cells, by immunofluorescence in HeLa cells and by ELISA using extracts from human fibroblast. References: Ni W et al. Preventing oxaliplatin-induced neuropathic pain: Using berberine to inhibit the activation of NF-?B and release of pro-inflammatory cytokines in dorsal root ganglions in rats. Exp Ther Med 21:135 (2021), Ge R  et al. Upregulated microRNA-126 induces apoptosis of dental pulp stem cell via mediating PTEN-regulated Akt activation. J Clin Lab Anal 35:e23624 (2021), Zhang Y  et al. Long Non-coding RNA CASC15 Promotes Intrahepatic Cholangiocarcinoma Possibly through Inducing PRDX2/PI3K/AKT Axis. Cancer Res Treat 53:184-198 (2021), Reed JC  et al. Identification of an Antiretroviral Small Molecule That Appears To Be a Host-Targeting Inhibitor of HIV-1 Assembly. J Virol 95:N/A (2021), Yao W  et al. TNK2-AS1 upregulated by YY1 boosts the course of osteosarcoma

through targeting miR-4319/WDR1. Cancer Sci 112:893-905 (2021), Zhang Z   et al. lncRNA CASC9 sponges miR-758-3p to promote proliferation and EMT in bladder cancer by upregulating TGF-ß2. Oncol Rep 45:265-277 (2021).
anti-hKid/Kif22 antibody was validated by immunoblot using appropriate molecular weigth markers in extracts from wild-type and knockout HEK-293T cells and extracts from HeLa cells. We verified the molecular weigth of the target protein in extracts from HeLa cells by immunoblotting following RNAi which led to loss of the band. References: Li C  et al. NuSAP governs chromosome oscillation by facilitating the Kid-generated polar ejection force. Nat Commun 7:10597 (2016).
anti-Kif4A antibody was validated by immunoblot using appropriate molecular weigth controls in mouse thymus tissue lysate and extracts from HeLa and HEK293T cells as well as by intracellular immunofluorescence followed by flow cytometry in HEK293T cells. We verified the molecular weigth of the target protein in extracts from HeLa cells by immunoblotting following RNAi which led to loss of the band.

# Eukaryotic cell lines

Policy information about cell lines

| | |
|---|---|
| Cell line source(s) | All cells lines were derived from a HeLa cell line ("Kyoto" strain) obtained from S. Narumiya (Kyoto University, Japan). The original commercial source of HeLa cells is the American Type Culture Collection (ATCC). |
| Authentication | Wild-type HeLa Kyoto cells were validated by a Multiplex human Cell line Authentication test (MCA), 21.04.16. |
| Mycoplasma contamination | Routine mycoplasma tested showed all cell lines were free of mycoplasma contamination. |
| Commonly misidentified lines (See ICLAC register) | HeLa cells are not in the list of commonly misidentified cell lines. |

