## [Peer Review File · Nature]

Manuscript Title: A mitotic chromatin phase transition prevents perforation by microtubules

Reviewer Comments & Author Rebuttals

Reviewer Reports on the Initial Version:

Referees' comments:

Referee #1 (Remarks to the Author):

Schneider et al. examine how histone deacetylation can control the solubility of chromosomes to form an immiscible mitotic chromatin phase. The authors suggest that this phase acts as a physical barrier to microtubule perforation, thus protecting the genome during eukaryotic cell division. This work focuses on an interesting, fundamental question. However, there are critical discrepancies between the data and claims and (2) a key finding has already been published by the same group of authors, putting the novelty of this work into question (reference 13, Gibson et al). For these reasons, I cannot recommend it for publication in Nature. Below, I elaborate on these two points.

Figure 1

1) It is not clear if compound toxicity is a factor in many of the observations (lagging chromosomes, microtubule perforation). The dose dependent effects of this compound should be reported, and its reversibility should be demonstrated. This will help ensure that the observations reported are due to acetylation state and not to compound toxicity. While overexpressing the histone acetyltransferase p300, the authors observed the chromatin phenotypes observed with the compound. However, their main findings demonstrating increased microtubule perforation and differences in chromatin solubility were not reproduced with this overexpression approach.

2) The data presented in panel 1A does not match the conclusions drawn in the text. For example the authors state: "In condensin-depleted cells, we observed an unstructured mass of compact chromatin forming a plate between the spindle poles, indicating that polar ejection forces efficiently push bulk chromatin towards the spindle center". Similarly, the authors state that the localization of "loosely connected kinetochores towards the spindle poles confirms that condensin is required for chromosomes to resist pulling. Thus, condensin depletion allows the mechanisms responsible for the resistance to pushing and pulling forces to be decoupled". As only snapshots of cells are presented, it is incorrect to make conclusions about forces and how their origins can be decoupled.

3) Overall, it is unclear to me why the results from Figure 1 are discussed/presented in the context of pushing forces. Many parts of the manuscript (intro, results for Figure 1 and 4) should be re-written to omit the focus on spindle forces. The authors state that we do not understand how microtubule-generated pushing forces affect chromosomes and reason that we already know how condensin protects chromosomes from microtubule-generated pulling. This is problematic because it reduces the 3D chromatin structure to only 2 dimensions, with chromosomes experiencing 2 simple, opposite forces. A strain map across the chromosomes is likely more complicated than a simple pull/push.

4) The extent of microtubule perforation into chromosomes is not clear. As only 2 examples are presented in panel D and the quantification (E and F) is minimal, it is not clear if this is a subtle effect. An additional problem is that the 2 panels in fig 1D compare different regions between

control (pole-facing side) and treated (in between chromosomes) cells. Equivalent sides should be compared. A thorough classification of this perforation effect should include analyses that determine whether microtubule perforation is region-specific. Specifically, perforation should be mapped across the chromosome periphery. From the current data, one cannot tell if there are a few patches where microtubules penetrate the chromosomes, or if this is a global effect. Overall, judging from the small panels in Figure 1 and the subtle change in anaphase defects (Extended Figure 1F, G) (which may come from the compound alone), I am not convinced that this microtubule perforation phenotype is prominent.

5) Extended figure 1, related to this data needs improvement. In particular:

- a. S1D (pericentrin/CenP localization)- quantify this effect and include statistics. How many cells were used to show this?
- b. S1G- show the distribution of cells, as the mean is not very informative if the distribution is skewed. Also, show more examples of chromosomes in anaphase.

Figure 2

1) The authors investigate how “histone acetylation affects the material properties of chromatin” and conclude that “..acetylation regulates chromatin solubility”. These results were presented by many of these authors in Gibson et al, where acetylation was shown to modulate phase separation. While the results from Gibson et al focused on in vitro condensates, data from cells was also presented, diminishing the novelty of these findings. The authors cite their previous work, however, they do so without explicitly stating they previously showed that the chromatin acetylation state affects its phase separation. For example, Gibson et al. was cited after the following sentences:

- a. “Assembly of mitotic chromosomes involves DNA looping by condensins and chromatin compaction by global histone deacetylation.”
- b. “Nucleosomal interactions are thought to increase when histones are deacetylated during mitotic entry, contributing to global chromatin compaction”
- c. “...we used synthetic nucleosome arrays”

2) A hydrogel model is proposed to support their observations, but the basis of this model is unexplained. The definition of a hydrogel is not explained. Why is this model preferred over others? For example, why was a hydrogel proposed instead of a simple gel? It is also not clear if this is the first presentation of such a model describing chromosomes and if the data indeed supports this model.

3) Conclusions are drawn about the material property of chromosomes without enough evidence. A few examples:

- a. “Shortly after microinjection of Alu1, chromosomes lost their elongated shape, forming round condensates that fused to one another, indicating a liquid state.” Many assemblies can fuse and are not liquid per se. This is too strong a statement.
- b. “These [images in TSA treated cells] support a model describing chromosomes as a hydrogel.”
- c. Without explaining their model or showing enough evidence, it is presumed to be correct: “We reasoned in accordance with principles from polymer chemistry, that the hydrogel material of mitotic chromosomes that resists...”
- d. Based on FRAP data, the authors write: “Thus deacetylation is a major factor in establishing an immiscible chromatin phase in mitotic cells”

Figure 3

1) While the data presented in Figure 3 is interesting, it only considers one possibility for excluding microtubules from chromosomes and ends. Is this a general principle in excluding all microtubule-

associated factors? For example- what about microtubule severing enzymes?

2) When reasoning that charge, and not pore size, is the factor which prevents perforation of microtubules, only tubulin dimer-sized macromolecules are considered. On this scale, charge dominates, however one can imagine that the physical force of a polymerizing microtubule may overcome this repulsion. To address this, measure exclusion from the chromatin region (+/- TSA) for a range of macromolecular sizes. This can help estimate a pore size and establish a regime where physical size is also important (relevant for a pre-established microtubule, polymerizing from the cytosolic tubulin pool).

Figure 4

1) This experiment seems tangential from the manuscript's main point and is problematic for a few reasons:

a. The relevant conclusion of this experiment is that the surface tension of the chromatin immiscible phase can resist microtubule perforation after being pushed by microtubules. This is not consistent with the model proposed previously (that microtubules are excluded from chromatin due to charge-mediated exclusion of tubulin dimers). Instead, the reader must now have in mind a model wherein a physical barrier prevents perforation of existing microtubules that polymerize from a cytosolic pool of tubulin. This is confusing: is the proposition that a 'surface' prevents penetration of existing microtubules OR that charge mediated macromolecular exclusion of tubulin dimers controls microtubule exclusion? Which effect is dominant?

b. Is the surface tension of this immiscible phase boundary (AluI-digested chromosomes) equivalent to that of undigested chromosomes? This should be addressed, otherwise it is difficult to interpret these results.

General comments:

1) It is not clear how to understand microtubule perforation when chromatin is 'dissolved' upon hyperacetylation. What defines the chromatin boundary? How should the reader conceptualize perforation into a liquid like state?

2) How does TSA treatment or p300 overexpression affect the levels of the barrier-to-autointegration factor (BAF)? Some of these authors previously showed that BAF cross bridges mitotic chromosomes (Samwer et al., 2017) to form a mechanically stiff layer at the surface of mitotic chromosomes in anaphase. Is this surface still forming in anaphase to reduce the effects of hyperacetylated metaphase chromosomes? What is BAF's response to hyperacetylation?

Referee #2 (Remarks to the Author):

The physical principle on how long chromatin fibers are organized into mitotic chromosomes remains unclear and an important issue in biophysics as well as cell biology.

To approach this issue, Schneider et al. manipulated the histone acetylation state of mitotic chromosomes in living cells. The authors first found that hyperacetylated mitotic chromosomes decondensed drastically with condensin-depletion. They also demonstrated hypoacetylation-induced chromatin compaction prevented microtubule perforation. Interestingly, AluI-digested mitotic chromatin formed liquid-like droplet bodies (chromatin condensates) in a condensin-independent manner, and these bodies were dissolved by histone hyperacetylation. Furthermore,

the chromatin condensates excluded negatively charged macromolecules, including tubulins, and were pushed to the cell periphery by microtubules, suggesting that histone deacetylation during mitosis allows mitotic chromosomes to resist perforation by microtubules.

Overall, the presented data is very impressive. The paper would make a big impact on the cell biology and biophysics fields. For publication in Nature, there are several points to be addressed. My specific comments are the following:

Major points:

1) It is known that Ki-67, which is a positively charged large molecule and locates at the chromosome periphery, is involved in mitotic chromosome formation (e.g., PMID: 29487178, 27362226, 27610954, and 24867636). It would be critical to show how Ki-67 behavior and function are affected upon the authors' manipulations: TSA-, AluI-, and AluI/TSA-treatments in the presence or absence of condensin.

2) "chromatin solubility," "insoluble chromatin phase," and "mitotic chromatin is insoluble." I am not so sure whether these phrases and sentences often used in the text are appropriate because dense mitotic chromosomes are readily accessible to diffusing proteins in living cells (e.g., PMID: 23246002, 15623580). Condensin should also be making loops during the formation process of mitotic chromosomes.

3) The authors nicely demonstrated that condensin is not a key determinant of mitotic chromatin condensation or compaction. It would be nice to stress this point.

4) Fig. 2a. I wonder where condensin is in the AluI-treated chromatin bodies and AluI/TSA treated ones, although condensin-depleted chromatin forms similar bodies upon AluI-treatment. Is condensin associated with the bodies or excluded from them?

5) The AluI-treated chromatin bodies look like apoptotic chromatin, which is a highly condensed structure with linker DNA digestion. It would be nice to show or discuss how they are similar and different. Apoptotic chromatin and mitotic chromosomes seem to have distinct properties (e.g., PMID: 23246002).

6) TSA concentration used (5 μ M) is pretty high. For instance, \sim 0.33 μ M in Ref. 10 and 0.33 -0.66 μ M in other papers (PMID: 16317046; PMID: 28712725). To minimize possible indirect effects, lower concentrations of TSA should also be examined.

Minor points:

1) Double treatments of condensin-depletion and TSA changed chromosome morphology drastically. One of the features of mitotic chromosomes is the phosphorylation of Ser-10 in histone H3. The authors may want to see the H3Ser10P in the treated chromosomes.

2) Fig. 1a. I found there are cytoplasmic Hoechst signals (foci). What are they? I am just afraid of mycoplasma contamination or something, which could affect all the results.

3) "since entanglements occurring due to growth of microtubules..." This sentence is not clear to me, and should be rephrased better.

4) The method descriptions for some critical experiments are insufficient. For instance, how did the authors deplete SMC4 in mitotic cells with AID2 system? In G2 synchronized cells or asynchronous ones? How much Alu1 was injected into a mitotic cell? What is 1 volume of Alu1-stock?

Referee #3 (Remarks to the Author):

During cell division, the eukaryotic chromatin condenses to form metaphase chromosomes that are precisely organized in the central plane of the dividing cell by a balance of pulling and pushing forces applied by microtubules of the mitotic spindle. The chromatin inside condensed chromosomes does not have a stiff fiber-like higher-order structure but rather forms flexible loops organized by rings of condensin protein and behaves as a condensin-dependent hydrogel under external pulling. Recently, the liquid-liquid phase separation (LLPS) hypothesis was put forward as a mechanism for chromatin condensation in repressed heterochromatin and metaphase chromosomes. Still, the liquid or solid state of the chromatin condensates is disputable as the liquid state was previously shown to depend on DNA fragmentation and specific experimental conditions in vitro. In relation to metaphase chromosomes, how such a fluid (or just soft) chromosomal structure could withstand the pushing and pulling forces from the mitotic spindle without chromosomal breaks and mis-segregations remains a fundamental open question.

In this work, the authors combine a conditional condensin knockout technique with a histone deacetylase inhibitor-induced histone hyperacetylation to study whether these two factors play any role in resisting pushing force and penetration of the chromosome bodies by the mitotic spindle in living metaphase cells. They also designed an unconventional and elegant approach to study chromatin condensates and their pushing by microtubules in situ to draw parallels between the whole chromosomes and droplet-like chromatin condensates and thus investigate the mechanism of chromosome resistance to pushing force. The resulting manuscript presents several exciting and seminal findings: i. Condensin protein is shown to be dispensable for withstanding the pushing force applied by spindle microtubules on metaphase chromosomes in living cells while histone deacetylation plays a crucial role in this process – a truly remarkable finding provided that the prevailing view is that condensed chromosomes can be assembled and supported by condensin without significant contribution from the histones. ii. An increased histone acetylation can disrupt chromatin condensates in mitotic cytoplasm under the same physiological conditions as those that are sufficient to dissolve chromatin droplets thus revealing profound mechanistic parallels between the two events. iii DNA integrity is not required for withstanding the pushing force of microtubules in a striking contrast with previous chromosome pulling experiments. iv. By showing that chromatin condensates acquire a surprising property to resist penetration by microtubules as well as by modified negatively charged DsRed protein the work suggests that the chromatin droplets have a negatively charged outer shell that forms a distinct physical boundary on the surface of the condensates. v. Spindle microtubules are likely to apply the pushing force directly on the surface of chromatin condensates rather than act through chromokinesin proteins. This finding implies that the

condensed chromatin boundary is strong enough to relay the pushing force of polymerizing microtubules into concerted mobility of all nucleosome arrays within the chromatin condensate.

Overall, this manuscript is based on an innovative experimental design, is technically sound, clearly written, and it puts forward a new fundamental mechanism underlying chromosome integrity and mobility during cell division. This work will inspire further theoretical and experimental studies on the physical properties of chromatin condensates and their boundaries. For biomedical applications, such as gene editing and therapy, the results of this work are likely to aid in designing new molecular tools for targeting metaphase chromosomes and chromatin condensates. Still, I have some concerns that need to be experimentally addressed to ensure its publication in Nature:

1. This work uses HDAC inhibition by 5 μ M TSA to observe chromosome transitions caused by histone hyperacetylation in living cells. Such treatment may have many pleiotropic and deleterious effects. It had been shown that just 20 nM can cause growth inhibition and apoptosis in HeLa cells (see e.g. You and Park, PMID: 23165748) and thus cause DNA fragmentation. In its turn, DNA fragmentation may significantly change the properties of the chromosome as it appears from the Alu experiments. Therefore, for live-cell imaging experiments the control, TSA-treated and p300 cells should be examined for the cell viability, and absence of DNA breaks in situ (e.g. TUNEL assay).

2. The immunofluorescence experiments (extended data 2 and 3) are not sufficient to convince the reader that the TSA treatment can effectively change the histone charge for altering chromatin condensation. In order to understand whether histone acetylation could change the physical properties and dissolve chromatin condensation directly, by changing the histone charge, the authors should demonstrate efficiency of TSA and p300 treatments by direct measurement of histone charges such as AUT gels (see Shechter et al., 2007 PMID: 17545981) or mass-spectrometry.

3. Page 2, Fig. 1 b, c and elsewhere: This work intermittently uses different statistical tests to show absence or presence of significant differences between pairs of data sets in the same experiment (e.g. t-test, Mann-Whitney test, Kolmogorov-Smirnov test). This depends on whether the data have normal or non-normal distributions and similar or different variances. Since it is hard for the reader to evaluate the data normality and variance, the use of different statistical tests leaves an impression that the t-test, for example, is applied arbitrary when the dataset pairs are expected to be similar. I suggest that at least one statistical test (in addition to the tests optimal for each given type of datasets) should be consistently applied to all datasets as a universal discriminator between significant and non-significant differences.

4. Page 13-14 (methods) describes CLEM – correlative light and electron microscopy. However, no figure shows any H2B fluorescence correlated with EM. Fig 1. D appears to show segmentation based on electron density, not on H2B fluorescence. The parallel H2B fluorescence images should be included to show that the featured EM densities correspond to the histone fluorescence in chromosomes.

Additional minor points:

Page 1, right, 6th line from the top: “stained DNA with Hoechst to...” Hoechst is a general brand

name, please indicate which type was used, e.g. Hoechst 33342

Page 15, right, 4th line from the top: here “DNA channel” refers to Hoechst 33342 staining and in the next paragraph “DNA density” in a similar experiment seems to be derived from H2B fluorescence. The nature of fluorescence signals should be indicated in all cases.

Author Rebuttals to Initial Comments:

Nature manuscript 2021-07-10665 by Schneider et al., response to the referees' comments

We thank all three reviewers for their thoughtful and constructive suggestions on how to improve our manuscript. We addressed all concerns by new experiments and by revising the text, as explained point-by-point below. The new data corroborate our conclusions and provide additional insights into the organization of mitotic chromosomes.

The most important points of our revision are:

- **A more extensive validation that the mitotic phenotypes resulting from TSA treatment are due to hyperacetylation and not unspecific toxic side-effects of the compound.** We show that TSA induces full chromatin decompaction already at a 10-fold lower concentration compared to that used in the original manuscript. We further show that the mitotic phenotypes can be fully reversed by removing TSA and that the short TSA treatments used throughout our study do not induce apoptosis or DNA damage. We used p300 overexpression as a complementary approach to show that hyperacetylation solubilizes AluI-induced chromatin fragments in mitotic cytoplasm. Together, these data corroborate the specific role of acetylation in the regulation of chromatin phase separation.
- **An analysis of chromosome-spindle interactions by time-lapse microscopy.** By simultaneously visualizing chromatin, kinetochores, and microtubules in live cells entering mitosis, we show that condensin-depleted chromatin is rapidly moved away from the spindle pole, while kinetochores are detached from the main mass of chromatin as they move towards the pole. In TSA-treated cells depleted of condensin, the spindle asters do not displace chromatin but rather invade into a decompacted mass of chromatin. These experiments corroborate our model where an acetylation-regulated phase separation provides chromatin with the mechanical resistance to be pushed by polar ejection forces, instead of being perforated by spindle microtubules.
- **A more detailed investigation of the phase boundary between mitotic chromatin and cytoplasm by studying the protein Ki-67.** Ki-67 is a component of the mitotic chromosome periphery that has a C-terminal domain binding to chromatin and an N-terminal domain that is excluded from chromatin. Given this amphiphilic molecular organization, our hydrogel phase transition model predicts that Ki-67's targeting to the boundary between chromatin and cytoplasm should not be perturbed by condensin depletion or AluI-mediated chromatin fragmentation, which we indeed observed to be the case. Our model further predicts that hyperacetylation should lead to a more diffuse localization of Ki-67, as it impairs the formation of a phase boundary between mitotic chromatin and cytoplasm, which we indeed observed to be the case. These findings corroborate that acetylation-regulated chromatin solubility contributes to the formation of a sharp phase boundary towards cytoplasm.
- **An extended electron tomography analysis of microtubule perforation into mitotic chromosomes.** We increased the sample size to 10 electron tomograms per condition and performed additional quantifications to corroborate our conclusion that microtubules almost never perforate into unperturbed mitotic chromosomes, whereas they frequently grow deeply into hyperacetylated chromosomes of TSA-treated cells.
- **An investigation of how molecular size affects the exclusion of macromolecules from mitotic chromatin.** New microinjection experiments show that with increasing molecular size the exclusion of negatively charged dextrans from mitotic chromosomes is more efficient. Furthermore, TSA treatment diminishes the exclusion of large dextrans from mitotic chromatin, even for dextrans that have a gyration radius of 19 nm. These experiments show that electrical charge and molecular size

both contribute to the exclusion from mitotic chromosomes and that the decompaction induced by TSA enables access by soluble macromolecules in the size range of the microtubule diameter.

- **An investigation of how two microtubule-associated proteins localize relative to mitotic chromatin.** We show that the microtubule severing enzyme katanin and the microtubule plus-tip binding protein EB3, which both carry negative charge at cytoplasmic pH, are excluded from mitotic chromatin in a TSA-dependent manner. These observations corroborate our model of acetylation-dependent exclusion of negatively charged macromolecules from mitotic chromosomes.

Referee #1 (Remarks to the Author):

Schneider et al. examine how histone deacetylation can control the solubility of chromosomes to form an immiscible mitotic chromatin phase. The authors suggest that this phase acts as a physical barrier to microtubule perforation, thus protecting the genome during eukaryotic cell division. This work focuses on an interesting, fundamental question. However, there are critical discrepancies between the data and claims and (2) a key finding has already been published by the same group of authors, putting the novelty of this work into question (reference 13, Gibson et al). For these reasons, I cannot recommend it for publication in Nature. Below, I elaborate on these two points.

We thank the reviewer for the constructive suggestions on how to improve our manuscript. The specific recommendations for further validation and extension of our study are very helpful and have led us to perform numerous experiments that corroborated the key conclusions of our manuscript, as explained below. However, the criticism about novelty is based on a misconception, as we clearly referred to the previous observation of liquid-liquid phase separation of purified nucleosome arrays and its regulation by acetylation, as reported in Gibson et al., 2019, and we do not claim novelty on this point. The Gibson et al. study was indeed a key rationale for our present study, as it raised the question of whether and under which conditions endogenous chromatin might undergo a phase transition, how such chromatin phase transition might affect the material properties of chromosomes, and what might be the functional relevance. These questions might not have been obvious for readers outside the chromosome field, as the introductory and discussion sections of our original manuscript were very concise. To clarify the novelty and relevance of our work, we revised the respective sections of the manuscript.

Our study provides novel insights into the following fundamental questions:

1) Under which conditions does endogenous chromatin undergo phase separation?

The work by Gibson et al., 2019, shows that purified recombinant nucleosome arrays form immiscible droplets in buffer solutions containing physiological salt concentrations and that the purified nucleosome arrays enrich in chromatin-dense regions when injected into interphase nuclei. This manuscript further shows that acetylation of purified recombinant nucleosome arrays prevents their phase separation in vitro and reduces their accumulation in chromatin-dense regions of interphase nuclei. It has remained unclear if and under which conditions endogenous chromatin might undergo a phase transition, and what might be the physiological function of such a phase transition.

Endogenous chromatin contains thousands of different proteins and a multitude of different posttranslational modifications, which might have profound effects on phase separation properties. Our study is the first to show that endogenous chromatin undergoes a global phase transition upon mitotic entry – from a soluble state in interphase nuclei to an insoluble state in mitosis. Our work

further shows that this phase transition critically depends on a single type of posttranslational modification – acetylation – despite the extensive change in protein composition and of posttranslational modification patterns observed on endogenous chromatin during mitotic entry. Our discoveries have become possible through an innovative live-cell chromosome fragmentation assay, and the study of how purified nucleosome arrays interact with endogenous mitotic chromatin.

2) How does the transition from a soluble to an insoluble chromatin state affect the material properties of chromosomes?

In contrast to the short chromatin fragments that undergo liquid-liquid phase separation *in vitro* (Gibson et al., 2019), endogenous human chromatin fibers are up to ~100,000-fold longer and folded into consecutive loops by condensin complexes. Mitotic chromosomes have a defined shape and respond elastically to mechanical forces, which is apparently inconsistent with a liquid state. How putative phase transitions of endogenous chromatin affect the material properties of mitotic chromosomes has been unknown.

Chromosomes have been described as hydrogels, in which a flexible chromatin fiber is cross-linked by condensin and expanded throughout its volume by an aqueous liquid component. By elucidating acetylation as a key regulator of chromatin solubility, our study provides a molecular explanation of how such chromatin hydrogels collapse into compact bodies with a sharp boundary in mitosis. Our study thus conceptualizes how DNA looping and chromatin phase separation cooperate in the formation of mitotic chromosomes.

Our acetylation-regulated hydrogel model explains why mitotic chromatin can compact to a full extent in the absence of condensin (then forming a polymer melt) and why chromatin hyperacetylation alone has only a relatively mild effect on chromatin decompaction, as the cross-links provided by condensin constrain dispersion of the chromatin fiber (then forming a swollen hydrogel). Only in the absence of condensin and upon hyperacetylation, the soluble chromatin fiber disperses completely in cytoplasm. In our revised manuscript, we provide a more detailed explanation about these concepts.

3) What is the physiological relevance of mitotic chromatin phase separation?

The observation that purified chromatin fragments phase separate *in vitro* (Gibson et al., 2019) raised the question of what the physiological relevance of this phenomenon might be in cells. Our study shows that acetylation-regulated chromatin phase separation protects mitotic chromosomes against microtubule perforation. This discovery reveals a previously unrecognized problem of spindle mechanics: if microtubules growing out from the spindle poles in mammalian cells were to grow through the DNA loops formed by condensin, this would result in massive entanglements inconsistent with independent chromosome motility, as required for faithful segregation. Our study thus shows how mitotic chromosomes form discrete bodies with a sharp surface boundary, rather than loose bottlebrush structures with chromatin fiber loops extending into the cytoplasm, with important implications for faithful chromosome segregation.

In summary, our manuscript shows that histone deacetylation during mitosis generates a sharp global reduction in chromatin solubility that converts chromosomes into phase-separated bodies. This immiscible chromatin phase excludes negatively charged macromolecules such as tubulin, thereby restricting the growth of microtubules inside chromosome arms. We demonstrate that the phase boundary of immiscible chromatin creates a barrier to microtubule perforation that is sufficient to provide mechanical resistance to pushing forces independently of condensin, whereas condensin-mediated connection of DNA loops is necessary to prevent shearing as microtubules pull on kinetochores. Importantly, our results explain how it is possible to generate mechanical resistance at the surface of chromosomes while still allowing dynamic DNA looping by condensin inside

chromosomes. The material properties emerging from the mitotic chromatin phase transition thereby protect against entanglements and potential segregation defects caused by microtubule perforation. We are convinced that our manuscript offers a significant conceptual advance at the interface of cell biology and biophysics that will be of interest to a broad readership.

Figure 1

1) It is not clear if compound toxicity is a factor in many of the observations (lagging chromosomes, microtubule perforation). The dose dependent effects of this compound should be reported, and its reversibility should be demonstrated. This will help ensure that the observations reported are due to acetylation state and not to compound toxicity. While overexpressing the histone acetyltransferase p300, the authors observed the chromatin phenotypes observed with the compound. However, their main findings demonstrating increased microtubule perforation and differences in chromatin solubility were not reproduced with this overexpression approach.

We thank the reviewer for pointing out how to further validate the specificity of the mitotic phenotypes induced by TSA. We performed a series of new experiments addressing this concern. First, we investigated the phenotypes of various TSA concentrations, finding that full mitotic chromatin decompaction is induced already by a ten-fold lower concentration of 500 nM (new Extended Data Fig. 4a-d). 500 nM TSA treatment of condensin-depleted cells also resulted in the formation of a spindle within a diffuse mass of chromatin, as observed for cells treated with 5 μ M TSA, corroborating the relevance of deacetylation to protect against microtubule perforation (new Extended Data Fig. 4c, d). We also tested whether the phenotypes induced by TSA can be reverted, by treating cells first with 500 nM TSA for 3 h and then removing TSA for 8 h before studying mitotic phenotypes. After this treatment, mitotic chromosomes compacted to the full extent and had a spindle morphology indistinguishable from untreated control cells (new Extended Data Fig. 4a-d). Thus, TSA treatment is fully reversible within less than one cell cycle.

To further assess the toxicity of TSA, we imaged the Polarity Sensitive Indicator of Viability & Apoptosis (pSIVA; Kim et al., 2010, PMID 19966809). These experiments show that 2.5 h treatment with 5 μ M TSA does not induce apoptosis (Extended Data Fig. 3e, f). We also tested if TSA impairs DNA integrity, by imaging a component of the DNA double-strand break repair machinery, gamma-H2A.X, using immunofluorescence. These experiments showed that 2.5 h treatment with TSA does not elevate gamma-H2A.X levels above those of untreated control cells (new Extended Data Fig. 3g, h). Hence, the TSA treatment that we used to induce full decompaction of mitotic chromatin does not impair DNA integrity or cell viability.

To further test whether mitotic chromatin is solubilized by hyperacetylation and not by unspecific side-effects of TSA, we performed a new experiment in which we injected AluI into cells overexpressing the acetyltransferase p300. The AluI injection resulted in a dispersion of chromatin throughout the cytoplasm, similar to the phenotype resulting from TSA (new Extended Data Fig. 7e, f; compare Fig. 2e, f). Together, these new data corroborate that the mitotic phenotypes induced by TSA are due to hyperacetylation rather than unspecific toxicity of the compound.

2) The data presented in panel 1A does not match the conclusions drawn in the text. For example the authors state: “In condensin-depleted cells, we observed an unstructured mass of compact chromatin forming a plate between the spindle poles, indicating that polar ejection forces efficiently push bulk chromatin towards the spindle center”. Similarly, the authors state that the localization of “loosely connected kinetochores towards the spindle poles confirms that condensin is required for chromosomes

to resist pulling. Thus, condensin depletion allows the mechanisms responsible for the resistance to pushing and pulling forces to be decoupled". As only snapshots of cells are presented, it is incorrect to make conclusions about forces and how their origins can be decoupled.

We agree that the forces exerted by the mitotic spindle on chromosomes are complex and best studied by time-lapse microscopy. We hence performed new experiments in which we imaged live cells entering mitosis to investigate in more detail how chromosomes are moved by the spindle.

We proposed in our original manuscript that bulk chromatin is moved to the spindle center by polar ejection forces because kinetochores were detached and therefore not expected to direct the movement of chromosomes towards the spindle center, while prior work showed that chromosome arms are subject to microtubule-based pushing away from the poles (i.e., polar ejection forces). However, in bipolar spindles it is difficult to distinguish the effect of pole-directed pulling on kinetochores from anti-poleward ejection forces exerted on chromosome arms. For this reason, prior investigations of polar ejection forces often used assays in which the separation of centrosomes is suppressed by inhibitors of kinesin-5 to induce a monopolar spindle geometry (e.g., Barisic et al., 2014, PMID: 25383660; Maiato et al., 2017, PMID: 28218637). We hence decided to use the monopolar spindle assay to study by 3D-confocal time-lapse imaging how condensin depletion and TSA treatment affect the response of chromosomes to spindle forces acting on kinetochores and chromosome arms. These new data are presented in a revised first section of the manuscript:

"The chromatin of condensin-depleted cells might be moved by polar ejection forces. To study how chromosomes are moved by the mitotic spindle, we imaged mitotic entry of HeLa cells stably expressing mCherry-tagged core histone 2B (H2B-mCherry) and EGFP-tagged CENP-A, and visualized microtubules by SiR-tubulin. As it is difficult to distinguish the effect of poleward and anti-poleward forces in a bipolar spindle configuration, we induced a monopolar spindle geometry by inhibiting kinesin-5 using S-trityl-L-cysteine (STLC)²⁶. When condensin-expressing control cells entered mitosis, chromosomes first moved towards the spindle pole shortly after nuclear disassembly and then arranged in a rosette with kinetochores facing towards the pole and chromosome arms facing away from the pole, such that the region surrounding the spindle pole remained free of chromosomes (Extended Data Fig. 2a, d; Supplementary Video 1). This arrangement is consistent with a balance between pole-directed microtubule pulling at kinetochores and polar ejection forces pushing on chromosome arms^{19-21,23,22}. When condensin-depleted cells entered mitosis, chromatin formed a compact mass that moved away from the spindle pole, while kinetochores approached the spindle pole, resulting in the detachment of a large fraction of kinetochores from the bulk mass of chromatin (Extended Data Fig. 2b, d, e; Supplementary Video 2). Thus, condensin-depleted chromatin remains responsive to polar ejection forces, whereas it is not stiff enough to resist the tension generated at kinetochores." [...]

"To investigate more specifically how hyperacetylation affects the response of chromosomes to polar ejection forces, we imaged mitotic entry of live condensin-depleted cells treated with TSA, using STLC to induce a monopolar spindle geometry. The chromatin of these cells remained diffuse and completely decompacted, while the spindle aster assembled and moved into the decondensed chromatin regions. Kinetochores then moved towards the spindle pole, but the bulk mass of chromatin was not displaced towards the cell periphery such that the regions surrounding spindle poles did not clear from chromatin, and kinetochores remained embedded in chromatin (Extended Data Fig. 2c-e; Supplementary Video 3). Thus, deacetylation has an important role in the response of chromatin to polar ejection forces."

Our new data support the conclusion that condensin-depleted chromatin is pushed away from spindle poles by polar ejection forces and that the pushing of chromatin away from the poles depends on deacetylation.

3) Overall, it is unclear to me why the results from Figure 1 are discussed/presented in the context of pushing forces. Many parts of the manuscript (intro, results for Figure 1 and 4) should be re-written to omit the focus on spindle forces. The authors state that we do not understand how microtubule-generated pushing forces affect chromosomes and reason that we already know how condensin protects chromosomes from microtubule-generated pulling. This is problematic because it reduces the 3D chromatin structure to only 2 dimensions, with chromosomes experiencing 2 simple, opposite forces. A strain map across the chromosomes is likely more complicated than a simple pull/push.

The main point of our paper is that chromatin phase separation provides mitotic chromosomes with material properties that resist perforation by spindle microtubules. In this context, we think it is important to refer to prior work demonstrating that microtubules contacting chromosome arms exert pushing forces away from the poles rather than just growing through chromatin (Ault et al., 1991, PMID: 1685159; Rieder and Salmon, 1994, PMID: 8294508; Barisic et al., 2014, PMID: 25383660; Maiato et al., 2017, PMID: 28218637). We realize that the discussion of pushing and pulling forces in our original manuscript might have been too concise and potentially confusing.

We rephrased large parts of the main text based on the original data and the new time-lapse experiments, as explained above. While we appreciate the complexity of spindle forces in three dimensions, we would like to clarify that the terms “pulling” and “pushing” do not imply a reduction to two dimensions, as they refer to orientations relative to the spindle pole in all three spatial dimensions. While the directionality of forces is difficult to dissect in bipolar spindles, the induction of monopolar spindles allows straightforward discrimination of poleward versus anti-poleward forces. We clarify this in the revised main section discussing the data shown in Fig. 1 and the corresponding Extended Data Figures and supplemental videos.

We thank the reviewer for raising these questions, as the new data and textual revisions have helped us clarify how deacetylation protects mitotic chromosomes against microtubule perforation such that they are pushed away from spindle poles.

4) The extent of microtubule perforation into chromosomes is not clear. As only 2 examples are presented in panel D and the quantification (E and F) is minimal, it is not clear if this is a subtle effect. An additional problem is that the 2 panels in fig 1D compare different regions between control (pole-facing side) and treated (in between chromosomes) cells. Equivalent sides should be compared. A thorough classification of this perforation effect should include analyses that determine whether microtubule perforation is region-specific. Specifically, perforation should be mapped across the chromosome periphery. From the current data, one cannot tell if there are a few patches where microtubules penetrate the chromosomes, or if this is a global effect. Overall, judging from the small panels in Figure 1 and the subtle change in anaphase defects (Extended Figure 1F, G) (which may come from the compound alone), I am not convinced that this microtubule perforation phenotype is prominent.

We thank the reviewer for suggesting how to improve the electron microscopy experiments. We quantified 12 additional electron tomograms and show four additional examples in the revised manuscript. We also highlight microtubule perforation sites at the chromosome surface (new Extended Data Fig. 6a-b; revised Fig. 1d-f). We further separately analyzed prometaphase cells, where most chromosome surfaces orient towards neighboring chromosomes, and metaphase cells, where most chromosome surfaces orient towards spindle poles, finding that TSA induces frequent

microtubule perforation at both mitotic stages (new Extended Data Fig. 6c). Overall, the new data corroborate that microtubules very rarely perforate unperturbed mitotic chromosomes, but frequently perforate hyperacetylated chromosomes, and that perforation occurs at many sites, both at pole-facing regions as well as at regions in between chromosomes.

We would like to note that the anaphase defect is not subtle. In TSA treated cells anaphase is substantially delayed and a large fraction of cells do not enter anaphase even hours after mitotic entry. At 45 min after mitotic entry, only 15% of TSA-treated cells entered anaphase, compared to 90% in control cells (see Extended Data Fig. 6e). Moreover, for those TSA-treated cells that eventually enter anaphase, the incidence of lagging chromosomes is more than 3-fold higher than in control cells (Extended Data Fig. 6f-h).

5) Extended figure 1, related to this data needs improvement. In particular:

a. S1D (pericentrin/CenpA localization)- quantify this effect and include statistics. How many cells were used to show this?

We thank the reviewer for pointing out that this experiment was lacking quantification. We now indicate the number of analyzed cells in the revised legend ($n=40$ cells for control $n=59$ for Δ Condensin, $n=35$ for Δ Condensin+TSA, $n=40$ for TSA). We also quantified the fraction of kinetochores detached from chromatin per cell (new Extended Data Fig. 1g). The new data show that a large fraction of kinetochores is detached in condensin-depleted cells.

b. S1G- show the distribution of cells, as the mean is not very informative if the distribution is skewed. Also, show more examples of chromosomes in anaphase.

In the revised manuscript, we show three examples for TSA-treated anaphase cells with lagging chromosomes (revised Extended Data Fig. 6f) and quantified the number of lagging chromosomes per cell, presented in a histogram (new Extended Data Fig. 6h).

Figure 2

1) The authors investigate how “histone acetylation affects the material properties of chromatin” and conclude that “..acetylation regulates chromatin solubility”. These results were presented by many of these authors in Gibson et al, where acetylation was shown to modulate phase separation. While the results from Gibson et al focused on in vitro condensates, data from cells was also presented, diminishing the novelty of these findings. The authors cite their previous work, however, they do so without explicitly stating they previously showed that the chromatin acetylation state affects its phase separation. For example, Gibson et al. was cited after the following sentences:

a. “Assembly of mitotic chromosomes involves DNA looping by condensins and chromatin compaction by global histone deacetylation.”

b. “Nucleosomal interactions are thought to increase when histones are deacetylated during mitotic entry, contributing to global chromatin compaction”

c. “...we used synthetic nucleosome arrays”

The critique about an unjustified claim of novelty is based on a misconception, as we clearly referred to the previous work showing that acetylation affects chromatin phase separation in vitro: on p. 5 of the original manuscript, we wrote “*In vitro, the unmodified nucleosome arrays form liquid condensates under physiological salt concentrations, in contrast to the acetylated nucleosome arrays (Extended Data Fig. 4e, f)*”¹³. Ref. 13 is the paper by Gibson et al., 2019, and the data shown in Extended Data Fig. 4e, f of the original manuscript is merely a validation that the nucleosome arrays we use for microinjection experiments are functional.

To avoid any misunderstanding, we moved the citation of the paper before referring to our validation experiment: “*In vitro, the unmodified nucleosome arrays form liquid condensates under physiological salt concentrations, in contrast to the acetylated nucleosome arrays*”¹³ (Extended Data Fig. 4e, f).” The new finding with the nucleosome arrays in our present paper is about their interaction with **mitotic** chromatin: the unmodified arrays partition into mitotic chromatin, whereas the pre-acetylated arrays distribute diffusely throughout chromatin and cytoplasm (Fig. 2m, n).

We would like to note that the Gibson et al., 2019 study only investigated the phase separation behavior of recombinant nucleosome arrays – either in buffer solutions or after injection into cells. Endogenous chromatin is composed of thousands of different proteins and it contains many different types of posttranslational modification, many of which change during the transition from interphase to mitosis (Ohta et al., 2010, PMID: 20813266; Ginno et al., 2018, PMID: 30279501; Zhiteneva et al., 2017, PMID: 28903997). Furthermore, mitotic chromatin is surrounded by cytoplasm rather than buffer, which might result in very different phase separation behavior compared to simple salt solutions. Because of these differences, the phase separation properties of endogenous chromatin cannot be directly inferred from observations of recombinant nucleosome arrays as used in the previous study by Gibson et al., 2019.

The phase separation properties of endogenous chromatin have remained unclear because of the lack of assays to measure chromatin solubility inside cells. We have overcome this limitation by developing several new methods to measure and perturb chromatin solubility. Our AluI-mediated in situ chromatin fragmentation assay showed that endogenous mitotic chromatin is insoluble in cytoplasm and that its immiscibility depends on deacetylation. Our perturbations of condensin and acetylation further showed that – unlike purified nucleosome array condensates – endogenous mitotic chromatin forms an immiscible, collapsed hydrogel rather than a liquid condensate. These discoveries answer long-standing questions about the material properties of mitotic chromosomes.

These insights will be of broad interest to cell biologists and soft matter physicists, in light of the rapidly evolving field of phase separation and the controversies surrounding the relationship between in vitro observations and behavior in the cellular environment for many types of biological condensates. To clarify the novelty and impact of our work, we revised large parts of the section about chromatin phase separation and provide as a rationale for our investigation:

“To elucidate the mechanism underlying microtubule exclusion from mitotic chromosomes, we investigated how acetylation affects the material properties of chromatin. Recent work has demonstrated that phase separated biomolecular condensates can form highly dense structures that exert and resist forces³². Moreover, we found that purified nucleosome arrays condense into liquid droplets in physiological salt solutions, and these condensates dissolve upon acetylation¹³, supporting the idea that mitotic chromatin might form an immiscible phase. However, endogenous chromatin contains thousands of different proteins³³ and is subject to various posttranslational modifications besides acetylation^{12,27}, with unknown effects on phase separation. Under which conditions endogenous chromatin might undergo a phase transition, how such chromatin phase transition might affect the material properties of chromosomes, and what might be the functional relevance is not known.”

2) A hydrogel model is proposed to support their observations, but the basis of this model is unexplained. The definition of a hydrogel is not explained. Why is this model preferred over others? For example, why was a hydrogel proposed instead of a simple gel? It is also not clear if this is the first presentation of such a model describing chromosomes and if the data indeed supports this model.

We realize that this part of the manuscript might have been difficult to understand for readers outside the chromosome field and revised this section for clarification. A gel is defined as a semi-solid consisting of a cross-linked polymer network that is expanded throughout its volume by a liquid component. A hydrogel is defined as a gel in which the liquid component is water. As cytoplasm is an aqueous environment, we prefer to use the term hydrogel rather than the more generic term gel.

The idea that mitotic chromatin might be a hydrogel has been proposed more than twenty years ago, based on theoretical considerations and the observation that purified chromosomes reversibly swell and compact upon changing salt concentrations *in vitro* (Marko and Siggia, 1997; PMID: 9362064; Poirier and Marko, 2002, PMID: 12438695; Poirier et al., 2002, PMID: 11948697; cited in our original manuscript when referring to the concept of hydrogels). The discovery that condensin organizes chromatin fibers into consecutive loops (Naumova et al., 2013, PMID: 24200812) and that condensin is required to confer resistance to pulling forces (Gerlich et al., 2006, PMID: 16488867; Houlard et al., 2015, PMID: 25961503; Sun et al., 2018, PMID: 30143891) provides a molecular basis for cross-linking in the chromatin fiber constituting the polymer component of the hydrogel. However, whether and under which conditions the chromatin network might undergo a phase transition inside cells and how this might be regulated at the molecular level has remained unclear.

We discuss our discovery of an acetylation-regulated chromatin phase transition in the context of the hydrogel model, as it provides a unified conceptual framework explaining several of key observations:

- hyperacetylation induces mitotic chromosome swelling rather than complete dispersion of the chromatin fiber; only when condensin is depleted chromatin fully decompacts and diffusely fills the cytoplasm
- condensin depletion or chromatin fragmentation do not affect the degree of chromatin compaction in the deacetylated state

We discuss these considerations in our revised manuscript to clarify how our data support an acetylation-regulated hydrogel phase transition model for mitotic chromosome compaction.

3) Conclusions are drawn about the material property of chromosomes without enough evidence. A few examples:

a. “Shortly after microinjection of AluI, chromosomes lost their elongated shape, forming round condensates that fused to one another, indicating a liquid state.” Many assemblies can fuse and are not liquid per se. This is too strong a statement.

We would like to note that in addition to the round morphology and coalescence, the liquid-like state of chromatin condensates formed after AluI injection is demonstrated by the rapid fluorescence recovery of H2B-mCherry after photobleaching. However, the FRAP experiment is only presented in the following paragraph. We, therefore, addressed this concern by revising this sentence to: “*Shortly after microinjection of AluI, chromosomes lost their elongated shape, forming round condensates that fused to one another, consistent with a liquid-like state.*”

We then substantiate this conclusion after presenting the FRAP data in the following paragraph: *“Thus, mitotic chromatin is insoluble in cytoplasm, and when the long-range constraints of the fiber network are eliminated, the short-range dynamics manifest in liquid-like behavior.”*

b. “These [images in TSA treated cells] support a model describing chromosomes as a hydrogel.”

We rewrote the section about the hydrogel model, such that the concepts and the interpretation are now introduced and discussed more clearly.

c. Without explaining their model or showing enough evidence, it is presumed to be correct: “We reasoned in accordance with principles from polymer chemistry, that the hydrogel material of mitotic chromosomes that resists...”

This critique is based on a misunderstanding, as the full sentence in our original manuscript was formulated as a hypothesis, not a conclusion: *“We reasoned, in accordance with principles from polymer chemistry³⁶, that the hydrogel material of mitotic chromosomes that resists microtubule perforation **might** arise as a consequence of intrinsic chromatin phase separation at the very long length scale of human chromosomes.”* This sentence was intended to provide a rationale for studying potential phase separation properties of endogenous chromatin inside cells. We rephrased the section about chromatin solubility and the hydrogel model for a more detailed explanation of prior work and our new findings, as explained above.

d. Based on FRAP data, the authors write: “Thus deacylation is a major factor in establishing an immiscible chromatin phase in mitotic cells”

This point also seems to be based on a misunderstanding, as the cited sentence was not related to FRAP data. The quoted sentence was a conclusion derived in a paragraph presenting the localization of AluI-induced chromatin fragments in the presence or absence of TSA (p. 3, right column, 2nd paragraph).

Regarding the FRAP experiments, we would like to clarify that they reveal very limited chromatin mobility in unperturbed chromosomes and rapid mobility in chromatin condensates after AluI injection, consistent with gel material properties of native chromatin and an insoluble liquid state of fragmented chromatin. As the interpretation of FRAP experiments might have been difficult to understand in the original manuscript, we rephrased this part:

“To assess the mobility of chromatin, we measured fluorescence recovery after photobleaching H2B-mCherry. Native mitotic chromosomes recovered very little H2B-mCherry fluorescence after photobleaching, consistent with constrained mobility within a large polymer network. After AluI digestion, however, H2B-mCherry recovered rapidly and completely from photobleaching (Fig. 2c, d), consistent with a liquid state. Imaging Halo-tagged SMC4 further showed that condensin did not form axial structures inside the chromatin condensates and instead evenly distributed throughout the cell, validating efficient chromosome fragmentation by AluI (Extended Data Fig. 7c, d). Thus, mitotic chromatin is insoluble in cytoplasm, and when the long-range constraints of the fiber network are eliminated, the short-range dynamics manifest in liquid-like behavior.”

Figure 3

1) While the data presented in Figure 3 is interesting, it only considers one possibility for excluding microtubules from chromosomes and ends. Is this a general principle in excluding all microtubule-associated factors? For example- what about microtubule severing enzymes?

We propose that the acetylation-dependent exclusion of negatively charged macromolecules from mitotic chromatin is a general principle that applies not only to tubulin but other cytoplasmic proteins as well. Our *in vitro* experiments with purified chromatin and tubulin, however, show that microtubule-associated factors are not necessary for the exclusion of microtubules from chromatin condensates (Fig. 3e, f). Nevertheless, we agree that it is interesting to determine how microtubule-associated factors partition relative to mitotic chromosomes and cytoplasm, and how this is affected by acetylation.

We hence investigated the localization of a microtubule plus-tip protein (EB3) and a subunit of the microtubule-severing ATPase Katanin (p80), which both have a predicted negative charge at cytoplasmic pH. Live imaging showed that both proteins are excluded from mitotic chromosomes and that the exclusion was diminished after TSA treatment (new Extended Data Fig. 9e-h). These observations indicate that our model of acetylation-dependent exclusion of negatively charged macromolecules from chromatin is also valid for two important microtubule-associated factors.

2) When reasoning that charge, and not pore size, is the factor which prevents perforation of microtubules, only tubulin dimer-sized macromolecules are considered. On this scale, charge dominates, however one can imagine that the physical force of a polymerizing microtubule may overcome this repulsion. To address this, measure exclusion from the chromatin region (+/- TSA) for a range of macromolecular sizes. This can help estimate a pore size and establish a regime where physical size is also important (relevant for a pre-established microtubule, polymerizing from the cytosolic tubulin pool).

This comment is based on a misunderstanding, as we did not claim that pore size is not relevant for the exclusion of microtubules from mitotic chromatin. We only stated that pore size is not limiting at a size range of tubulin dimers, as demonstrated by the analysis of DsRed partitioning. On p. 5 of the original manuscript we stated: “*Thus, macromolecules in the size range of tubulin are not generally excluded from mitotic chromatin.*” In the following paragraph, we investigate pairs of similarly sized macromolecules differing in their charge and concluded: “*Thus, electrical charge is a key determinant of macromolecular access to mitotic chromatin.*” We did not state anywhere in our manuscript that macromolecular size is irrelevant for the exclusion of microtubules from mitotic chromosomes.

We agree with the reviewer that molecular size is an interesting parameter worth further investigation. We hence performed microinjection experiments using larger dextrans (20 kDa and 70 kDa molecular weight, with gyration radii of 10 nm and 19 nm, respectively). The new data show that with increasing molecular size the dextrans are excluded from chromatin more efficiently; furthermore, TSA treatment diminished the exclusion of 20 kDa as well as 70 kDa dextrans from mitotic chromatin (new Extended Data Fig.9i-l). Together with the data of our original manuscript, the new data show that while macromolecules of the size of tubulin dimers are already excluded quite efficiently from mitotic chromatin, the exclusion is more pronounced for larger molecules. Importantly, even macromolecules with diameters close to the width of microtubules are only excluded from mitotic chromatin if it is not hyperacetylated. We thank the reviewer for suggesting to investigate size effects, as the new data reveal interesting material properties of mitotic chromatin.

Figure 4

1) This experiment seems tangential from the manuscript's main point and is problematic for a few reasons:

a. The relevant conclusion of this experiment is that the surface tension of the chromatin immiscible phase can resist microtubule perforation after being pushed by microtubules. This is not consistent with the model proposed previously (that microtubules are excluded from chromatin due to charge-mediated exclusion of tubulin dimers). Instead, the reader must now have in mind a model wherein a physical barrier prevents perforation of existing microtubules that polymerize from a cytosolic pool of tubulin. This is confusing: is the proposition that a 'surface' prevents penetration of existing microtubules OR that charge mediated macromolecular exclusion of tubulin dimers controls microtubule exclusion? Which effect is dominant?

Our *in vitro* and cell-based data show that chromatin phase separation counteracts microtubule perforation by two mechanisms: by limiting the soluble pool of tubulin dimers inside chromatin and by forming a surface that resists perforation by microtubules polymerizing in the surroundings. Both effects are not mutually exclusive and arise through the same process: transition from a soluble chromatin state to an insoluble state. It is therefore not possible to determine which effect is dominant.

The specific observations that support our model are: condensates of nucleosome arrays or mitotic chromosomes show similar patterns of partitioning for molecules based on charge and size, with decreased concentration of tubulin dimers within them, and intrinsic chromatin condensates composed only of nucleosome arrays are sufficient to prevent invasion by microtubules *in vitro*.

We realize that these concepts are new in the cell division field and might have been difficult to understand because of the very concise presentation in our original manuscript. To make the text more accessible for readers outside the chromatin and phase separation fields, we extended and rephrased large parts of the manuscript to provide more explanation about the hydrogel model and the mechanical properties arising from the transition of a soluble to an insoluble chromatin state.

b. Is the surface tension of this immiscible phase boundary (AluI-digested chromosomes) equivalent to that of undigested chromosomes? This should be addressed, otherwise it is difficult to interpret these results.

This is an interesting question, but there are no techniques to measure the surface tension of chromosomes inside cells. To address how the surface of intact mitotic chromosomes relates to the surface formed on AluI-digested chromatin condensates, we performed a series of new experiments in which we used the protein Ki-67, a component of the mitotic chromosome periphery, as a probe for the boundary between chromatin and cytoplasm.

Prior work had shown that Ki-67's C-terminal domain is attracted to mitotic chromatin, whereas its N-terminal domain is excluded from chromatin such that it localizes in the surrounding cytoplasm (Cuylen et al., *Nature*, 2016, PMID: 27362226). These observations suggest that Ki-67 targets to the chromosome surface through its amphiphilic organization, in line with our model of a phase boundary between immiscible mitotic chromatin and cytoplasm. This model predicts that Ki-67 should also target to the surface of AluI-digested chromatin droplets, even though they lack the characteristic 3D organization of chromatin fibers into consecutive loops forming a thread. To test this hypothesis, we imaged live mitotic cells expressing Ki-67-EGFP and found that Ki-67 indeed

enriched at the surface of AluI-digested chromatin condensates – as much as on the surface of intact chromosomes (new Fig. 2o-q).

Our model further predicts that hyperacetylation by TSA should abrogate the formation of a phase boundary between chromatin and cytoplasm so that Ki-67 is expected to localize more diffusely around the periphery of chromatin regions. Indeed, imaging of Ki-67-EGFP in mitotic cells treated with TSA led to a re-localization of Ki-67 such that no sharp surface accumulation was detected on chromosomes anymore (new Fig. 2o-q). The new data corroborate the acetylation-regulated phase separation model of mitotic chromatin and further indicate that the phase boundary of intact chromosomes resembles that of AluI-digested chromatin condensates.

Our experiments with purified chromatin and tubulin demonstrate that the surface tension on liquid chromatin condensates provides sufficient resistance to withstand microtubule polymerization *in vitro* (Fig. 3e, f). Overall, the multitude of complementary cell-based assays and *in vitro* experiments provides strong support for our model where the phase boundary of immiscible chromatin has sufficient surface tension to withstand the forces generated by microtubule polymerization both in liquid condensates as well as in intact chromosomes.

General comments:

1) It is not clear how to understand microtubule perforation when chromatin is ‘dissolved’ upon hyperacetylation. What defines the chromatin boundary? How should the reader conceptualize perforation into a liquid like state?

We realize based on this comment that some parts of our original manuscript might have been difficult to understand. We therefore rephrased the text to provide an extended introduction and discussion of the different concepts. According to our model, the boundary of hyperacetylated chromatin is similar to that of a swollen gel in which the polymer network is in a soluble state, whereas the boundary of deacetylated chromatin resembles that of a collapsed gel in its immiscible state, then resembling the phase boundary of a polymer melt. We propose that microtubules can perforate the swollen/soluble hydrogel state, but not the immiscible/collapsed state of the hydrogel. The resistance of the chromatin hydrogel to microtubule perforation arises from the transition to an insoluble state. We show that this transition requires deacetylation and that the immiscible chromatin hydrogel excludes soluble tubulin dimers and has enough surface tension to withstand microtubule perforation, such that astral microtubules displace chromatin towards the cell periphery even when the continuity of the chromatin fiber is interrupted by AluI-mediated fragmentation.

2) How does TSA treatment or p300 overexpression affect the levels of the barrier-to-autointegration factor (BAF)? Some of these authors previously showed that BAF cross bridges mitotic chromosomes (Samwer et al., 2017) to form a mechanically stiff layer at the surface of mitotic chromosomes in anaphase. Is this surface still forming in anaphase to reduce the effects of hyperacetylated metaphase chromosomes? What is BAF’s response to hyperacetylation?

Our previous study by Samwer et al., 2017, showed that BAF forms a dense chromatin network on the surface of anaphase chromosomes to guide nuclear envelope assembly, such that all chromosomes are packaged into a single nucleus rather than into a set of micronuclei. Our prior study further showed that during mitosis, BAF completely dissociates from chromatin by VRK1-mediated phosphorylation at its DNA-binding site and that BAF also associates with interphase chromatin, when the acetylation levels are high. Based on these findings, we do not expect that TSA treatment affects BAF localization during mitosis. Furthermore, as BAF is not bound to chromosomes from prometaphase to mid anaphase, we can rule out a BAF-dependent network on

chromosomes contributing the exclusion of microtubules during these early stages of mitosis, which were investigated in our current manuscript.

To address the question raised by the reviewer, we recorded time-lapse videos of cells expressing BAF-EGFP in the presence or absence of TSA, finding that neither the dissociation of BAF from chromosomes during metaphase nor the binding of BAF to mitotic chromosomes during mitotic exit is affected by TSA (see Supporting Fig. 1). While we find this observation interesting, we think that these data are not directly related to the main points of our study. As our revised manuscript already exceeds the length of regular *Nature* articles, we suggest not including it. If the reviewer thinks that these data should be included, however, we would not object and shorten the manuscript elsewhere.

Supporting Figure 1. Characterization of BAF binding to metaphase and anaphase chromosome ensembles in the absence or presence of TSA. **a-b**, Time lapse imaging of mitotic exit in the absence or presence of TSA. **a**, HeLa cells expressing eGFP-BAF were stained with 250 nM SiR-DNA. Metaphase cells were identified based on cell morphology and mitotic exit recorded. Single confocal Z-section. **b**, HeLa cells expressing eGFP-BAF were stained with 250 nM SiR-DNA and treated with TSA. Metaphase cells were identified based on cell morphology and mitotic exit recorded. Single confocal Z-section. **c**, Quantification of eGFP-BAF fluorescence on chromatin relative to cytosol after anaphase onset ($t=0$ min) and throughout mitotic exit in cells as illustrated in **a**, **b**. Curve and range indicate mean \pm SD, $n=16$ cells for control, $n=14$ cells for TSA. Scale bars, 5 μm .

Referee #2 (Remarks to the Author):

The physical principle on how long chromatin fibers are organized into mitotic chromosomes remains unclear and an important issue in biophysics as well as cell biology.

To approach this issue, Schneider et al. manipulated the histone acetylation state of mitotic chromosomes in living cells. The authors first found that hyperacetylated mitotic chromosomes decondensed drastically with condensin-depletion. They also demonstrated hypoacetylation-induced chromatin compaction prevented microtubule perforation. Interestingly, AluI-digested mitotic chromatin formed liquid-like droplet bodies (chromatin condensates) in a condensin-independent manner, and these bodies were dissolved by histone hyperacetylation. Furthermore, the chromatin condensates excluded negatively charged macromolecules, including tubulins, and were pushed to the cell periphery by microtubules, suggesting that histone deacetylation during mitosis allows mitotic chromosomes to resist perforation by microtubules.

Overall, the presented data is very impressive. The paper would make a big impact on the cell biology and biophysics fields. For publication in Nature, there are several points to be addressed. My specific comments are the following:

We thank the reviewer for the appreciation of our work and the constructive suggestions for further improvements. We addressed all points as explained below.

Major points:

1) It is known that Ki-67, which is a positively charged large molecule and locates at the chromosome periphery, is involved in mitotic chromosome formation (e.g., PMID: 29487178, 27362226, 27610954, and 24867636). It would be critical to show how Ki-67 behavior and function are affected upon the authors' manipulations: TSA-, AluI-, and AluI/TSA-treatments in the presence or absence of condensin.

This is indeed a very interesting aspect, as our mitotic chromatin phase separation model predicts that neither condensin depletion nor AluI-mediated fragmentation of chromatin should affect the targeting of Ki-67 to the chromatin surface, as Ki-67's localization should result from its amphiphilic attraction to cytoplasm and chromatin phases rather than binding to specific genomic regions. To test this, we imaged live cells expressing Ki-67-EGFP and found that, consistent with our model, Ki-67 still enriched at the surface of chromatin regions both under SMC4-degradation as well as AluI injection conditions (new Fig. 2o-q and Extended Data Fig. 8e, f). Our model further predicts that TSA treatment should suppress the formation of a phase boundary between chromatin and cytoplasm, which is expected to impair the concentration of Ki-67 at a sharp chromosome boundary. Imaging TSA-treated cells indeed showed that Ki-67 localized more diffusely and entered into chromatin regions (new Fig. 2o-q). Furthermore, complete dispersion of chromatin in TSA-treated cells depleted of condensin showed a homogenous distribution of Ki-67-EGFP throughout chromatin (new Extended Data Fig. 8e, f). To test a potential requirement of Ki-67 for the formation of immiscible chromatin condensates, we microinjected AluI into Ki-67 knockout cells, finding that the formation of chromatin condensates does not require Ki-67 (new Extended Data Fig. 8g, h).

These new data provide additional support for an acetylation-regulated phase transition of a chromatin hydrogel. Moreover, they corroborate that Ki-67's targeting to the chromosome surface arises from its amphiphilic properties where the C-terminus is attracted to the mitotic chromatin

phase and the N-terminus attracted to the cytoplasm phase. These data hence substantiate the model of a phase boundary formed between immiscible chromatin and cytoplasm.

2) "chromatin solubility," "insoluble chromatin phase," and "mitotic chromatin is insoluble."

I am not so sure whether these phrases and sentences often used in the text are appropriate because dense mitotic chromosomes are readily accessible to diffusing proteins in living cells (e.g., PMID: 23246002, 15623580). Condensin should also be making loops during the formation process of mitotic chromosomes.

We thank the reviewer for requesting clarification, as we realize that providing more background information about the hydrogel theory will help to understand the relevance of our work. In the immiscible state, the hydrogel is still accessible for macromolecules that partition into that phase (e.g., positively charged dextrans or GFP supercharged mutants). A key insight from our study is that in the immiscible state, chromatin fragments form a liquid with rapid lateral mobility, rather than forming an immobile aggregate (see Fig. 2c-d). This explains how condensin can continue to dynamically extrude loops even in the highly compacted state of mitotic chromosomes. To clarify these points, we extended and rewrote parts of the main text. We also specifically refer to the fact that our model can explain how condensin can mediated dynamic looping in the compact mitotic state:

“The immiscible mitotic chromatin excludes tubulin dimers and forms a surface that provides resistance to microtubule perforation while allowing local chromatin fiber sliding internally, as required for continuous dynamic loop formation by condensin^{6,14,60}. In parallel, condensin-mediated linkages establish a hydrogel that withstands tension generated at kinetochores¹⁴⁻¹⁶. Jointly, these molecular activities shape discrete chromosome bodies with a defined surface despite continuous internal remodeling of the chromatin fiber.”

3) The authors nicely demonstrated that condensin is not a key determinant of mitotic chromatin condensation or compaction. It would be nice to stress this point.

We agree that this is an important point and revised parts of our manuscript to clarify the role of condensin. We state in the first section of the main text: *“Thus, histone deacetylation is necessary and sufficient for complete compaction of mitotic chromatin even in the absence of condensin. In contrast, condensin is neither necessary nor sufficient for complete chromatin compaction during mitosis, yet it can concentrate chromatin to some extent even when histones are hyperacetylated.”*

4) Fig. 2a. I wonder where condensin is in the AluI-treated chromatin bodies and AluI/TSA treated ones, although condensin-depleted chromatin forms similar bodies upon AluI-treatment. Is condensin associated with the bodies or excluded from them?

This is an interesting question that we addressed by imaging cells expressing Halo-tagged condensin before and after microinjecting AluI. We found that condensin evenly distributed throughout chromatin droplets after AluI-mediated fragmentation, at a concentration similar to that in surrounding cytoplasm (new Extended Data Fig.7c, d). This observation is consistent with a model where condensin can still access and interact with chromatin after fragmentation, but chromosome axes cannot be formed anymore. Regarding the specific request to also study the localization of condensin in TSA/AluI-treated chromatin bodies, we would like to clarify that under this condition, chromatin does not form condensates, as it is evenly dispersed throughout the cytoplasm. We present the new data in the revised manuscript: *“Imaging Halo-tagged SMC4 further showed that condensin*

did not form axial structures inside the chromatin condensates and instead evenly distributed throughout the cell, validating efficient chromosome fragmentation by AluI (Extended Data Fig. 7c, d)."

5) The AluI-treated chromatin bodies look like apoptotic chromatin, which is a highly condensed structure with linker DNA digestion. It would be nice to show or discuss how they are similar and different. Apoptotic chromatin and mitotic chromosomes seem to have distinct properties (e.g., PMID: 23246002).

While we agree that it would be interesting to study the material properties of chromatin in apoptotic cells as well, we are concerned that experiments in this direction are beyond the scope of our manuscript, as it already covers so many different aspects of chromosome and spindle biology. However, one important question arising in this context is whether the chromatin fragmentation by AluI induces apoptosis within the time window during which we analyzed the chromatin condensates. To address this aspect, we imaged the early apoptosis biosensor pSIVA, which probes lipid inversion at the plasma membrane (Kim et al., 2010, PMID: 19966809). This experiment showed that AluI injection does not lead to detectable pSIVA fluorescence even one hour after injection (Extended Data Fig. 7a, b), longer than any analysis that we performed for AluI-induced condensates.

We think that further investigations of how the chromatin material properties change during apoptosis and other physiological processes are interesting avenues of future research. We provide this perspective in our revised conclusions section: *"It will be interesting to determine how chromatin adapts its material properties to other physiological processes, as for example apoptosis."*

6) TSA concentration used (5 μ M) is pretty high. For instance, \sim 0.33 μ M in Ref. 10 and 0.33 -0.66 μ M in other papers (PMID: 16317046; PMID: 28712725). To minimize possible indirect effects, lower concentrations of TSA should also be examined.

We thank the reviewer for pointing out how the specificity of phenotypes resulting from TSA treatment could be further validated. We thus tested the effect of lower concentrations and found that 500 nM TSA induced mitotic chromatin decompaction as efficiently as 5 μ M (new Extended Data Fig. 4). We also tested if the phenotypes induced by TSA treatment can be reverted by washout, which is indeed the case (new Extended Data Fig. 4). We also assessed whether TSA treatment induce apoptosis or DNA damage at the time frame of our experiments, which was not the case (new Extended Data Fig. 3e-h). Together, these data corroborate the specificity of the mitotic phenotypes resulting from TSA treatment.

Minor points:

1) Double treatments of condensin-depletion and TSA changed chromosome morphology drastically. One of the features of mitotic chromosomes is the phosphorylation of Ser-10 in histone H3. The authors may want to see the H3Ser10P in the treated chromosomes.

We appreciate this suggestion, but this experiment is unfortunately not feasible, as the affinity of the antibody raised against H3-pS10 is also affected by the acetylation state of nearby lysines, and the detection of H3-pS10 thus expected to be affected by hyperacetylation induced by TSA.

However, we would like to note that multiple experiments clearly indicate that condensin-depleted and TSA-treated cells are in a mitotic state. First, we observed the formation of a bipolar spindles despite the gross perturbations of chromosome morphology (Fig. 1a). Second, we probed the mitotic state by an antibody specific for Cdk1-dependent phosphorylations on a broad substrate range (Extended Data Fig. 1d, e). To further validate the mitotic state under the perturbation conditions used in our study, we recorded time-lapse videos of cells entering mitosis, with markers for microtubules (SiR-tubulin), chromatin (H2B-mCherry), and kinetochores (CENP-A-EGFP) (new Extended Data Fig. 2 and Supplemental Videos 1-3). These data show that condensin-depleted and TSA-treated cells assemble a spindle that attaches to kinetochores, further validating the mitotic state.

2) Fig. 1a. I found there are cytoplasmic Hoechst signals (foci). What are they? I am just afraid of mycoplasma contamination or something, which could affect all the results.

As we regularly test all cells used in our lab for mycoplasma contaminations, we can exclude that the fluorescent background signal in the Hoechst channel is due to mycoplasma infection. The faint cytoplasmic foci in the fluorescence channel used for Hoechst imaging is unspecific background signal that is always observed with our standard illumination settings in HeLa cells. We would like to note that consistent condensin-depletion/TSA treatment phenotypes are observed in fixed cells (Extended Data Fig. 1f), where no cytoplasmic background signal is visible. Moreover, in our revised manuscript we included videos for condensin-depleted and TSA-treated live cells in which chromatin is visualized by H2B-mCherry, where there is also no background in the cytoplasm (Extended Data Fig. 2 and Supplementary Videos 1-3). We can therefore confidently exclude that the low-level background signal in Fig. 1a compromises the validity of the phenotype interpretation.

3) "since entanglements occurring due to growth of microtubules..." This sentence is not clear to me, and should be rephrased better.

As chromosomes are composed of a network of chromatin fiber loops, we expect that microtubules that grow through chromosome arms would potentially cause entanglements that impair the independent motility of chromosomes during segregation. We clarified this model by rephrasing the respective sentence in the manuscript.

4) The method descriptions for some critical experiments are insufficient. For instance, how did the authors deplete SMC4 in mitotic cells with AID2 system? In G2 synchronized cells or asynchronous ones? How much AluI was injected into a mitotic cell? What is 1 volume of AluI-stock?

We thank the reviewer for pointing out that some experimental details were missing in the methods section, and revised our manuscript for more comprehensive description of procedures.

Referee #3 (Remarks to the Author):

During cell division, the eukaryotic chromatin condenses to form metaphase chromosomes that are precisely organized in the central plane of the dividing cell by a balance of pulling and pushing forces applied by microtubules of the mitotic spindle. The chromatin inside condensed chromosomes does not have a stiff fiber-like higher-order structure but rather forms flexible loops organized by rings of condensin protein and behaves as a condensin-dependent hydrogel under external pulling. Recently, the liquid-liquid phase separation (LLPS) hypothesis was put forward as a mechanism for chromatin condensation in repressed heterochromatin and metaphase chromosomes. Still, the liquid or solid state of the chromatin condensates is disputable as the liquid state was previously shown to depend on DNA fragmentation and specific experimental conditions *in vitro*. In relation to metaphase chromosomes, how such a fluid (or just soft) chromosomal structure could withstand the pushing and pulling forces from the mitotic spindle without chromosomal breaks and mis-segregations remains a fundamental open question.

In this work, the authors combine a conditional condensin knockout technique with a histone deacetylase inhibitor-induced histone hyperacetylation to study whether these two factors play any role in resisting pushing force and penetration of the chromosome bodies by the mitotic spindle in living metaphase cells. They also designed an unconventional and elegant approach to study chromatin condensates and their pushing by microtubules *in situ* to draw parallels between the whole chromosomes and droplet-like chromatin condensates and thus investigate the mechanism of chromosome resistance to pushing force. The resulting manuscript presents several exciting and seminal findings: i. Condensin protein is shown to be dispensable for withstanding the pushing force applied by spindle microtubules on metaphase chromosomes in living cells while histone deacetylation plays a crucial role in this process – a truly remarkable finding provided that the prevailing view is that condensed chromosomes can be assembled and supported by condensin without significant contribution from the histones. ii. An increased histone acetylation can disrupt chromatin condensates in mitotic cytoplasm under the same physiological conditions as those that are sufficient to dissolve chromatin droplets thus revealing profound mechanistic parallels between the two events. iii DNA integrity is not required for withstanding the pushing force of microtubules in a striking contrast with previous chromosome pulling experiments. iv. By showing that chromatin condensates acquire a surprising property to resist penetration by microtubules as well as by modified negatively charged DsRed protein the work suggests that the chromatin droplets have a negatively charged outer shell that forms a distinct physical boundary on the surface of the condensates. v. Spindle microtubules are likely to apply the pushing force directly on the surface of chromatin condensates rather than act through chromokinesin proteins. This finding implies that the condensed chromatin boundary is strong enough to relay the pushing force of polymerizing microtubules into concerted mobility of all nucleosome arrays within the chromatin condensate.

Overall, this manuscript is based on an innovative experimental design, is technically sound, clearly written, and it puts forward a new fundamental mechanism underlying chromosome integrity and mobility during cell division. This work will inspire further theoretical and experimental studies on the physical properties of chromatin condensates and their boundaries. For biomedical applications, such as gene editing and therapy, the results of this work are likely to aid in designing new molecular tools for targeting metaphase chromosomes and chromatin condensates. Still, I have some concerns that need to be experimentally addressed to ensure its publication in Nature:

We thank the reviewer for the positive assessment of our work and for the thoughtful suggestions on how to further improve the manuscript. We addressed the specific concerns as explained below.

1. This work uses HDAC inhibition by 5 μ M TSA to observe chromosome transitions caused by histone hyperacetylation in living cells. Such treatment may have many pleiotropic and deleterious effects. It had been shown that just 20 nM can cause growth inhibition and apoptosis in HeLa cells (see e.g. You and Park, PMID: 23165748) and thus cause DNA fragmentation. In its turn, DNA fragmentation may significantly change the properties of the chromosome as it appears from the AluI experiments. Therefore, for live-cell imaging experiments the control, TSA-treated and p300 cells should be examined for the cell viability, and absence of DNA breaks in situ (e.g. TUNEL assay).

We thank the reviewer for suggesting how to further validate the specificity of the TSA phenotypes and performed several new experiments to address this concern. Regarding the previously observed toxic effect of TSA, we would like to note that toxicity is only observed days after TSA addition, whereas in our experiments we apply TSA only 2.5 h before studying mitotic phenotypes.

To address the referee's concern, we determined whether 5 μ M TSA treatment over 2.5 h induces apoptosis, using the early apoptosis marker pSIVA (Kim et al., 2010, PMID: 19966809) and found no detectable effect on cell viability (new Extended Data Fig. 3d, f). We also studied the effect of ten-fold lower TSA concentration, and found that the chromatin decompaction phenotype is fully penetrant in 500 nM TSA (new Extended Data Fig. 4). To further corroborate potential effects resulting from toxic side effects of TSA treatment, we tested if the phenotype induced by 3 h TSA treatment can be reverted by removing the compound 8 h before imaging mitotic cells, finding a chromosome morphology and compaction similar to controls (new Extended Data Fig. 4).

We also investigated whether TSA induces DNA double-strand breaks, using immunofluorescence staining against γ H2A.X, an early factor of the DNA damage response pathway, finding no increase of γ H2A.X on mitotic chromosomes of TSA-treated cells (new Extended Data Fig. 3g, h).

Together, these data provide strong support for the conclusion that TSA induces mitotic chromosome decompaction via hyperacetylation and not pleiotropic toxic side effects.

2. The immunofluorescence experiments (extended data 2 and 3) are not sufficient to convince the reader that the TSA treatment can effectively change the histone charge for altering chromatin condensation. In order to understand whether histone acetylation could change the physical properties and dissolve chromatin condensation directly, by changing the histone charge, the authors should demonstrate efficiency of TSA and p300 treatments by direct measurement of histone charges such as AUT gels (see Shechter et al., 2007 PMID: 17545981) or mass-spectrometry.

Regarding the efficiency of TSA treatments, we would like to refer to a large body of published literature that shows with various techniques, including mass spectrometry, that TSA induces hyperacetylation on various core histones (e.g., Galasinski et al., 2002, PMID: 11709551; Cozzolino et al., 2021, PMID: 33669725). Our assessment of histone acetylation after TSA treatment by immunofluorescence provides a clear indication of hyperacetylation on three different core histones, based on three different antibodies. We would like to note that these experiments were carefully controlled by comparison of acetylation levels of TSA-treated cells with untreated interphase as well as mitotic cells, whereby the decreased acetylation detected in mitotic cells compared to interphase cells validates the sensitivity of our immunofluorescence assay.

The consistency of our observations with the previous investigations of TSA, and the similar phenotypes observed after p300 overexpression or TSA treatment, provide evidence for the efficiency of the compound. In the revised manuscript, we have included various additional controls further validating toxicity and specificity of TSA, including assays for apoptosis and DNA damage,

an analysis of 10-fold lower concentration of TSA (500 nM), and a demonstration that the mitotic phenotypes can be reversed by removing TSA after 3 h treatment (see the response to point 1). While we appreciate the suggestion for further detailed characterization of histone modifications by AUT gels or mass spectrometry, we are concerned that such experiments would further delay the publication of our manuscript, given the very large number of new experiments that were conducted in the framework of this revision.

Regarding the overall electrical charge of chromatin, we would like to note that this is not only affected by histones and their acetylations, but thousands of other proteins associated with chromatin. It will be hence be impossible to infer the overall charge of endogenous chromatin from the charge detected on histones *in vitro*, e.g. by AUT gels or mass spectrometry. For this reason, we investigated electrostatic interactions between soluble components and endogenous chromatin in live cells using probes with defined electrical charge, like GFP charge mutants or dextrans.

3. Page 2, Fig. 1 b, c and elsewhere: This work intermittently uses different statistical tests to show absence or presence of significant differences between pairs of data sets in the same experiment (e.g. t-test, Mann-Whitney test, Kolmogorov-Smirnov test). This depends on whether the data have normal or non-normal distributions and similar or different variances. Since it is hard for the reader to evaluate the data normality and variance, the use of different statistical tests leaves an impression that the t-test, for example, is applied arbitrary when the dataset pairs are expected to be similar. I suggest that at least one statistical test (in addition to the tests optimal for each given type of datasets) should be consistently applied to all datasets as a universal discriminator between significant and non-significant differences.

Our choice of statistical tests was not arbitrary but based on a standardized quantitative procedure: we applied t-tests only where the samples were distributed normally and with equal variance, as determined by Shapiro–Wilk/D’Agostino–Pearson and F-test/Levene’s tests ($\alpha = 0.05$). The explanation of our statistical assessment was provided in the methods subsection “Statistical analyses and data reporting” on p. 18 of the original manuscript.

However, to avoid the impression that the type of test was chosen to fit our conclusions, we decided to use, as the reviewer suggested, one statistical test consistently applied to all datasets as a universal discriminator between significant and non-significant differences.

Since the Welch’s t-test robustness to the violation of normality is not guaranteed at smaller sample sizes, we chose the robust non-parametric Wilcoxon-Mann-Whitney-Test (Mann-Whitney-Test). With this new testing procedure, we still detected statistical significance for all data previously found to be significant, and we updated all p-values accordingly.

4. Page 13-14 (methods) describes CLEM – correlative light and electron microscopy. However, no figure shows any H2B fluorescence correlated with EM. Fig 1. D appears to show segmentation based on electron density, not on H2B fluorescence. The parallel H2B fluorescence images should be included to show that the featured EM densities correspond to the histone fluorescence in chromosomes.

We thank the reviewer for pointing out how to improve the presentation of the CLEM data. We would like to clarify that we used the H2B fluorescence signal only to identify mitotic cells and then segmented the chromatin regions based on the texture in EM tomograms, as the resolution of light microscopy is insufficient to obtain a detailed segmentation contour. However, we do agree that it is important to show that electron dense regions indeed represent chromatin, and therefore included

overlay images of H2B-mCherry fluorescence and electron density for two examples (Extended Data Fig. 6d).

Additional minor points:

Page 1, right, 6th line from the top: “stained DNA with Hoechst to...” Hoechst is a general brand name, please indicate which type was used, e.g. Hoechst 33342

We thank the reviewer for pointing out that information about the specific compound used to stain DNA was missing and have added it to the revised manuscript.

Page 15, right, 4th line from the top: here “DNA channel” refers to Hoechst 33342 staining and in the next paragraph “DNA density” in a similar experiment seems to be derived from H2B fluorescence. The nature of fluorescence signals should be indicated in all cases.

We agree and have provided the missing information in the revised manuscript.

Reviewer Reports on the First Revision:

Referees' comments:

Referee #1 (Remarks to the Author):

The authors have addressed many of the concerns I raised, and the findings reported in the different revised figures are improved and the conclusions have strengthened. Unfortunately, I am still not convinced about one of the central claims (I quote): "...chromatin compartment that resists microtubule growth by the exclusion of soluble tubulin and formation of a mechanical barrier towards polymerization".

I accept that tubulin is likely excluded from the chromatin compartments and the finding the growing microtubules can somehow move these chromatin compartments. However, there is a fundamental issue with the reasoning that depletion of tubulin monomers from the chromatin somehow prevents microtubules from growth into chromosome regions. Specifically, data in Figure 3f shows that tubulin concentration is reduced to 30-40% in the chromatin compared to buffer. This reduction in concentration is insufficient to block microtubule growth from growing filament ends. The authors are likely aware of the studies that have shown that the concentration of tubulin needed for growth of existing filaments in the presence of templates is ~7-9 μ M, and based on estimates of cellular tubulin concentrations, a depletion of much greater than 30-40% is needed to block growth and thereby prevent microtubules from penetrating chromatin.

In addition, the abstract states (I quote): "Our study highlights the different contributions of DNA loop formation and chromatin phase separation to genome segregation in dividing cells."

The proposal that chromatin phase separation makes a different contribution than DNA loop formation is based on analyses of histone acetylation, and separately, the links between acetylation and phase separation. However, DNA loop formation and histone acetylation (and following the authors' reasoning, phase separation) may be linked and interdependent and therefore, the statement that 'different contributions' are being highlighted by the study is not an entirely accurate statement. This concern is not merely about this statement, but the overall conceptual advance reported in this manuscript.

Overall, there are several nice data reported by the authors but unfortunately, I do not find that these data do not come together strongly enough to support a major conceptual advance with a broad and sustained impact. I am afraid, I am unable to recommend this manuscript for publication in Nature.

Referee #2 (Remarks to the Author):

The authors have adequately addressed most of my comments and very much improved the manuscript. This paper would be a landmark paper in the fields of chromosome cell biology and biophysics. I would support the publication of this paper after some minor issues are addressed:

1. Fig. 1a and b. How did the authors quantify chromatin fraction at the spindle center?
2. Page 5. "This resulted in homogeneously dispersed chromatin fragments with almost no local condensates (Fig. 2e, f; Supplementary Video 5)".
I still see some foci in the Figs and Video 5. What are they?
3. Page 6. "Two negatively charged microtubule-regulating proteins, ..., were also excluded from mitotic chromatin in a TSA-dependent manner (Extended Data Fig. 9e-h)."
"Both dextrans were excluded from chromatin in a TSA-dependent manner,..."
Both sentences are a bit confusing. I think "included" is better than "excluded"?

Referee #3 (Remarks to the Author):

This is a revised manuscript in which the authors have constructively addressed all main critique points raised by the reviewers. Importantly, they have convincingly addressed the concern expressed by all three reviewers that the mitotic phenotypes resulting from TSA treatment with a rather high concentration of TSA, an HDAC inhibitor, could be due to hyperacetylation and not unspecific toxic side-effects of the compound by showing that a much lower concentration of TSA is efficient and that the phenotypes caused by TSA are reversible and do not induce apoptosis.

The authors have also presented several additional experiments corroborating their main findings and streamlined the statistical analysis. Although not all experiments suggested by the reviewer's have been carried out, such as describing the level of histone charge changes leading to the dissipation of hydrogels, I concur with the authors that this question may be too complicated to be answered within the scope of this study.

There was a significant concern expressed by one of the reviewers questioning the novelty of this manuscript in relation to the earlier paper of Gibson et al (Cell, 2019). In my opinion, the present manuscript presents a truly innovative and substantial advance in understanding chromatin biology as it provides a new physiological role of the hydrogels during mitosis which was not considered in the 2019 paper.

Altogether, the new experiments and clearer interpretations presented in the revision have strengthened my previous conclusion that this manuscript describe exciting new findings based on an innovative experimental design, is technically sound, clearly written, and it puts forward a new fundamental mechanism underlying chromosome integrity and mobility during cell division. The revised manuscript has been carefully edited by the authors and may be recommended for publication in its present form.

Author Rebuttals to First Revision:

Response to the Referees' comments (Nature manuscript 2021-07-10665A by Schneider et al.)

Referee #1 (Remarks to the Author):

The authors have addressed many of the concerns I raised, and the findings reported in the different revised figures are improved and the conclusions have strengthened. Unfortunately, I am still not convinced about one of the central claims (I quote): "...chromatin compartment that resists microtubule growth by the exclusion of soluble tubulin and formation of a mechanical barrier towards polymerization".

I accept that tubulin is likely excluded from the chromatin compartments and the finding the growing microtubules can somehow move these chromatin compartments. However, there is a fundamental issue with the reasoning that depletion of tubulin monomers from the chromatin somehow prevents microtubules from growth into chromosome regions. Specifically, data in Figure 3f shows that tubulin concentration is reduced to 30-40% in the chromatin compared to buffer. This reduction in concentration is insufficient to block microtubule growth from growing filament ends. The authors are likely aware of the studies that have shown that the concentration of tubulin needed for growth of existing filaments in the presence of templates is ~7-9 μ M, and based on estimates of cellular tubulin concentrations, a depletion of much greater than 30-40% is needed to block growth and thereby prevent microtubules from penetrating chromatin.

We propose that mitotic chromatin forms a phase dense in negative charge, which excludes soluble tubulin and microtubules via electrostatic repulsion. To which extent microtubule polymerization inside chromatin is limited by repulsion of growing microtubules at the phase boundary versus sequestration of free tubulin from inside chromatin cannot be experimentally discriminated, as both effects emerge from the same repulsion mechanism. We therefore suggested that both effects are relevant for microtubule polymerization in the first paragraph of the conclusions section: "*The immiscible mitotic chromatin excludes tubulin dimers and forms a surface that provides resistance to microtubule perforation*". Importantly, our reconstitution experiments using recombinant nucleosome arrays and purified tubulin show that microtubule exclusion is a feature emerging from the intrinsic phase separation properties of chromatin, independently of other cellular factors.

We disagree with the critique that the concentration of tubulin inside cells is too high for the observed sequestration of tubulin inside chromatin to potentially affect polymerization. While the concentration of tubulin inside cells is indeed higher than the critical concentration required for microtubule polymerization, we must consider the concentration of **free tubulin** – not the concentration of **total tubulin** – as the parameter relevant for polymerization rates. In a mitotic steady state, a large fraction of tubulin is incorporated in the spindle such that the concentration of free tubulin in the cytoplasm is very low and expected to be close to the critical concentration required for polymerization. This is apparent in Fig. 3a of our manuscript, where free tubulin in cytoplasmic regions outside the spindle is barely detectable (left panel; compare with the distribution of depolymerized tubulin upon nocodazole treatment, right panels). Based on the principles of steady-state dynamics, it is therefore reasonable to propose that a further reduction of free tubulin inside chromatin contributes to the exclusion of microtubule from mitotic chromosomes, besides the barrier effect imposed by the phase boundary on microtubule tips.

Nevertheless, to address the reviewer's concern, we rephrased the manuscript to remove any direct claims on the potential relevance of reduced free tubulin concentration inside mitotic chromatin.

In addition, the abstract states (I quote): "Our study highlights the different contributions of DNA loop formation and chromatin phase separation to genome segregation in dividing cells."

The proposal that chromatin phase separation makes a different contribution than DNA loop formation is based on analyses of histone acetylation, and separately, the links between acetylation and phase separation. However, DNA loop formation and histone acetylation (and following the authors' reasoning, phase separation) may be linked and interdependent and therefore, the statement that

‘different contributions’ are being highlighted by the study is a not an entirely accurate statement. This concerns is not merely about this statement, but the overall conceptual advance reported in this manuscript.

Our manuscript presents several experiments that directly establish a link between chromatin phase separation and microtubule exclusion – independently of perturbations affecting acetylation. We show that pure chromatin forms condensates that exclude tubulin and microtubules via an intrinsic propensity to phase separate (Fig. 3e, f). Moreover, we show that fragmentation of chromosomes by injection of AluI forms an immiscible liquid-like phase that resists polymerizing microtubules (Fig. 4; Extended Data Fig. 10).

We also show that condensin is not required for phase separation of AluI-fragmented chromatin, in experiments that do not involve perturbations of acetylation (Extended Data Fig. 7). Moreover, we show that acetylation suppresses chromatin phase separation in a way that does not impair the organization of chromosomes into thread-like structures that resist tension at kinetochores (Extended Data Fig. 1f, g). Our experiments with injecting mixtures of in vitro-acetylated nucleosome arrays and native nucleosome arrays into cells (Fig. 2m, n) further demonstrate that chromatin partitioning into an immiscible phase must involve a mechanism that is independent of condensin.

Together, these findings provide clear evidence that chromatin phase separation and DNA loop formation make distinct contributions to mitotic chromosomes.

Overall, there are several nice data reported by the authors but unfortunately, I do not find that these data do not come together strongly enough to support a major conceptual advance with a broad and sustained impact. I am afraid, I am unable to recommend this manuscript for publication in Nature.

We are disappointed that this reviewer does not recognize the importance and coherence of our discoveries. There might be a misperception about the relevance of our work, given that this reviewer’s expertise is not in the chromosome or phase separation fields. We would like to summarize once more how our work provides conceptual advance at multiple levels:

Our study provides insights into the organization and material properties of chromosomes and reveals a previously unrecognized aspect of how chromosomes interact with microtubules of the mitotic spindle. We show that histone deacetylation during mitosis generates a sharp global reduction in chromatin solubility that converts chromosomes into phase-separated bodies. This immiscible chromatin phase excludes negatively charged macromolecules such as tubulin and creates a barrier towards microtubule perforation. We demonstrate that creating a barrier to microtubule perforation is sufficient to provide mechanical resistance to pushing forces independently of condensin, whereas condensin-mediated connection of DNA loops is necessary to prevent shearing as microtubules pull on kinetochores. Thus, the material properties required to resist the pushing and pulling forces acting on chromosomes can be uncoupled. Importantly, our results explain how it is possible to generate mechanical resistance at the surface of chromosomes while still allowing dynamic DNA looping by condensin inside chromosomes. The material properties emerging from the mitotic chromatin phase transition counteract entanglements and potential segregation defects caused by microtubule perforation. As such, we are convinced that our manuscript offers a significant conceptual advance at the interface of cell biology and biophysics that will be of broad interest.

The two other reviewers assess our work as a “landmark paper in the fields of chromosome cell biology and biophysics” (reviewer #2) and “truly innovative and substantial advance in understanding chromatin biology” (reviewer #3).

Referee #2 (Remarks to the Author):

The authors have adequately addressed most of my comments and very much improved the manuscript. This paper would be a landmark paper in the fields of chromosome cell biology and biophysics. I would support the publication of this paper after some minor issues are addressed:

We thank the reviewer for the positive assessment of our manuscript and for pointing out some remaining points that need clarification. Below, we provide answers to the specific questions.

1. Fig. 1a and b. How did the authors quantify chromatin fraction at the spindle center?

The quantification procedure was explained in the methods part in the section about image analysis, under the headline DNA congression analysis. We rephrased this paragraph for improved readability.

2. Page 5. "This resulted in homogeneously dispersed chromatin fragments with almost no local condensates (Fig. 2e, f; Supplementary Video 5)".

I still see some foci in the Figs and Video 5. What are they?

While the large majority of chromatin is dispersed after AluI injection into TSA-treated cells, there are indeed some small condensates. It was previously shown that short-term treatment with TSA induces hyperacetylation throughout the genome except of some constitutive heterochromatin regions like those in the vicinity of centromeres (reviewed in Taddei et al., 2005, PMID: 15940285). We therefore suspect that the few foci detected after AluI injection into TSA-treated cells represent constitutive heterochromatin regions resistant to TSA-induced hyperacetylation. These foci, however, represent only a small fraction of the genome and can be explained by known limitations of the perturbation procedure, hence they do not affect the conclusions drawn in the paper. However, based on the reviewer's comment we realize that it is important to comment on this observation and revised the text accordingly.

3. Page 6. "Two negatively charged microtubule-regulating proteins, ..., were also excluded from mitotic chromatin in a TSA-dependent manner (Extended Data Fig. 9e-h)."

"Both dextrans were excluded from chromatin in a TSA-dependent manner,..."

Both sentences are a bit confusing. I think "included" is better than "excluded"?

We thank the reviewer for pointing out that these sentences were difficult to understand and rephrased them:

"Two negatively charged microtubule-regulating proteins, p80-Katanin⁵² and the microtubule plus-tip component EB3³³, were also excluded from mitotic chromatin, whereby TSA treatment resulted in a less efficient exclusion from mitotic chromatin (Extended Data Fig. 9e-h)."

"Both dextrans were excluded from chromatin, whereby the exclusion was more efficient at larger molecular weight; TSA treatment reduced the exclusion of both dextrans from mitotic chromosomes (Extended Data Fig. 9i-l)."

Referee #3 (Remarks to the Author):

This is a revised manuscript in which the authors have constructively addressed all main critique points raised by the reviewers. Importantly, they have convincingly addressed the concern expressed by all three reviewers that the mitotic phenotypes resulting from TSA treatment with a rather high concentration of TSA, an HDAC inhibitor, could be due to hyperacetylation and not unspecific toxic side-effects of the compound by showing that a much lower concentration of TSA is efficient and that the phenotypes caused by TSA are reversible and do not induce apoptosis.

The authors have also presented several additional experiments corroborating their main findings and streamlined the statistical analysis. Although not all experiments suggested by the reviewer's have been carried out, such as describing the level of histone charge changes leading to the dissipation of hydrogels, I concur with the authors that this question may be too complicated to be answered within the scope of this study.

There was a significant concern expressed by one of the reviewers questioning the novelty of this manuscript in relation to the earlier paper of Gibson et al (Cell, 2019). In my opinion, the present manuscript presents a truly innovative and substantial advance in understanding chromatin biology as it provides a new physiological role of the hydrogels during mitosis which was not considered in the 2019 paper.

Altogether, the new experiments and clearer interpretations presented in the revision have strengthened my previous conclusion that this manuscript describe exciting new findings based on an innovative experimental design, is technically sound, clearly written, and it puts forward a new fundamental mechanism underlying chromosome integrity and mobility during cell division. The revised manuscript has been carefully edited by the authors and may be recommended for publication in its present form.

We thank the reviewer for the positive assessment of our work.